# Differentially Private Synthetic Data via Foundation Model APIs 1: Images

**Zinan Lin**
Microsoft Research
zinanlin@microsoft.com

**Sivakanth Gopi**
Microsoft Research
sigopi@microsoft.com

**Janardhan Kulkarni**
Microsoft Research
jakul@microsoft.com

**Harsha Nori**
Microsoft Research
hanori@microsoft.com

**Sergey Yekhanin**
Microsoft Research
yekhanin@microsoft.com

## Abstract

Generating *differentially private (DP) synthetic data* that closely resembles the original private data is a scalable way to mitigate privacy concerns in the current data-driven world. In contrast to current practices that train customized models for this task, we aim to *generate DP Synthetic Data via APIs (DPSDA)*, where we treat foundation models as blackboxes and only utilize their inference APIs. Such API-based, training-free approaches are easier to deploy as exemplified by the recent surge in the number of API-based apps. These approaches can also leverage the power of large foundation models which are only accessible via their inference APIs. However, this comes with greater challenges due to strictly more restrictive model access and the need to protect privacy from the API provider.

In this paper, we present a new framework called *Private Evolution (PE)* to solve this problem and show its initial promise on synthetic images. Surprisingly, PE can match or even outperform state-of-the-art (SOTA) methods *without any model training*. For example, on CIFAR10 (with ImageNet as the public data), we achieve FID$\leq$7.9 with privacy cost $\epsilon = 0.67$, significantly improving the previous SOTA from $\epsilon = 32$. We further demonstrate the promise of applying PE on large foundation models such as Stable Diffusion to tackle challenging private datasets with a small number of high-resolution images. The code and data are released at https://github.com/microsoft/DPSDA.

## 1 Introduction

While data-driven approaches have been successful, privacy is a major concern. For example, statistical queries of a dataset may leak sensitive information about individual users (Dwork et al., 2014). Entire training samples can be reconstructed from deep learning models (Haim et al., 2022; Fredrikson et al., 2015; Carlini et al., 2021a; 2023a;b; 2019; 2021b; Choquette-Choo et al., 2021; Tramèr et al., 2022; Wang et al., 2023). *Differential privacy (DP)* is the gold standard in quantifying and mitigating these concerns (Dwork et al., 2006). DP algorithms ensure that information about individual samples in the original data cannot be inferred with high confidence from algorithm outputs.

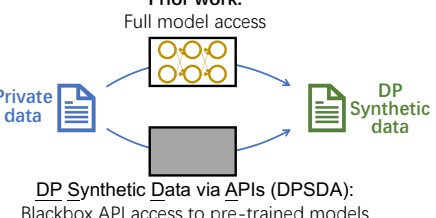

Figure 1: We consider the problem of generating DP synthetic data with API access to pre-trained models without any model training. This is in contrast to prior work which assumes full access to pre-trained models and requires training.

*Differentially private synthetic data* is the holy grail of DP research (Hu et al., 2023; Jordon et al., 2019; Lin et al., 2020; Beaulieu-Jones et al., 2019; Dockhorn et al., 2022; Yin et al., 2022; Yu et al., 2021; He et al., 2022; Li et al., 2021; Ghalebikesabi et al., 2023; Yue et al., 2022; Harder et al., 2023; 2021; Savage, 2023; Lin, 2022; Tang et al., 2023). The goal is to generate a synthetic dataset that is statistically similar to the original data while ensuring DP. The benefits are: (1) Thanks to the post-processing property of DP (Dwork et al., 2014), we can use any existing *non-private* algorithm (e.g., training machine learning (ML) models) as-is on the synthetic data without incurring additional privacy loss. This is more scalable than redesigning and reimplementing every algorithm for DP. (2) Synthetic data can be shared freely with other parties without violating privacy. This is useful in situations when sharing data is necessary, such as when organizations (e.g., hospitals) want to release datasets to support open research

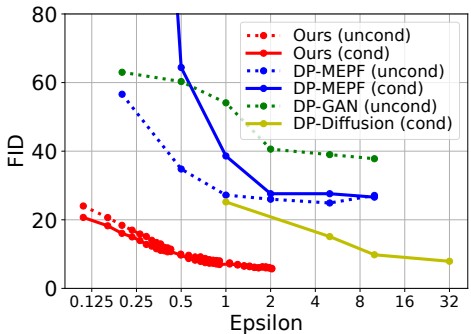

Figure 2: Private Evolution (PE) framework for DP synthetic data. **Left: Intuition of PE.** Though private data and pre-trained generative models have very different distributions, the support of the former is likely to be covered by the support of the latter. We gradually shift the distribution of generated data toward private data through PE. **Right: Algorithm of PE.** We maintain a sample set (population), and iteratively select the most similar ones to the private samples (parents) and mutate them to generate the next population (offspring). The initial population and offspring are generated with foundation model APIs. Parent selection is done in DP using private samples.

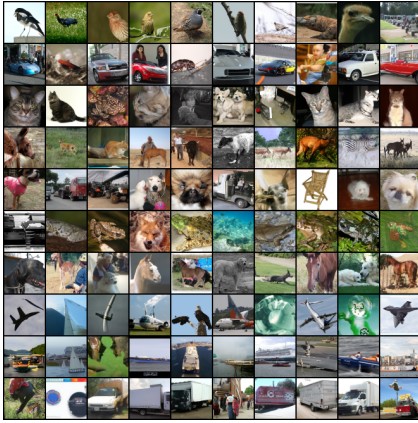

Figure 3: Generated samples on CIFAR10 with $(0.67, 10^{-5})$-DP. Each row corresponds to one class. FID=7.87. See App. J for real and generated images side-by-side.

Figure 4: FID (Heusel et al., 2017) (lower is better) v.s. privacy cost $\epsilon$ on CIFAR10 ($\delta = 10^{-5}$). (Un)cond means (un)conditional generation. Ours achieves the best privacy-quality trade-off compared to DP-MEPF (Harder et al., 2023), DP-GAN, DP-Diffusion (Ghalebikesabi et al., 2023).

initiatives (Beaulieu-Jones et al., 2019; Lin et al., 2020). (3) Since synthetic data is DP, developers can look at the data directly, which makes algorithm development and debugging a lot easier.

At the same time, with the recent advancement of large foundation models, API-based solutions are gaining tremendous popularity, exemplified by the surge of GPT4-based applications. In contrast to the traditional paradigm that trains/fine-tunes customized ML models for each application, API-based solutions treat ML models as blackboxes and only utilize APIs[1] that provide the input/output functions of the models. In fact, many foundation models including GPT4, Bard, and DALLE2 only provide API access without releasing model weights or code. Key reasons for the success of API-based solutions are that APIs offer a clean abstraction of ML and are readily available and scalable. Therefore, implementing and deploying these API-based algorithms is easier and faster even for developers without ML expertise. Such an approach can also leverage powerful foundation models that are only accessible through APIs. Unfortunately, SOTA DP synthetic data algorithms today are still in the old paradigm (Ghalebikesabi et al., 2023; Li et al., 2021): they need a customized training process for each dataset, whose implementation requires significant ML engineering efforts (§ 3.1).

Motivated from these observations, we ask the following ambitious question (Fig. 1):

*Can we generate DP synthetic data using blackbox APIs of foundation models?*

We treat API providers as *untrusted* entities so we also want to protect user privacy from them, i.e., the API queries we make during generation should also be DP. If successful, we can potentially democratize the deployment of DP synthetic data in the industry similar to how API-based solutions have facilitated other applications. This is a challenging task, however, as we do not have access to model weights and gradients by assumption. In this paper, we conduct the first exploration of the potential and limits of this vision on DP synthetic *images*. Surprisingly, we show that not only is such a vision realizable, but that it also has the potential to match or improve SOTA training-based DP synthetic image algorithms despite more restrictive model access. Our contributions are:

---

[1]See https://platform.openai.com/docs/introduction for examples of APIs. For example, a text completion API can complete a text prompt using a foundation model such as GPT4. An image variation API can produce variations of a given image using a foundation model such as DALLE2.

**(1) New problem (§ 3).** We highlight the importance of *DP Synthetic Data via APIs (DPSDA)*. Such algorithms are easy to implement and deploy and can leverage the foundation models behind APIs.

**(2) New framework (§ 4).** We propose *Private Evolution (PE)* algorithm for achieving our goal (Fig. 2). We consider using 2 popular APIs: random generation and sample variation (i.e., generating a sample similar to the given one).[4,5] The key idea is to iteratively use private samples to vote for the most similar samples generated from the blackbox model and ask the blackbox models to generate more of those similar samples. We theoretically prove that the distribution of the generated samples from PE will converge to the private distribution under some modeling assumptions (App. E). PE only requires (existing) APIs of the foundation models, and does not need any model training.

**(3) Experimental results (§ 5).**[2] Some key results are: (a) Surprisingly, without any training, PE can still outperform SOTA training-based DP image generation approaches on some datasets (Figs. 3 and 4). For example, to obtain FID$\leq 7.9$ on CIFAR10 dataset, PE (with blackbox access to an ImageNet-pre-trained model) only needs $\epsilon = 0.67$. In contrast, DP fine-tuning of an ImageNet-pre-trained model (prior SOTA) requires $\epsilon = 32$ (Ghalebikesabi et al., 2023).

(b) We show that PE works even when there is significant distribution shift between private and public data. We create a DP synthetic version (with $\varepsilon = 7.58$) of Camelyon17, a medical dataset for classification of breast cancer metastases, using the same ImageNet-pre-trained model. A downstream classifier trained on our DP synthetic data achieves a classification accuracy of 79.56% (prior SOTA based on DP fine-tuning is 91.1% with $\epsilon = 10$ (Ghalebikesabi et al., 2023)).

(c) We set up new challenging benchmarks that the DP synthetic image literature has not studied before. We show that with powerful foundation models such as Stable Diffusion (Rombach et al., 2022), PE can work with high-resolution (512x512) image datasets with a small size (100 images), which are common in practice but challenging for current DP synthetic image algorithms.

## 2 BACKGROUND AND RELATED WORK

**Differential Privacy (DP).** We say a mechanism $\mathcal{M}$ is $(\epsilon, \delta)$-DP if for any two neighboring datasets $\mathcal{D}$ and $\mathcal{D}'$ which differ in a single entry (i.e., $\mathcal{D}'$ has one extra entry compared to $\mathcal{D}$ or vice versa) and for any set $S$ of outputs of $\mathcal{M}$, we have $\mathbb{P}(\mathcal{M}(\mathcal{D}) \in S) \leq e^{\epsilon} \mathbb{P}(\mathcal{M}(\mathcal{D}') \in S) + \delta$. Intuitively, this means that any single sample cannot influence the mechanism output too much.

**DP synthetic data.** Given a private dataset $\mathcal{D}$, the goal is to generate a DP synthetic dataset $\mathcal{M}(\mathcal{D})$ which is statistically similar to $\mathcal{D}$. One method is to *train generative models from scratch on private data* (Lin et al., 2020; Beaulieu-Jones et al., 2019; Dockhorn et al., 2022) with DP-SGD (Abadi et al., 2016), a DP variant of stochastic gradient descent. Later studies show that *pre-training generative models on public data* before fine-tuning them on private data with DP-SGD (Yin et al., 2022; Yu et al., 2021; He et al., 2022; Li et al., 2021; Ghalebikesabi et al., 2023; Yue et al., 2022) gives better privacy-utility trade-offs due to knowledge transfer from public data (Yin et al., 2022; Ganesh et al., 2023), smaller gradient spaces (Li et al., 2022), or better initialization (Ganesh et al., 2023). This approach achieves SOTA results on several data modalities such as text and images. In particular, DP-Diffusion (Ghalebikesabi et al., 2023) achieves SOTA results on DP synthetic images by pre-training diffusion models (Sohl-Dickstein et al., 2015; Ho et al., 2020) on public datasets and fine-tuning them on the private dataset. Some other methods do not depend on DP-SGD (Jordon et al., 2019; Harder et al., 2023; 2021; Vinaroz et al., 2022; Cao et al., 2021). For example, DP-MEPF (Harder et al., 2023) trains generative models to produce synthetic data that matches the (privatized) statistics of the private features.

Note that all these methods obtain generative models *whose weights are DP*, which can then be used to draw DP synthetic data. It is stronger than our goal which only requires DP synthetic data (Lin et al., 2021). In this paper, we do not do any model training and only produce DP synthetic data.

## 3 DP SYNTHETIC DATA VIA APIS (DPSDA)

### 3.1 MOTIVATION

As discussed in § 2, SOTA DP synthetic data algorithms require training or fine-tuning generative models with DP-SGD. There are some obstacles to deploying them in practice.

*(1) Significant engineering effort.* Deploying normal ML training pipelines is hard; deploying *DP* training pipelines is even harder because most ML infrastructure is not built around this use case.

---

[2]In the experiments of this paper, we only experimented with APIs from local models where the user has full control of the model weights and runs it in a controlled environment.

Recently, there has been significant progress in making DP training more efficient (Li et al., 2021; He et al., 2022) and easy to use (Opacus (Yousefpour et al., 2021) and Tensorflow Privacy). However, incorporating them in new codebases and new models is highly non-trivial. For example, Opacus requires us to implement our own per-sample gradient calculator for new layers. Common layers and loss functions that depend on multiple samples (e.g., batch normalization) are often not supported.

*(2) Inapplicability of API-only models.* It may be appealing to take advantage of the powerful foundation models in DP synthetic data generation. However, due to the high commercial value of foundation models, many companies choose to only release inference APIs of the models but not the weights or code. Examples include popular models such as DALLE 2 (Ramesh et al., 2022) and GPT 3/4 (Brown et al., 2020; OpenAI, 2023) from OpenAI and Bard from Google. In such cases, existing training-based approaches are not applicable.[3]

In contrast, DP synthetic data approaches that only require model inference APIs could potentially be deployed more easily, as they do not require ML or DP expertise to conduct modifications inside the model and require minimal modifications when switching to a different model (as long as they support the same APIs). In addition, such an approach is compatible with both the models running locally and the models behind APIs.

## 3.2 PROBLEM FORMULATION

We now give a formal statement of DPSDA. We first define a core primitive for DP synthetic data.

**DP Wasserstein Approximation (DPWA).** Given a private dataset $S_{\mathrm{priv}} = \{x_i : i \in [N_{\mathrm{priv}}]\}$ with $N_{\mathrm{priv}}$ samples (e.g., images), a distance function $d(\cdot, \cdot)$ between samples and some $p \geq 1$, the goal is to design an $(\epsilon, \delta)$-DP algorithm $\mathcal{M}$ that outputs a synthetic dataset $S_{\mathrm{syn}} = \{x_i' : i \in [N_{\mathrm{syn}}]\}$ with $N_{\mathrm{syn}}$ samples (as a multiset) whose distance to $S_{\mathrm{priv}}$, $W_p(S_{\mathrm{priv}}, S_{\mathrm{syn}})$, is minimized. Here $W_p$ is the Wasserstein $p$-distance w.r.t. the distance function $d(\cdot, \cdot)$ (see App. B for the definition).

**DPSDA.** We want to solve DPWA where $\mathcal{M}$ is given blackbox access to foundation models trained on public data via APIs.[1] API queries should also be $(\epsilon, \delta)$-DP as API providers cannot be trusted.

In some applications, besides the raw samples $x_i$, we may also care about some auxiliary information such as class labels of images. In such cases, we may write $S_{\mathrm{priv}} = \{(x_i, y_i) : i \in [N_{\mathrm{priv}}]\}$ (and $S_{\mathrm{syn}} = \{(x_i', y_i') : i \in [N_{\mathrm{syn}}]\})$ where $y_i$ (and $y_i'$) is the auxiliary information of $i$-th sample.

When the distance function $d(\cdot, \cdot)$ is $\ell_2$ (in the sample space), DPWA is closely related to DP Clustering (Ghazi et al., 2020; Su et al., 2016; Balcan et al., 2017) and DP Heatmaps (Ghazi et al., 2022). But a direct application of them does not work in our setting; see App. H for more discussions.

## 3.3 SCOPE OF THIS WORK

**Data type.** In this paper, we instantiate the above framework on *images*. We consider both unconditional (i.e., no $y_i$) and conditional generation (e.g., $y_i$ can be image categories such as cats/dogs).

**APIs.** In our algorithm design and experiments, we use 2 APIs, both of which are either directly provided in the APIs of popular models (e.g., DALLE 2,[4] Stable Diffusion[5]) or can be easily implemented by adapting current APIs (e.g., using appropriate text prompts in GPT APIs[1]):

(1) RANDOM_API $(n)$ that randomly generates $n$ samples. Some APIs accept condition information such as text prompts in text-to-image generation,[4,5] which we omit in the notation for simplicity.

(2) VARIATION_API $(S)$ that generates variations for each sample in $S$. For images, it means to generate similar images to the given one, e.g., with similar colors or objects.[4,5] Some APIs also support setting the variation degree: VARIATION_API $(S, v)$, where larger $v$ indicates more variation.[6]

## 4 PRIVATE EVOLUTION (PE)

Foundation models have a broad and general model of our world from their extensive training data. Therefore, we expect that foundation models can generate samples close to private data with non-

---

[3]Some companies also provide model fine-tuning APIs, e.g., https://platform.openai.com/docs/guides/fine-tuning. However, they do not support DP fine-tuning and do not provide gradients. Also, uploading sensitive data to these APIs controlled by other companies can lead to privacy violations.

[4]See https://platform.openai.com/docs/guides/images/usage.

[5]See https://huggingface.co/docs/diffusers/api/pipelines/stable_diffusion/overview.

[6]If this is not implemented, we can simply compose the VARIATION_API $v$ times to achieve it.

negligible probability. The challenge is that by naively calling the APIs, the probability of drawing such samples is quite low. We need a way to *guide* the generation towards private samples.

Inspired by *evolutionary algorithms (EA)* (Davis, 1987) (App. C), we propose *Private Evolution (PE)* framework for generating DP synthetic data via APIs. See Fig. 2 for the intuition behind PE. The complete algorithm is in Alg. 1. Below, we discuss the components in detail.

---

**Algorithm 1:** Private Evolution (PE)

**Input** : Private samples: $S_{\mathrm{priv}} = \{x_i\}_{i=1}^{N_{\mathrm{priv}}}$
        Number of iterations: $T$
        Number of generated samples: $N_{\mathrm{syn}}$
        Noise multiplier for DP Nearest Neighbors Histogram: $\sigma$
        Threshold for DP Nearest Neighbors Histogram: $H$
**Output:** Synthetic data: $S_{\mathrm{syn}}$

1   $S_1 \leftarrow \mathsf{RANDOM\_API}\left(N_{\mathrm{syn}}\right)$
2   **for** $t \leftarrow 1, \ldots, T$ **do**
3      $histogram_t \leftarrow \mathsf{DP\_NN\_HISTOGRAM}\left(S_{\mathrm{priv}}, S_t, \sigma, H\right)$         `// See Alg. 2`
4      $\mathcal{P}_t \leftarrow histogram_t/\mathrm{sum}(histogram_t)$     `// `$\mathcal{P}_t$` is a distribution on `$S_t$
5      $S_t' \leftarrow$ draw $N_{\mathrm{syn}}$ samples with replacement from $\mathcal{P}_t$     `// `$S_t'$` is a multiset`
6      $S_{t+1} \leftarrow \mathsf{VARIATION\_API}\left(S_t'\right)$
7   **return** $S_T$

---

**Algorithm 2:** DP Nearest Neighbors Histogram (DP_NN_HISTOGRAM)

**Input** : Private samples: $S_{\mathrm{priv}}$
        Generated samples: $S = \{z_i\}_{i=1}^n$
        Noise multiplier: $\sigma$
        Threshold: $H$
        Distance function: $d\left(\cdot, \cdot\right)$
**Output:** DP nearest neighbors histogram on $S$

1   $histogram \leftarrow [0, \ldots, 0]$
2   **for** $x_{\mathrm{priv}} \in S_{\mathrm{priv}}$ **do**
3      $i = \arg\min_{j \in [n]} d\left(x_{\mathrm{priv}}, z_j\right)$
4      $histogram[i] \leftarrow histogram[i] + 1$
5   $histogram \leftarrow histogram + \mathcal{N}\left(0, \sigma I_n\right)$        `// Add noise to ensure DP`
6   $histogram \leftarrow \max\left(histogram - H, 0\right)$     `// 'max', '-' are element-wise`
7   **return** $histogram$

---

**Initial population (Line 1).** We use RANDOM_API to generate the initial population. If there is public information about the private samples (e.g., they are dog images), we can use this information as prompts to the API to seed a better initialization.

**Fitness function (Line 3 or Alg. 2).** We need to evaluate how useful each sample in the population is for modeling the private distribution. Our idea is that, if a sample in the population is surrounded by many private samples, then we should give it a high score. To implement this, we define the fitness function of a sample $x$ as *the number of private samples whose nearest neighbor in the population is $x$*. A higher fitness value means that more private samples are closest to it. More details are below:

*(1) Distance function (Line 3, Alg. 2).* To define "nearest neighbor", we need a distance function that measures the similarity of two samples. A naive way is to use $\ell_2$ distance $d\left(x, z\right) = \|x - z\|_2$, where $x$ is from the private dataset and $z$ is from the population. However, it is well-known that $\ell_2$ distance on pixel space is not a good metric for images. For example, a small shift of an object can result in a high $\ell_2$ distance. We therefore compute the $\ell_2$ distance in the embedding space:

$$d\left(x, z\right) = \|\Phi\left(x\right) - \Phi\left(z\right)\|_2 \tag{1}$$

where $\Phi$ is a network for extracting image embeddings such as inception embedding (Szegedy et al., 2016) or CLIP embedding (Radford et al., 2021).

*(2) Lookahead.* The above approach gives high scores to the good samples in the *current* population. However, as we will see later, these good samples will be modified through VARIATION_API for the next population. Therefore, it is better to "look ahead" to compute the distance based on the modified samples as if they are kept in the population. We modify Eq. (1) to

compute the distance between the embedding of $x$ and the *mean embedding of $k$ variations of $z$*: $d(x, z) = \left\| \Phi(x) - \frac{1}{k} \sum_{i=1}^{k} \Phi(z^i) \right\|_2$, where $k$ is called *lookahead degree*, and $z^1, \ldots, z^k$ are variations of $z$ obtained via VARIATION_API.

*(3) Noise for DP (Line 5, Alg. 2).* Because this step utilizes private samples, we need to add noise to ensure DP. We add i.i.d. Gaussian noise from $\mathcal{N}(0, \sigma)$. The privacy analysis is presented in § 4.3.

*(4) Thresholding (Line 6, Alg. 2).* When the number of generated samples is large, the majority of the histogram will be DP noise added above. To make the signal-noise ratio larger, we set a threshold $H$ to each bin of the histogram. Similar ideas have been used in DP set union (Gopi et al., 2020).

In summary, we called the above fitness function *DP Nearest Neighbors Histogram*. Note that it is not the traditional "histogram" on a continuous space that requires binning. Instead, it is a histogram built on the generated samples: the value of $i$-th bin means the (privatized) number of private samples whose nearest neighbor among the generated ones is the $i$-th sample. See App. A for related work.

**Parent selection (Line 5).** We sample from the population according to the DP Nearest Neighbors Histogram so that a sample with more private samples around is more likely to be selected.

**Offspring generation (Line 6).** We use VARIATION_API to get variants of the parents as offsprings.

Please see App. E for the convergence analysis of PE.

### 4.1 CONDITIONAL GENERATION

The above procedure is for unconditional generation. To support conditional generation, i.e., each generated sample is associated with a label such as an image class (e.g., cats v.s. dogs), we take a simple approach: *we repeat the above process for each class of samples in the private dataset separately*. See Alg. 3 for the full algorithm.

### 4.2 GENERATING UNLIMITED NUMBER OF SAMPLES

Our algorithm in Alg. 1 is preset with a fixed number of generated samples $N_{\text{syn}}$. What if users want more samples afterward? In prior training-based methods (Ghalebikesabi et al., 2023), this is easy to achieve: one can draw an arbitrary number of samples from the trained generative models without additional privacy cost. In this section, we want to highlight that PE can also do that, again *with API access only*. We can simply generate an unlimited number of samples by calling variation API multiple times, each with the generated dataset as input: $[\text{VARIATION\_API}(S_{\text{syn}}), \ldots, \text{VARIATION\_API}(S_{\text{syn}})]$. In 5.1.3, we will see that this simple algorithm is sufficient to provide more useful samples for downstream applications.

### 4.3 PRIVACY ANALYSIS

Unlike the analysis of DP-SGD which requires complicated DP composition theorems (e.g., Gopi et al. (2021); Mironov (2017)) due to subsampling, PE does not have subsampling steps and therefore the privacy analysis is rather straightforward. The DP guarantee of the **unconditional version of PE (Alg. 1)** can be reasoned as follows:

- **Step 1: The sensitivity of DP Nearest Neighbors Histogram (Lines 1 to 4 in Alg. 2).** Each private sample only contributes one vote. If we add or remove one sample, the resulting histogram will change by 1 in the $\ell_2$ norm. Therefore, the sensitivity is 1.
- **Step 2: Regarding each PE iteration as a Gaussian mechanism.** Line 5 adds i.i.d. Gaussian noise with standard deviation $\sigma$ to each bin. This is a standard Gaussian mechanism (Dwork et al., 2014) with noise multiplier $\sigma$.
- **Step 3: Regarding the entire PE algorithm as $T$ adaptive compositions of Gaussian mechanisms**, as PE is simply applying Alg. 2 $T$ times sequentially.
- **Step 4: Regarding the entire PE algorithm as one Gaussian mechanism with noise multiplier $\sigma/\sqrt{T}$.** It is a standard result from Dong et al. (2022) (see Corollary 3.3 therein).
- **Step 5: Computing DP parameters $\epsilon$ and $\delta$.** Since the problem is simply computing $\epsilon$ and $\delta$ for a standard Gaussian mechanism, we use the formula from Balle & Wang (2018) directly.

For the **conditional version of PE (Alg. 3)**, since it does the unconditional version for each class separately (discussed in § 4.1), adding or removing one sample will only influence the results of one class. For that class, the impact due to the added/removed sample is also bounded, as seen in the privacy analysis above. Therefore, Alg. 3 is also DP. In fact, we can show that *the privacy guarantee of Alg. 3 is the same as Alg. 1, and it protects the labels of samples in the same level as DP-SGD (Ghalebikesabi et al., 2023)*. Please refer to App. D for more details.

This privacy analysis implies that releasing all the (intermediate) generated sets $S_1, \ldots, S_T$ also satisfies the same DP guarantees. Therefore PE provides the same privacy even from the API provider.

# 5 EXPERIMENTS

In § 5.1, we compare PE with SOTA training-based methods on standard benchmarks to understand its promise and limitations. In § 5.2, we present proof-of-concept experiments to show how PE can utilize the power of large foundation models. We did (limited) hyper-parameter tunings in the above experiments; following prior DP synthetic data work (Yu et al., 2021; Ghalebikesabi et al., 2023), we ignore the privacy cost of hyper-parameter tuning. However, as we will see in the ablation studies (§ 5.3 and App. N), PE stably outperforms SOTA across a wide range of hyper-parameters, and the results can be further improved with better hyper-parameters than what we used. Detailed hyper-parameter settings and more results such as *generated samples* and *their nearest images in the private dataset* are in Apps. J to L.

## 5.1 COMPARISONS TO STATE-OF-THE-ART

**Public information.** We use standard benchmarks (Ghalebikesabi et al., 2023) which treat ImageNet (Deng et al., 2009) as public data. For fair comparisons, we only use ImageNet as public information in PE: *(1) Pre-trained model.* Unlike the SOTA (Ghalebikesabi et al., 2023) which trains customized diffusion models, we simply use public ImageNet pre-trained diffusion models (pure image models without text prompts) (Nichol & Dhariwal, 2021). RANDOM_API and VARIATION_API are implemented using the same pre-trained model (see App. J). *(2) Embedding (Eq. (1)).* We use ImageNet inception embedding (Szegedy et al., 2016). PE is not sensitive to embedding choice though and we get good results even with CLIP embeddings (Fig. 41 in App. N).

**Baselines.** We compare with DP-Diffusion (Ghalebikesabi et al., 2023), DP-MEPF (Harder et al., 2023), and DP-GAN (Harder et al., 2023; Goodfellow et al., 2020). DP-Diffusion (Ghalebikesabi et al., 2023) is the current SOTA that achieves the best results on these benchmarks. Baseline results are taken from their paper.

**Outline.** We test PE on private datasets that are either similar to or differ a lot from ImageNet in § 5.1.1 and 5.1.2. We demonstrate that PE can generate an unlimited number of useful samples in § 5.1.3. We show that PE is computationally cheaper than train-based methods in App. P.

Figure 5: Downstream classification accuracy (higher is better) on CIFAR10 ($\delta = 10^{-5}$). The baseline results are taken from Ghalebikesabi et al. (2023). Two "ensemble" lines are from ensembles of 5 classifiers. The other two lines show the average accuracy of 5 independently trained classifiers with error bars. Our PE achieves better accuracy across almost all settings with smaller privacy costs.

### 5.1.1 MODERATE

DISTRIBUTION SHIFT (IMAGENET $\rightarrow$ CIFAR10)

We treat CIFAR10 (Krizhevsky et al., 2009) as private data. Given that both ImageNet and CIFAR10 are natural images, it is a relatively easy task for PE (and also for the baselines). Figs. 3 to 5 show the results. Surprisingly, despite the fact that we consider strictly more restrictive model access and do not need training, PE still outperforms the SOTA training-based methods. Details are below.

**Sample quality.** Fig. 4 shows the trade-off between privacy cost and FID, a popular metric for image quality (Heusel et al., 2017). For either conditional or unconditional generation, PE outperforms the baselines significantly. For example, to reach FID≤ 7.9, Ghalebikesabi et al. requires $\epsilon = 32$, Harder et al. cannot achieve it even with infinity $\epsilon$, whereas our PE only needs $\epsilon = 0.67$.

**Downstream classification accuracy.** We train a downstream WRN-40-4 classifier (Zagoruyko & Komodakis, 2016) from scratch on 50000 generated samples and test the accuracy on CIFAR10 test set. This simulates how users would use synthetic data, and a higher accuracy means better utility. Fig. 5 shows the results (focus on the left-most points with num of generated samples = 50000 for now). Harder et al. (2023) achieves 51% accuracy with $\epsilon = 10$ (not shown). Compared with the SOTA (Ghalebikesabi et al., 2023), PE achieves better accuracy (+6.1%) with less privacy cost. Further with an ensemble of 5 classifiers trained on the same data, PE is able to reach an accuracy of 84.8%. For reference, the SOTA DP classifier pre-trained on ImageNet (without DP) and fine-tuned on CIFAR10 (with DP) (De et al., 2022) achieves 94.8% and 95.4% accuracies with epsilon=1 and

Real          Generated $((9.92, 3 \cdot 10^{-6})$-DP, FID=10.66$)$

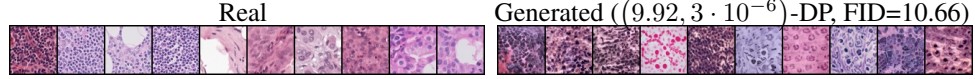

Figure 6: Real and generated images from Camelyon17. More in App. K.

2 respectively. It is not surprising that DP classifiers outperform PE (and other DP synthetic data approaches) on classification tasks, as DP classifiers are targeted at and optimized for a single task whereas DP synthetic data is general-purpose.

The above results suggest that when private and public images are similar, PE is a promising framework given its better privacy-utility trade-off and the API-only requirement.

### 5.1.2 LARGE DISTRIBUTION SHIFT (IMAGENET → CAMELYON17)

Next, we consider a hard task for PE, where the private dataset is very different from ImageNet. We use Camelyon17 dataset (Bandi et al., 2018; Koh et al., 2021) as private data which contains 302436 images of histological lymph node sections with labels on whether it has cancer (real images in Figs. 6 and 22). Despite the large distribution shift, training-based methods can update the model weights to adapt to the private distribution (given enough samples). However, PE can only draw samples from APIs as is.

We find that even in this challenging situation, PE can still achieve non-trivial results. Following Ghalebikesabi et al. (2023), we train a WRN-40-4 classifier from scratch on 302436 generated samples and compute the test accuracy. We achieve 80.33% accuracy with $(10.00, 3 \cdot 10^{-6})$-DP. Prior SOTA (Ghalebikesabi et al., 2023) is 91.1% with $(10, 3 \cdot 10^{-6})$-DP. Random guess is 50%. Fig. 6 (more in Fig. 21) shows that generated images from PE are very similar to Camelyon17 despite that the pre-trained model is on ImageNet. Fig. 23 further shows how the generated images are gradually moved towards Camelyon17 across iterations.

**Why PE works under large distribution shifts.** Even though the diffusion model is trained on natural images, the support of the generated distribution spans the entire sample space. PE is effective in guiding the model to generate samples from the region that is low-density in the original pre-trained distribution but high-density in the private distribution. See App. K for more explanations.

**Limitation.** These results demonstrate the effectiveness of PE. But when public models that are similar to private data are not available and when there is enough private data, the traditional training-based methods are still more promising at this point if the privacy-utility trade-off is the only goal. However, given the benefit of API-only assumption and the non-trivial results that PE already got, it is worth further exploiting the potential of PE in future work. Indeed, we expect these results can be improved with further refinement of PE (App. N).

### 5.1.3 GENERATING UNLIMITED NUMBER OF SAMPLES

We use the approach in § 4.2 to generate more synthetic samples from § 5.1.1 and train classifiers on them. The results are in Fig. 5. Similar as Ghalebikesabi et al. (2023), the classifier accuracy improves as more generated samples are used. With an ensemble of 5 classifiers, we reach 89.13% accuracy with 1M samples. *This suggests that PE has the same capability as training-based methods in generating an unlimited number of useful samples.*

We also see two interesting phenomena: *(1) The gap between PE and DP-Diffusion diminishes as more samples are used.* We hypothesize that it is due to the limited improvement space: As shown in Ghalebikesabi et al. (2023), even using an ImageNet pre-trained classifier, the best accuracy DP-Diffusion achieves is close to the best points in Fig. 5. *(2) The benefit of PE is more evident over ensembling, especially when having more generated samples.* We hypothesize it is due to different ways of generating more samples. In Ghalebikesabi et al. (2023), the newly generated samples are from the same distribution as the first 50000 samples. In contrast, the newly generated samples in PE are from a different distribution (see § 4.2), which could be more diverse and therefore are more beneficial for ensembling approaches.

### 5.2 MORE CHALLENGING BENCHMARKS WITH LARGE FOUNDATION MODELS

We demonstrate the feasibility of applying PE on large foundation models with Stable Diffusion (Rombach et al., 2022).

**Data.** Ideally we want to experiment with a dataset that has no overlap with Stable Diffusion's training data.[7] We take the safest approach: we construct two datasets with photos of the author's

---

[7]The training set of Stable Diffusion is public. However, it is hard to check if a public image or its variants (e.g., cropped, scaled) have been used to produce images in it. Therefore, we resort to our own private data.

Real     Generated $((6.62, 10^{-3})$-DP)     Real     Generated $((6.62, 10^{-3})$-DP)

Figure 8: Real & generated images from Cat Cookie (left) and Cat Doudou (right). More in App. L.

two cats that have never been posted online. Each dataset has 100 512x512 images. Such high-resolution datasets with a small number of samples represent a common need in practice (e.g., in health care), but are challenging for DP synthetic data: to the best of our knowledge, no prior training-based methods have reported results on datasets with a similar resolution or number of samples. The dataset is released at `https://github.com/microsoft/DPSDA` as a new benchmark. See App. L for all images.

**API implementation.** We use off-the-shelf Stable Diffusion APIs (see App. L).

**Results.** We run PE for these two datasets with the same hyperparameters. Fig. 8 show examples of generated images for each of the cat datasets. We can see that Private Evolution correctly captures the key characteristics of these two cats. See App. L for all generated images.

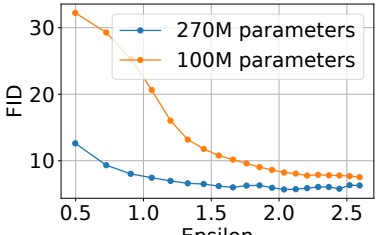

Figure 7: Ablation studies on the pre-trained model. Both are public diffusion models trained on ImageNet (Nichol & Dhariwal, 2021). The 270M network is conditional, whereas the 100M network is unconditional.

## 5.3 ABLATION STUDIES

**Pre-trained network.** Fig. 7 shows the results with two different ImageNet pre-trained networks: one is larger (270M) with ImageNet class labels as input; the other one is smaller (100M) without label input (see App. I for implementation details). In all experiments in § 5.1, we used the 270M network. Two takeaways are: (1) The 270M network (trained on the same dataset) improves the results. This is expected as larger and more powerful models can learn public distributions better. This suggests the potential of PE with future foundation models with growing capabilities. (2) Even with a relatively weak model (100M), PE can still obtain good results that beat the baselines (though with a slower convergence speed), suggesting the effectiveness of PE.

More ablation studies on the lookahead degree $k$, the number of generated samples $N_{\text{syn}}$, the threshold $H$, and the embedding network are in App. N, where we see that PE obtains good results across a wide range of hyper-parameters, and the main results can be improved with better hyper-parameters.

## 6 LIMITATIONS AND FUTURE WORK

**Algorithms.** (1) We did not take the number of API calls into account when optimizing PE. One future work is to optimize the number of API calls along with privacy-utility tradeoffs. (2) We considered two APIs: RANDOM_API and VARIATION_API. It is interesting to consider PE variants that leverage the large set of APIs.[4,5] (3) When the distributions of private data and foundation models are too different, PE achieved non-trivial classification results, but was still worse than SOTA (§ 5.1.2). It is interesting to understand the limits of PE and explore potential improvements. (4) PE requires an embedding network (Eq. (1)) that projects the samples into a space for measuring the similarity between samples. While for images there are plenty of open-source embedding networks to choose, it may not be the case for other modalities. (5) Recent papers show the phenomenon of Model Autophagy Disorder (MAD), where repeatedly *training* the next generative model using synthetic data from the previous one can result in degraded sample quality (Alemohammad et al., 2023). While PE also repeatedly uses synthetic data to create new synthetic data in the main loop (Alg. 1), it is different in two aspects: (a) Instead of purely relying on the synthetic data, PE utilizes the signals from private data to guide the generation; (b) PE does repeated *inference* instead of repeated *training*. It would be interesting to study the MAD effect in the context of PE. (6) Solving DPSDA in the Local/Shuffle DP model and in federated learning settings.

**Applications.** (1) New privacy-preserving vision applications that were previously challenging but are now possible due to PE's capability of generating high-resolution DP synthetic images with small dataset sizes. (2) The use of PE in other data modalities beyond images such as texts, tabular data, and time series data. (3) Besides DP, there are other parallel/orthogonal privacy concerns, notions, and metrics (Issa et al., 2019; Lin et al., 2022; 2023b; Commission). It is interesting to study if PE can be used to generate privacy-preserving synthetic data with respect to these privacy notations.

## 7 ETHICS STATEMENT

PE uses the APIs of *pre-trained* models. The DP guarantee of PE is rigorous for the data used in the PE algorithm (i.e., $S_{\mathrm{priv}}$). That being said, PE does not address the privacy of *pre-training* data of foundation models, which is a different goal. PE has no control over the pre-training data—any privacy breaches in the pre-training data are attributable to the data holder (e.g., leaking the data publicly) or the foundation model developer (e.g., using data without permission). However, as a PE user, it is advisable to ensure no overlap between the pre-training data and $S_{\mathrm{priv}}$ for liability reasons. Depending on whether the APIs are from *blackbox models*, which can only be accessed through APIs (e.g., DALLE3), or *local models*, whose weights and architectures are accessible by the users (e.g., Stable Diffusion), it has different implications.

- *Using APIs from blackbox models.* Since most blackbox models do not reveal their training dataset, it is safer to only consider $S_{\mathrm{priv}}$ that was never been shared or posted online. For instance, a hospital who wants to share a DP synthetic version of its proprietary medical records can safely run PE if it has never released these medical records to any other party, making it impossible for those records to be in the pre-training data of any foundation model.
- *Using APIs from local models.* For local models, we have full control over the model weights and architectures. We can pre-train the models on data that surely has no overlap with the private data. In all experiments of the paper, we use local models including Improved Diffusion (Nichol & Dhariwal, 2021) and Stable Diffusion (Rombach et al., 2022). We directly take the pre-trained models from prior work, and we make sure that the private data and the pre-training data have no overlap.

### ACKNOWLEDGEMENT

The authors would like to thank the anonymous reviewers for their valuable feedback and suggestions. The authors would also like to thank Sepideh Mahabadi for the insightful discussions and ideas, and Sahra Ghalebikesabi for the tremendous help in providing the experimental details of DP-Diffusion (Ghalebikesabi et al., 2023). The authors would like to extend their heartfelt appreciation to Cat Cookie and Cat Doudou for generously sharing their adorable faces in the new dataset, as well as to Wenyu Wang for collecting and pre-processing the photos.

[†] This paper is the full version of our previous workshop paper (Lin et al., 2023a).

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

## A  MORE RELATED WORK

**DP data selection.** One key component in PE is to use private samples to select similar generated *samples* (Alg. 2). Prior work studied similar problems in different applications.

In the federated learning setting, Hou et al. (2023) select public *datasets* that are similar to the clients' data, and then pre-train the model on the selected datasets before federated fine-tuning. Hong et al. (2022) first cluster public data, and then use clients' data to select the closest cluster centers (using a similar histogram approach), so that models trained on the selected *clusters* can have better performance for the clients. In contrast to these works, PE selects generated samples at a *sample-level*, so that we can improve the generated data in a more fine-grained manner.

Similar to ours, Yu et al. (2023) also conduct *sample-level* data selection. They select public *samples* that are similar to the private data for pre-training the model, before DP fine-tuning the model on the private data (Yu et al., 2023). Their data selection rule does not provide guarantees on the distance between the selected samples and the private samples, whereas our selection provides distribution convergence guarantees (App. E).

Nevertheless, given that all these methods deal with DP data selection, they can be interchangeably used in each application. It would be interesting to study such extensions in future work.

Furthermore, we apply such data selection *iteratively* on the generated data. Together with the use of foundation model APIs, we can generate DP synthetic data, which is not studied in prior work.

## B  DEFINITION OF WASSERSTEIN DISTANCE

Wasserstein distance is a widely used metric in designing (Arjovsky et al., 2017) and evaluating (Heusel et al., 2017) generative models. Given probability distributions $\mu, \nu$ on a metric space, the Wasserstein distance w.r.t. to a distance function $d(\cdot, \cdot)$ is defined as $W_p(\mu, \nu) = \inf_\gamma \left[ \mathbb{E}_{(x,y)\sim\gamma} d(x,y)^p \right]^{1/p}$ where the infimum is over all couplings $\gamma$ of $\mu, \nu$. Also given discrete point sets $S, T$, we use $W_p(S, T)$ to denote the $W_p$-distance between uniform distributions on $S$ and $T$.

## C  A BRIEF INTRODUCTION TO EVOLUTIONARY ALGORITHMS

Evolutionary algorithms (Davis, 1987) are inspired by biological evolution, and the goal is to produce samples that maximize an *objective value*. It starts with an *initial population* (i.e., a set of samples), which is then iteratively updated. In each iteration, it selects *parents* (i.e., a subset of samples) from the population according to the *fitness function* which describes how useful they are in achieving better *objective values*. After that, it generates *offsprings* (i.e., new samples) by modifying the parents, hoping to get samples with better objective values, and puts them in the population. By doing so, the population will be guided towards better objective values.

We cannot directly apply existing EA algorithms to our problem. Firstly, our objective is to produce a set of samples that are *jointly* optimal (i.e., closer to the private distribution, § 3.2), instead of optimizing an objective calculated from *individual* samples in typical EA problems. In addition, the differential privacy requirement and the restrictive model API access are unique to our problem. These differences require us to redesign all components of EA.

## D  MORE DETAILS ON PRIVATE EVOLUTION

Alg. 3 shows the full algorithm that supports both unconditional data and conditional data. For simplicity, we assume that the private dataset $S_{\text{priv}}$ is balanced, i.e., it has an equal number of samples in each label class. Otherwise, we can first estimate the counts using the Laplace mechanism and use these counts to generate synthetic data of appropriate size in each label class.

**Privacy analysis.** For ease of understanding the privacy guarantee of Alg. 3, we can consider a modified version of Alg. 3 as in Alg. 4, where we switch the order of the two for loops over $t$ and $c$. Apparently, this modified algorithm gives the same outcome as Alg. 3. The only lines that touch

---

**Algorithm 3:** Private Evolution (PE) for both labeled and unlabeled data.

---

**Input:** The set of private classes: $C$     ($C = \{0\}$ if for unconditional generation)

Private samples: $S_{\text{priv}} = \{(x_i, y_i)\}_{i=1}^{N_{\text{priv}}}$, where $x_i$ is a sample and $y_i \in C$ is its label

Number of iterations: $T$

Number of generated samples: $N_{\text{syn}}$   (assuming $N_{\text{syn}} \bmod |C| = 0$)

Noise multiplier for DP Nearest Neighbors Histogram: $\sigma$

Threshold for DP Nearest Neighbors Histogram: $H$

1   $S_{\text{syn}} \leftarrow \emptyset$
2   **for** $c \in C$ **do**
3     $private\_samples \leftarrow \{x_i | (x_i, y_i) \in S_{\text{priv}} \text{ and } y_i = c\}$
4     $S_1 \leftarrow \textsf{RANDOM\_API}(N_{\text{syn}}/|C|)$
5     **for** $t \leftarrow 1, \ldots, T$ **do**
6       $histogram_t \leftarrow \textsf{DP\_NN\_HISTOGRAM}(private\_samples, S_t, \sigma, H)$      `// See Alg. 2`
7       $\mathcal{P}_t \leftarrow histogram_t / \text{sum}(histogram_t)$    `// `$\mathcal{P}_t$` is a distribution on `$S_t$
8       $S'_t \leftarrow$ draw $N_{\text{syn}}/|C|$ samples with replacement from $\mathcal{P}_t$   `// `$S'_t$` is a multiset`
9       $S_{t+1} \leftarrow \textsf{VARIATION\_API}(S'_t)$
10    $S_{\text{syn}} \leftarrow S_{\text{syn}} \cup \{(x, c) | x \in S_T\}$
11 **return** $S_{\text{syn}}$

---

private data are Lines 5 and 6. The input to these lines is the entire private dataset, and the output is a histogram with size $N_{\text{syn}}$. Same as step 1 in the analysis of Alg. 1 (§ 4.3), each private sample only contributes one vote in the histogram. If we add or remove one sample, the resulting histogram will change by 1 in the $\ell_2$ norm. Therefore, the sensitivity of these lines is 1. The following privacy analysis follows exactly the same as steps 2-5 in § 4.3, and therefore, **the privacy guarantee of Alg. 3 is the same as Alg. 1.** It is important to emphasize that even though Alg. 3 utilizes the labels of the samples directly, the above analysis means that **Alg. 3 provides privacy protection to the label assignment of the samples (i.e., the labels each sample has) in the same way as the DP-SGD-fine-tuneing-based algorithms** (Ghalebikesabi et al., 2023).

---

**Algorithm 4:** Private Evolution (PE) for both labeled and unlabeled data. (Modified from Alg. 3 for the ease of privacy analysis.)

---

**Input:** The set of private classes: $C$     ($C = \{0\}$ if for unconditional generation)

Private samples: $S_{\text{priv}} = \{(x_i, y_i)\}_{i=1}^{N_{\text{priv}}}$, where $x_i$ is a sample and $y_i \in C$ is its label

Number of iterations: $T$

Number of generated samples: $N_{\text{syn}}$   (assuming $N_{\text{syn}} \bmod |C| = 0$)

Noise multiplier for DP Nearest Neighbors Histogram: $\sigma$

Threshold for DP Nearest Neighbors Histogram: $H$

1   $S_{\text{syn}} \leftarrow \emptyset$
2   $S_1^c \leftarrow \textsf{RANDOM\_API}(N_{\text{syn}}/|C|)$ for each $c \in C$
3   $private\_samples^c \leftarrow \{x_i | (x_i, y_i) \in S_{\text{priv}} \text{ and } y_i = c\}$ for each $c \in C$
4   **for** $t \leftarrow 1, \ldots, T$ **do**
5     **for** $c \in C$ **do**
6       $histogram_t^c \leftarrow \textsf{DP\_NN\_HISTOGRAM}(private\_samples^c, S_t^c, \sigma, H)$      `// See Alg. 2`
7     **for** $c \in C$ **do**
8       $\mathcal{P}_t \leftarrow histogram_t^c / \text{sum}(histogram_t^c)$    `// `$\mathcal{P}_t$` is a distribution on `$S_t^c$
9       $S'_t \leftarrow$ draw $N_{\text{syn}}/|C|$ samples with replacement from $\mathcal{P}_t$   `// `$S'_t$` is a multiset`
10      $S_{t+1}^c \leftarrow \textsf{VARIATION\_API}(S'_t)$
11 $S_{\text{syn}} \leftarrow S_{\text{syn}} \cup \{(x, c) | x \in S_T^c, c \in C\}$
12 **return** $S_{\text{syn}}$

---

# E  THEORETICAL EVIDENCE FOR CONVERGENCE OF PE

In this section, we will give some intuition for why PE can solve DPWA.

**Convergence of Non-Private Evolution.** We first analyze Alg. 1 when no noise is added to the histograms (i.e., we set $\sigma = 0$ and $H = 0$ in Line 3). We show that in this case, the evolution algorithm does converge to the private distribution in $O(d)$ iterations where $d$ is the dimension of the embedding space. Under some reasonable modeling assumptions (see App. F), we prove the following theorem. Here $D$ is the diameter of $S_{\text{priv}}$, $L \approx$ number of variations of each point in $S'_t$ that are added to $S_{t+1}$ in Line 6 of Alg. 1.

**Theorem 1.** *Assume that* $\log L \ll d$.[8] *With probability* $\geq 1-\tau$, *the non-private evolution algorithm (Alg. 1 with* $\sigma = H = 0$*) outputs* $S_{\text{syn}}$ *with Wasserstein distance* $W_p(S_{\text{priv}}, S_{\text{syn}}) \leq \eta$ *after* $T$ *iterations*[9] $\forall p \in [1, \infty]$ *whenever*

$$T \gg \frac{d \log(D/\eta)}{\log L} + \log(N_{\text{priv}}/\tau). \tag{2}$$

This theorem is nearly tight. In each iteration of PE, we get the voting information which is about $\tilde{O}(N_{\text{priv}} \log(L))$ bits. To converge to $S_{\text{priv}}$, we need at least $\tilde{\Omega}(N_{\text{priv}}d)$ bits of information. Therefore we do require at least $\tilde{\Omega}(d/\log L)$ iterations to converge. Here is some intuition for how the proof of Thm. 1 works. Fix some private point $x \in S_{\text{priv}}$. Let $z^* \in S_t$ be its closest point in $S_t$. In $S_{t+1}$, we generate variations of $z^*$ using the VARIATION_API$(z^*)$. We then prove that if $\|x - z^*\| \geq \eta$, then one of the variations will get closer to $x$ than $z^*$ by a factor of $(1 - (\log L)/d)$ with constant probability. Repeating this for $T$ iterations as in Eq. (2), will bring some point in $S_T$ $\eta$-close to $x$.

**Convergence of Private Evolution.** To get some intuition on the working of PE in the presence of noise, we make a simplifying assumption. We will assume that there are $B$ identical copies of each private point in $S_{\text{priv}}$. We call $B$ as multiplicity. Note that for any DP algorithm to converge to $S_{\text{priv}}$, we need that the private data is well clustered with some minimum cluster size. Any cluster with too few points cannot be represented in $S_{\text{syn}}$, because that would violate DP. And when there is a cluster of $B$ private points and the generated points in $S_t$ are still far from this cluster, then it is likely that all the $B$ private points will have a common closest point in $S_t$[10], i.e., they all vote for the same point in $S_t$ as a single entity. Therefore multiplicity is a reasonable modeling assumption to make to understand the working of PE. Note that, actually it is very easy to find $S_{\text{priv}}$ exactly using DP Set Union (Gopi et al., 2020) with the multiplicity assumption. The point of Thm. 2 is to give intuition about why PE works in practice; it is proved in App. F.2 under the same assumptions as in Thm. 1.

**Theorem 2.** *Let* $0 \leq \varepsilon \leq \log(1/2\delta)$. *Suppose each point in* $S_{\text{priv}}$ *has multiplicity* $B$. *Then, with high probability* $(\geq 1 - \tau)$, *Private Evolution (Alg. 1) with* $\sigma \gg \sqrt{T \log(1/\delta)}/\varepsilon$ *and* $H \gg \sigma\sqrt{\log(TLN_{\text{priv}}/\tau)}$, *when run for* $T$ *iterations, satisfies* $(\varepsilon, \delta)$*-DP and outputs* $S_{\text{syn}}$ *such that* $W_p(S_{\text{priv}}, S_{\text{syn}}) \leq \eta$, $\forall p \in [1, \infty]$, *whenever* $T$ *satisfies Eq. (2) and multiplicity* $B \gg H$.

Ignoring polylogarithmic factors in $d, L, N_{\text{priv}}, \log(D/\eta), \tau$, we need $T \gg d$ and $B \gg \sqrt{d \log(1/\delta)}/\varepsilon$ for Theorem 2 to hold. Thus, we should expect that PE will discover every cluster of private data of size $\gg \sqrt{d \log(1/\delta)}/\varepsilon$ in $O(d)$ iterations. We now compare this to previous work on DP clustering. Ghazi et al. (2020) gives an algorithm for densest ball, where they show an $(\varepsilon, \delta)$-DP algorithm which (approximately) finds any ball of radius $r$ which has at least $\gg \sqrt{d \log(1/\delta)}/\varepsilon$ private points. Thus intuitively, we see that PE compares favorably to SOTA DP clustering algorithms (though we don't have rigorous proof of this fact). If this can be formalized, then PE gives a very different algorithm for densest ball, which in turn can be used to solve DP clustering. Moreover PE is very amenable to parallel and distributed implementations. We therefore think this is an interesting theory problem for future work.

---

[8]If $\log L \gg d \log(D/\eta)$, i.e., if we generate an exponential number of points then by a simple epsilon-net argument we can prove that the algorithm will converge in a single step.

[9]Number of samples produced using VARIATION_API per iteration is $\leq L \cdot r \cdot N_{\text{priv}} = L \log(D/\eta)N_{\text{priv}}$.

[10]In fact, it is easy to formally prove that there will be a common approximate nearest neighbor.

**Why Private Evolution works well in practice.** We have seen that in the worst case, PE takes $\Omega(d)$ iterations to converge. In our experiments with CIFAR10 and Camelyon17, where $d = 2048$ is the embedding dimension, we see that PE actually converges in only about 20 iterations which is much smaller than $d$. We offer one plausible explanation for this via *intrinsic dimension*. Suppose the (embeddings of) realistic images lie on a low dimensional manifold $M$ inside $\mathbb{R}^d$ of dimension $d_{\text{intrinsic}} \ll d$ (see experimental results in App. G). Given an image $z$, VARIATION_API $(z)$ will create variations of $z$ which are also realistic images. Therefore the embeddings of these variations will also lie in the same manifold $M$, and PE is searching for the private points only inside the manifold $M$ without ever going outside it. Therefore the $d$ that matters for convergence is actually $d_{\text{intrinsic}} \ll d$. In this case, we expect that PE converges in $O(d_{\text{intrinsic}})$ iterations and discovers clusters of private points of size at least $\sqrt{d_{\text{intrinsic}} \log(1/\delta)}/\varepsilon$.

# F  PROOFS OF PE CONVERGENCE THEOREMS

We will slightly modify the algorithm as necessary to make it convenient for our analysis. We will make the following modeling assumptions:

- The private dataset $S_{\text{priv}}$ is contained in an $\ell_2$ ball of diameter $D$ and RANDOM_API will also produce initial samples in the same ball of diameter $D$. This is a reasonable assumption in practice, as images always have bounded pixel values: for the original images in UINT8 data type, each pixel is in the range of $[0, 255]$; in diffusion models, they are usually normalized to $[-1, 1]$ (i.e., 0 corresponds to -1 and 255 corresponds to 1), and all generated images are guaranteed to be in this range.

- The distance function used in Alg. 2 is just the $\ell_2$ norm, $d(x, z) = \|x - z\|_2$.

- The distribution of points we output is $S_{\text{syn}} = \mathcal{P}_T$ (i.e., we output a distribution of points).

- $S'_t \supset \text{supp}(\mathcal{P}_t) = \text{supp}(histogram_t)$.[11]

- $S_{t+1} = S'_t \cup \bigcup_{z \in S'_t}$ VARIATION_API $(z)$ where VARIATION_API $(z)$ samples $L$ samples each from Gaussian distributions $\mathcal{N}(z, \sigma_i^2 I)$ for $\sigma_i = \frac{D\sqrt{\log L}}{2^i d}$ where $1 \le i \le r$ and $r = \log(D/\eta)$ where $\eta > 0$ is the final Wasserstein distance.

## F.1  PROOF OF THM. 1

*Proof.* Fix a point $x \in S_{\text{priv}}$ and some iteration $t$. Suppose $z^* \in S_t$ is the closest point to $x$. Since $x$ will vote for $z^*$ in $histogram_t$, $z^* \in \text{supp}(\mathcal{P}_t) \subset S'_t$. Therefore VARIATION_API $(z^*) \subset S_{t+1}$. Let $V = $ VARIATION_API $(z^*)$.

**Claim 1.** *If $\|x - z^*\| \ge \eta$, then with probability at least $1/2$, some point in $V$ will get noticeably closer to $x$ than $z^*$, i.e.,*

$$\min_{z \in V} \|x - z\|_2 \le \left(1 - \frac{\log L}{4d}\right) \|x - z^*\|_2.$$

*Proof.* Let $s = \|x - z^*\|$ and let $\sigma \in \{\sigma_1, \sigma_2, \ldots, \sigma_r\}$ be such that $\sigma d/\sqrt{\log L} \in [s/2, s]$. Note that such a $\sigma$ exists since $s \in [\eta, D]$. We will now prove that one of the $L$ samples $z_1, z_2, \ldots, z_L \sim \mathcal{N}(z^*, \sigma^2 I_d)$ will get noticeably closer to $x$ than $z^*$. Let $z_i = z^* + \sigma w_i$ where $w_i \sim \mathcal{N}(0, I_d)$.

$$\min_{i \in [L]} \|x - z_i\|_2^2 = \|x - z^*\|_2^2 + \min_{i \in [L]} \left(\sigma^2 \|w_i\|_2^2 - 2\sigma \langle x - z^*, w_i \rangle\right)$$

$$\le s^2 + \max_{i \in [L]} \sigma^2 \|w_i\|_2^2 - \max_{i \in [L]} 2\sigma \langle x - z^*, w_i \rangle$$

Note that $\|w_i\|_2^2$ is a $\chi_d^2$ random variable. By using upper tail bounds for $\chi_d^2$ distribution and union bound over $i \in [L]$, we can bound

$$\Pr\left[\max_{i \in [L]} \|w_i\|_2^2 \ge 3d/2\right] \le L \exp(-\Omega(d)) \ll 1.$$

---

[11]This may not be true in the original algorithm due to sampling, we need to modify it so that $S'_t \supset \text{supp}(\mathcal{P}_t)$.

The distribution of $\langle x - z^*, w_i \rangle$ is the same as $\|x - z^*\|_2 \tilde{w}_i$ where $\tilde{w}_1, \ldots, \tilde{w}_L$ are i.i.d. $\mathcal{N}(0,1)$ random variables. By using the fact that max of $L$ i.i.d. Gaussians is at least $\sqrt{\log L}$ with probability at least $3/4$ (for $L \gg 1$), we get that

$$\Pr\left[\max_{i \in [L]} \langle x - z^*, w_i \rangle \leq s\sqrt{\log L}\right] \leq \frac{1}{4}.$$

Combining everything we get:

$$\frac{1}{2} \geq \Pr\left[\min_{i \in [L]} \|x - z_i\|_2^2 \geq s^2 + (3/2)d\sigma^2 - 2s\sigma\sqrt{\log L}\right]$$

$$\geq \Pr\left[\min_{i \in [L]} \|x - z_i\|_2^2 \geq \max_{\lambda \in [1/2,1]} s^2 + (3/2)d\left(\frac{\lambda s\sqrt{\log L}}{d}\right)^2 - 2s\left(\frac{\lambda s\sqrt{\log L}}{d}\right)\sqrt{\log L}\right]$$

$$\geq \Pr\left[\min_{i \in [L]} \|x - z_i\|_2^2 \geq s^2 + \left(\frac{s^2 \log L}{d}\right)\max_{\lambda \in [1/2,1]}(3\lambda^2/2 - 2\lambda)\right]$$

$$\geq \Pr\left[\min_{i \in [L]} \|x - z_i\|_2^2 \geq s^2\left(1 - \frac{\log L}{2d}\right)\right]$$

$$\geq \Pr\left[\min_{i \in [L]} \|x - z_i\|_2 \geq s\left(1 - \frac{\log L}{4d}\right)\right]$$

where last inequality uses the fact that $\sqrt{1-t} \leq 1 - \frac{t}{2}$ for $t \leq 1$. $\qquad\square$

Now in $T$ iterations, $\min_{z \in S_t} \|x - z\|_2$ will shrink by a factor of $\left(1 - \frac{\log L}{4d}\right)$ in at least $T/4$ iterations with probability $1 - \exp(-\Omega(T)) \geq 1 - \frac{\tau}{N_{\text{priv}}}$ (by standard Chernoff bounds). Note that in iterations where it doesn't shrink, it doesn't grow either since $S_t' \subset S_{t+1}$. Similarly, if $\min_{z \in S_t} \|x - z\|_2 \leq \eta$ for some iteration, it will remain so in all subsequent iterations. Therefore after $T \gg \frac{d \log(\tilde{D}/\eta)}{\log L}$ iterations, $\min_{z \in S_T} \|x - z\|_2 \leq \eta$ with probability at least $1 - \frac{\tau}{N_{\text{priv}}}$. By union bounding over all points we get that, with probability at least $1 - \tau$, for every point $x \in S_{\text{priv}}$ there is a point in $S_T$ which is $\eta$-close. This proves that $W_p(S_{\text{priv}}, \mathcal{P}_T) \leq \eta$. $\qquad\square$

### F.2 Proof of Thm. 2

*Proof.* Since we are doing $T$ iterations of Gaussian mechanism with noise level $\sigma$, we need to set $\sigma \gg \sqrt{T \log(1/\delta)}/\varepsilon$ to satisfy $(\varepsilon, \delta)$-DP (Dwork et al., 2014) when $\varepsilon \leq log(1/2\delta)$. Let $x \in S_{\text{priv}}$ be a point with multiplicity $B$. If $z^* \in S_t$ is the closest point to $x$, then it will get $B$ votes. After adding $\mathcal{N}(0, \sigma^2)$ noise, if $B \gg H \gg \sigma\sqrt{\log(TLN_{\text{priv}}/\tau)}$, then with probability at least $1 - \tau/(4T)$, the noisy votes that $z^*$ gets is still above the threshold $H$. Therefore $z^*$ will survive in $S_{t+1}$ as well. Also since $H \gg \sigma\sqrt{\log(TLN_{\text{priv}}/\tau)}$, with probability $1 - \tau/(4T)$, points in $S_t$ which do not get any votes (there are $LN_{\text{priv}}$ of them) will not survive even after adding noise and thresholding by $H$. Therefore, by union bounding over all $T$ iterations, with probability at least $1 - \tau/2$, the algorithm behaves identically to the non-private algorithm. Therefore by an identical proof as in the non-private analysis, we can prove that after $T$ iterations $W_p(S_{\text{priv}}, \mathcal{P}_T) \leq \eta$ with probability at least $1 - \tau$. $\qquad\square$

## G Intrinsic Dimension of Image Embeddings

To illustrate the intrinsic dimension of image embeddings, we use the following process:

1. We (randomly) take an image $x$ from CIFAR10.
2. We use VARIATION_API from App. J to obtain 3000 image variations of $x$: $x_1, \ldots, x_{3000}$, and their corresponding inception embeddings $g_1, \ldots, g_{3000} \in \mathbb{R}^{2048}$. 3000 is chosen so that the number of variations is larger than the embedding dimension.
3. We construct a matrix $M = [g_1 - g; \ldots; g_{3000} - g] \in \mathbb{R}^{3000 \times 2048}$, where $g$ is the mean$(g_1, \ldots, g_{3000})$.

4. We compute the singular values of $M$: $\sigma_1 \geq \sigma_2 \geq \ldots \geq \sigma_{2048}$.

5. We compute the minimum number of singular values $n$ needed so that the explained variance ratio[12] $\sum_{i=1}^{n} \sigma_i^2 / \sum_{i=1}^{2048} \sigma^2 \geq 0.8$. Intuitively, this $n$ describes how many dimensions are needed to reconstruct the embedding changes $M$ with a small error. We use it as an estimated intrinsic dimension of the image variations.

We conduct the above process with the variation degree $[98, 96, 94, 92, 90, 88, 86, 84, 82, 80, 78, 76, 74, 72, 70, 68, 66, 64, 62, 60]$ utilized in the CIFAR10 experiments (see App. J). We additionally add a variation degree of 100 which is the highest variation degree in the API that was used to generate the initial samples. We plot the estimated intrinsic dimension v.s. variation degree in Fig. 9. The raw original singular values of $M/\sqrt{3000}$ for variation degree=60 are in Fig. 10 (other variation degrees have similar trend). Two key observations are:

- As the variation degree increases, the estimated intrinsic dimension also increases. This could be because the manifold of image embeddings is likely to be non-linear, the above estimation of intrinsic dimension is only accurate when we perturb the image $x$ to a small degree so that the changes in the manifold can still be well approximated by a linear subspace. Using a larger variation degree (and thus larger changes in the embedding space) will overestimate the intrinsic dimension.

- Nevertheless, we always see that the singular values decrease rapidly (Fig. 10) and the estimated intrinsic dimension is much smaller than the embedding size 2048 (Fig. 9), which supports our hypothesis in App. E.

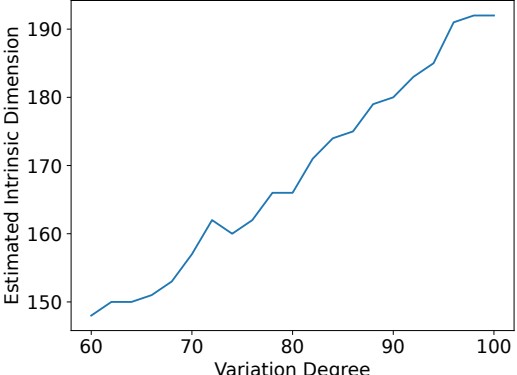

Figure 9: Estimated intrinsic dimension of inception embeddings of realistic images.

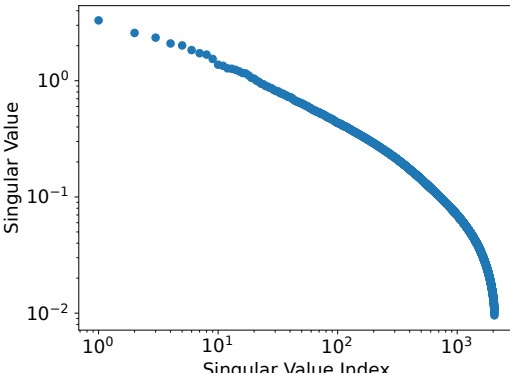

Figure 10: Singular values of inception embeddings of image variations at variation degree=60.

---

[12]See https://scikit-learn.org/stable/modules/generated/sklearn.decomposition.TruncatedSVD.html.

# H  RELATION OF DPWA TO PRIOR WORK

Recall that in DPWA, we want a DP algorithm to output $S_{\text{syn}}$ which is close to the distribution of $S_{\text{priv}}$ in Wasserstein distance w.r.t. some distance function $d(\cdot, \cdot)$. When the distance function $d(\cdot, \cdot)$ is just $\ell_2$ distance between the samples (i.e., $\ell_2$ in the pixel space for images), then DPWA is closely related to DP Clustering (Ghazi et al., 2020; Balcan et al., 2017; Su et al., 2016) and DP Heatmaps (Ghazi et al., 2022).

In (Ghazi et al., 2022), to give an algorithm for DP Heatmaps, the authors study DP sparse EMD[13] aggregation problem where we need to output a distribution of points which approximates the distribution of private data in EMD distance (i.e., $W_1$). They study this problem only in two dimensions and the running time of their algorithms (suitably generalized to higher dimensions) will be exponential in the dimension $d$.

The DP Clustering problem requires us to output a clustering of private data using DP. The most common clustering studied is $k$-means clustering where we should output $k$ cluster centers such that $k$-means cost is minimized, where $k$-means cost is the sum of squares of $\ell_2$-distance of each data point to its nearest cluster center. Note that in DPWA, if the number of synthetic data points $N_{\text{syn}}$ is specified to be $k$, then DP $k$-means clustering and DPWA with $W_2$ metric are equivalent. In (Ghazi et al., 2022), a polynomial time DP Clustering algorithm with an additional $k$-means cost (over what is non-privately possible) of $k\sqrt{d \log(1/\delta)}\text{polylog}(N_{\text{priv}}, d)/\epsilon$ is given. This can be converted into an upper bound on the Wasserstein distance. But this is not a practical algorithm. The privacy-utility tradeoffs are bad due to the large hidden constants in the analysis and the authors don't provide an implementation. There is a practical DP Clustering algorithm (along with an implementation) given in (Chang & Kamath, 2023) (but with no theoretical guarantees).

## H.1  WHY NOT JUST USE DP CLUSTERING?

We now explain why we can't just use prior work on DP Clustering to solve DPSDA say for images.

**Clustering in the image space.** We can use DP $k$-means Clustering to cluster the images w.r.t. $\ell_2$ metric in the pixel space. This doesn't work because $\ell_2$ distance in the pixel space doesn't capture semantic similarity. An image which is slightly shifted in pixel space gets very far in $\ell_2$ distance. And the dimension of the images is too large for prior DP Clustering algorithms to work well. Their convergence and privacy-utility tradeoffs depend too strongly on the dimension.

**Clustering in the embedding space.** We can use DP $k$-means Clustering to cluster the image embeddings w.r.t. $\ell_2$ metric in the embedding space. Note that this is the distance function we use in PE (Eq. (1)). Even after we find the cluster centers, it is hard to invert the embedding map (i.e., find an image whose embedding is close to a given vector in the embedding space).[14] Moreover the dimension of the embedding space is still too large for the above methods to be practical.

Our PE algorithm does much better because:

1. Its distance function is $\ell_2$ in the embedding space which captures semantic similarity,

2. It exploits the intrinsic dimension of the manifold of images in the embedding space which is much smaller than the embedding dimension (see App. E and App. G) and

3. There is no need to invert points in embedding space to the image space.

In an early experiment, we have tried DP clustering in the CLIP embedding space using the practical DP Clustering algorithm in (Chang & Kamath, 2023). We then inverted the cluster centers (which are in the embedding space) using unCLIP. But we found the resulting images are too noisy compared to the images we get from PE and the FID scores are also significantly worse than that of PE.

---

[13]Earth's Mover Distance, which is the another name for Wasserstein metric $W_1$.

[14]Some special embeddings such as CLIP embedding do have such an inverse map called unCLIP (Ramesh et al., 2022).

# I  IMPLEMENTATION DETAILS ON LABEL CONDITION

There are two meanings of "conditioning" that appear in our work:

1. Whether the pre-trained networks or APIs (e.g., ImageNet pre-trained diffusion models used in § 5.1.1 and 5.1.2) support conditional input (e.g., ImageNet class label).
2. Whether the generated samples are associated with class labels from the private data.

In DP fine-tuning approaches, these two usually refer to the same thing: if we want to generate class labels for generated samples, the common practice is to use a pre-trained network that supports conditional input (Ghalebikesabi et al., 2023). However, in PE, these two are completely orthogonal.

**Conditional pre-trained networks/APIs.** We first explain our implementation when the pre-trained networks or APIs support conditional inputs such as class labels or text prompts. When generating the initial population using RANDOM_API (Line 1), we will either randomly draw labels from all possible labels when no prior public information is available (which is what we do in CIFAR10 and Camelyon17 experiments where we randomly draw from all possible ImageNet classes), or use the public information as condition input (e.g., the text prompt used in Stable Diffusion experiments; see App. L). In the subsequent VARIATION_API calls (Line 6), for each image, we will use its associated class label or text prompt as the condition information to the API, and the output samples from VARIATION_API will be associated with the same class label or text prompt as the input sample. For example, if we use an image with "peacock" class to generate variations, all output images will be associated with "peacock" class for future VARIATION_API calls. Note that throughout the above process, all the condition inputs to the pre-trained networks/APIs are public information; they have nothing to do with the private classes.

**Conditional generation.** Conditional generation is achieved by Alg. 3, where we separate the samples according to their class labels, and run the main algorithm (Alg. 1) on each sample set. We can use either conditional or unconditional pre-trained networks/APIs to implement it.

Throughout the paper, "(un)condition" refers to 2, expect the caption in Fig. 7 which refers to 1.

# J  MORE DETAILS AND RESULTS ON CIFAR10 EXPERIMENTS

**Pre-trained model.** By default, we use the checkpoint `imagenet64_cond_270M_250K.pt` released in (Nichol & Dhariwal, 2021).[15] For the ablation study of the pre-trained network, we additionally use the checkpoint `imagenet64_uncond_100M_1500K.pt`.

**API implementation.** RANDOM_API follows the standard diffusion model sampling process. VARIATION_API is implemented with SDEdit (Meng et al., 2021), which adds noise to input images and lets the diffusion model denoise them. We use DDIM sampler (Song et al., 2020) and the default noise schedule to draw samples. Note that these choices are not optimal; our results can potentially be improved by using better noise schedules and the full DDPM sampling (Ho et al., 2020) which are known to work better. The implementation of the above APIs is straightforward without touching the core modeling part of diffusion models and is similar to the standard API implementations in Stable Diffusion (App. L).

**Hyperparameters.** We set number of iterations $T = 20$, lookahead degree $k = 8$, and number of generated samples $N_{\mathrm{syn}} = 50000$. For RANDOM_API and VARIATION_API, we use DDIM sampler with 100 steps. For VARIATION_API, we use SDEEdit (Meng et al., 2021) by adding noise till $[98, 96, 94, 92, 90, 88, 86, 84, 82, 80, 78, 76, 74, 72, 70, 68, 66, 64, 62, 60]$ timesteps for each iteration respectively. These timesteps can be regarded as the $v$ parameter in § 3.3.

For the experiments in Fig. 4, we use noise multiplier $\sigma = t \cdot \sqrt{2}$ and threshold $H = 2t$ for $t \in \{5, 10, 20\}$, and pick the the pareto frontier. Fig. 11 shows all the data points we got. Combining this figure with Fig. 4, we can see that PE is not very sensitive to these hyper-parameters, and even with less optimal choices PE still outperforms the baselines.

For the experiments in Fig. 5, we use noise multiplier $\sigma = \sqrt{2}$ and threshold $H = 2$ (i.e., $t = 1$).

---

[15] https://github.com/openai/improved-diffusion

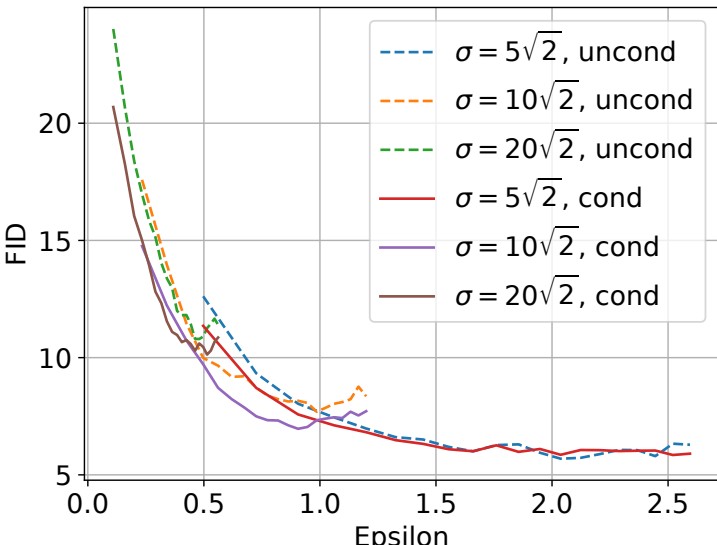

Figure 11: FID (lower is better) v.s. privacy cost $\epsilon$ ($\delta = 10^{-5}$) on CIFAR10 with different noise multipliers and thresholds. "(Un)cond" means (un)conditional generation.

For the experiments in Figs. 3 and 13, we use noise multiplier $\sigma = 10\sqrt{2}$ and threshold $H = 20$ (i.e., $t = 10$) and the number of PE iteration is 5 (i.e., the point of "Ours (cond)" in Fig. 4 that has FID$\leq 7.9$).

For downstream classification (Fig. 5), we follow (Ghalebikesabi et al., 2023) to use WRN-40-4 classifier (Zagoruyko & Komodakis, 2016). We use the official repo[16] without changing any hyperparameter except adding color jitter augmentation according to (Ghalebikesabi et al., 2023). The ensemble of the classifier is implemented by ensembling the logits.

**FID evaluation.** Compared to Fig. 4 in the main text, Fig. 12 shows the full results of two versions of (Harder et al., 2023). Baseline results are taken from (Harder et al., 2023; Ghalebikesabi et al., 2023).

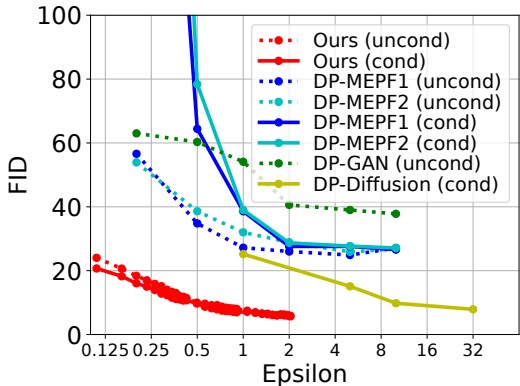

Figure 12: FID (Heusel et al., 2017) (lower is better) v.s. privacy cost $\epsilon$ on CIFAR10 ($\delta = 10^{-5}$). Baseline results are taken from (Harder et al., 2023; Ghalebikesabi et al., 2023). (Un)cond means (un)conditional generation. Ours achieves the best privacy-quality trade-off.

**Classification accuracy.** In § 5.1.1, we show the classification accuracy at a single privacy budget $\epsilon = 3.34$. In Table 1, we further show how the classification accuracy evolves with respect to different $\epsilon$s. These results are from the first 5 PE iterations.

---

[16] https://github.com/szagoruyko/wide-residual-networks/tree/master/pytorch

| $\epsilon$ | Accuracy |
|---|---|
| 1.36 | 72.46% |
| 1.99 | 78.78% |
| 2.50 | 80.83% |
| 2.94 | 81.15% |
| 3.34 | 81.74% |

Table 1: Classification accuracy v.s. privacy cost $\epsilon$ on CIFAR10 ($\delta = 10^{-5}$).

**Generated samples.** See Figs. 13 and 14 for generated images and real images side-by-side. Note that the pre-trained model we use generates 64x64 images, whereas CIFAR10 is 32x32. In Fig. 3, we show the raw generated 64x64 images; in Fig. 13, we scale them down to 32x32 for better comparison with the real images.

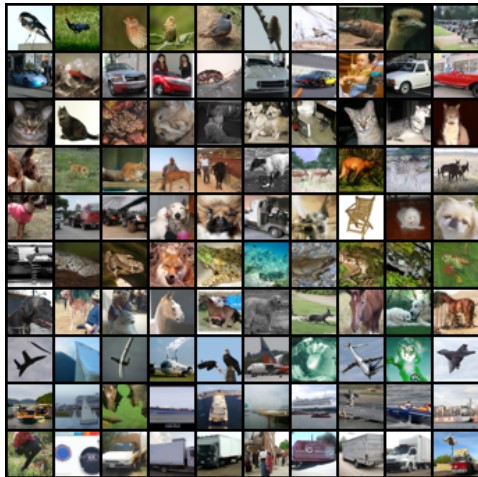

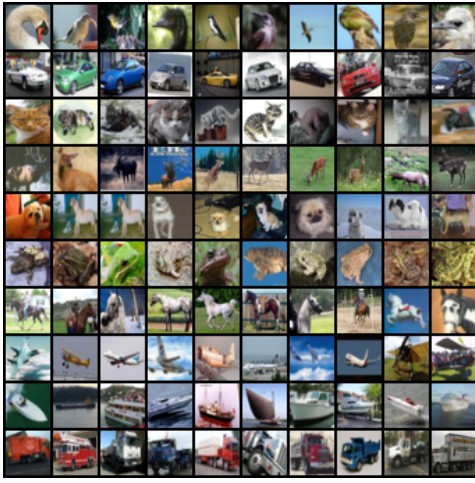

Figure 13: Generated samples on CIFAR10 with $(0.67, 10^{-5})$-DP. Each row corresponds to one class. FID=7.87.

Figure 14: Real samples from CIFAR10. Each row corresponds to one class.

**Nearest samples in the private dataset** Figs. 15 and 16 show generated images and their nearest neighbors in the private dataset evaluated using two distance metrics: $\ell_2$ distance in the inception embedding space and the pixel space. We can see that the generated images are different from private images. This is expected due to the DP guarantee.

**Distributions of the distances to nearest samples.** Continuing the above experiments, we further show the distribution of the distances between (1) generated samples and their nearest real samples and (2) real samples and their nearest generated samples in Figs. 17 and 18 respectively. Two key observations are: (1) During the early PE iterations, the distances tend to decrease. This means that PE is effective in pushing the generated distribution to be closer to the private distribution. (2) However, as PE continues, the distances stop decreasing. It is expected, as DP upper bounds the probability of reconstructing any sample in the private dataset.

**Samples with the highest and the lowest votes.** Fig. 19 shows the samples with the highest and the lowest votes in DP Nearest Neighbors Histogram across different PE iterations. We can see that DP Nearest Neighbors Histogram picks samples similar to the private data as desired. This is more obvious in the first two PE iterations, where DP Nearest Neighbors Histogram assigns high votes on the samples with correct classes and puts low votes on the samples with incorrect classes.

**Distribution of counts in DP Nearest Neighbors Histogram.** Fig. 20 shows that the standard deviation of counts in DP Nearest Neighbors Histogram tends to decrease with more PE iterations on CIFAR10. This means that the histogram becomes "more uniform" with more PE iterations. It is expected for the following reasons. In the beginning, only a few generated samples are close to the private data. Those generated samples get most of the votes from private samples, which results in a concentrated histogram. PE will then pick those highly voted generated samples and do more

variations over them. As a result, private samples that voted for the same generated sample may now find different closest generated samples and distribute the votes, which results in a "more uniform" histogram.

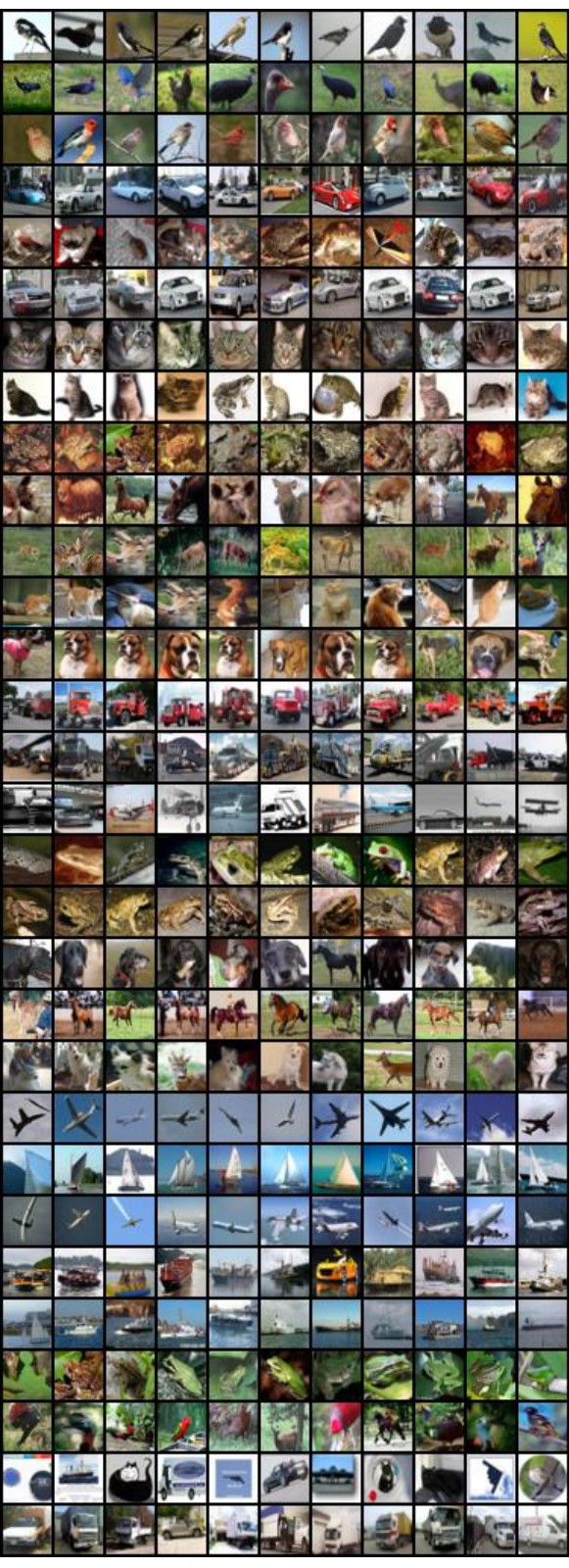

Figure 15: Nearest samples in the private dataset on CIFAR10. In each row, the first column is a generated image (from Fig. 13), and the other columns are its nearest neighbors in the private dataset, sorted by the distance in ascending order. Every three rows correspond to generated image from one class. The distance metric is $\ell_2$ in the **inception embedding space.**

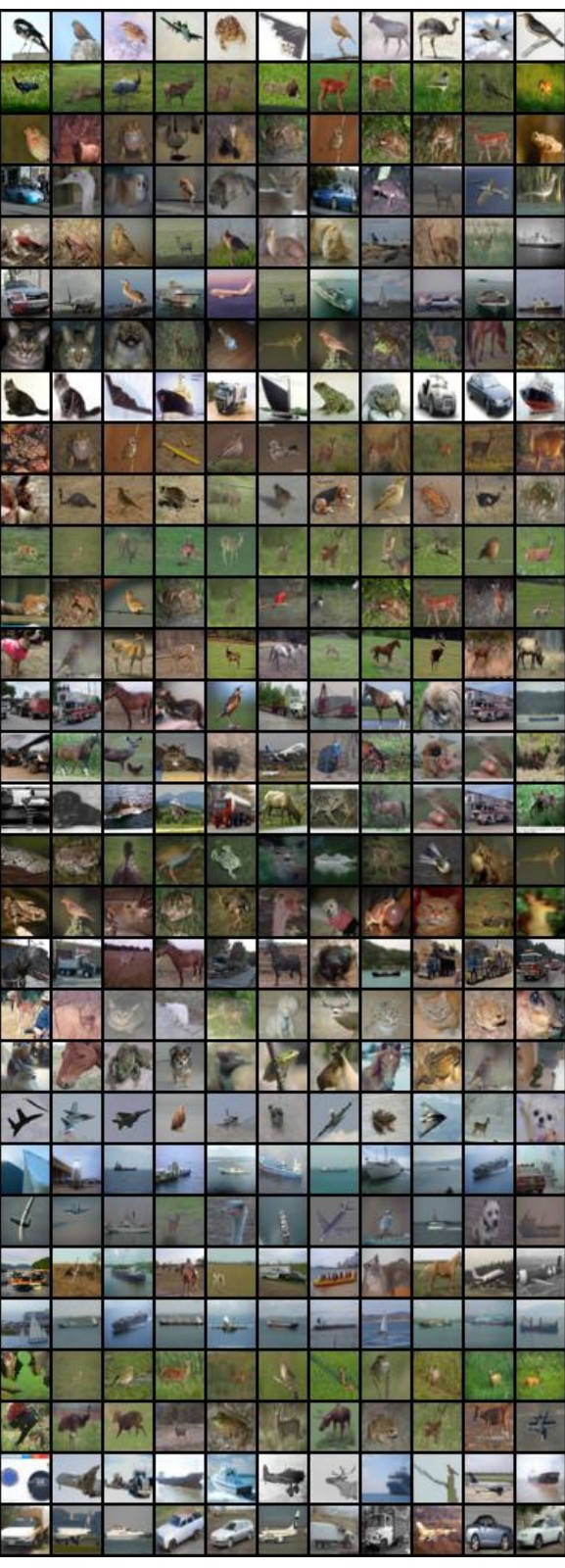

Figure 16: Nearest samples in the private dataset on CIFAR10. In each row, the first column is a generated image (from Fig. 13), and the other columns are its nearest neighbors in the private dataset, sorted by the distance in ascending order. Every three rows correspond to generated image from one class. The distance metric is $\ell_2$ in the **pixel space.**

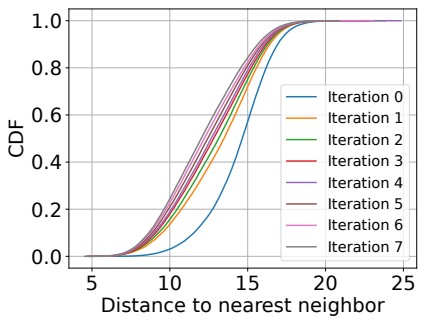 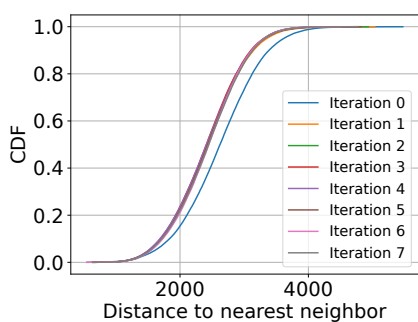

(a) The distance metric is $\ell_2$ in the **inception embedding space.**

(b) The distance metric is $\ell_2$ in the **pixel space.**

Figure 17: CDF of the distributions **between each generated sample and its nearest private samples** on CIFAR10 across different PE iterations. "Iteration 0" refers to the initial random samples from Line 1 in Alg. 1.

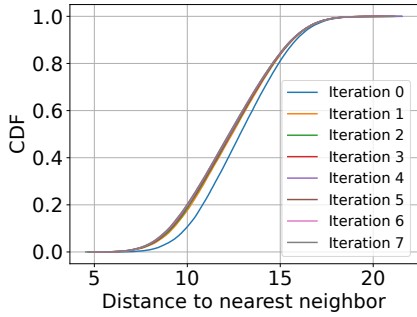 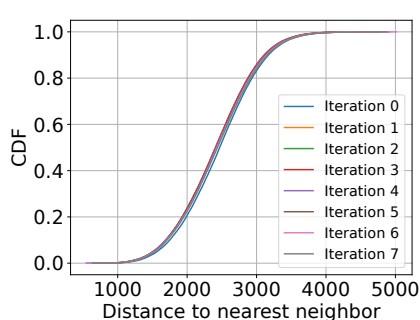

(a) The distance metric is $\ell_2$ in the **inception embedding space.**

(b) The distance metric is $\ell_2$ in the **pixel space.**

Figure 18: CDF of the distributions **between each private sample and its nearest generated samples** on CIFAR10 across different PE iterations. "Iteration 0" refers to the initial random samples from Line 1 in Alg. 1.

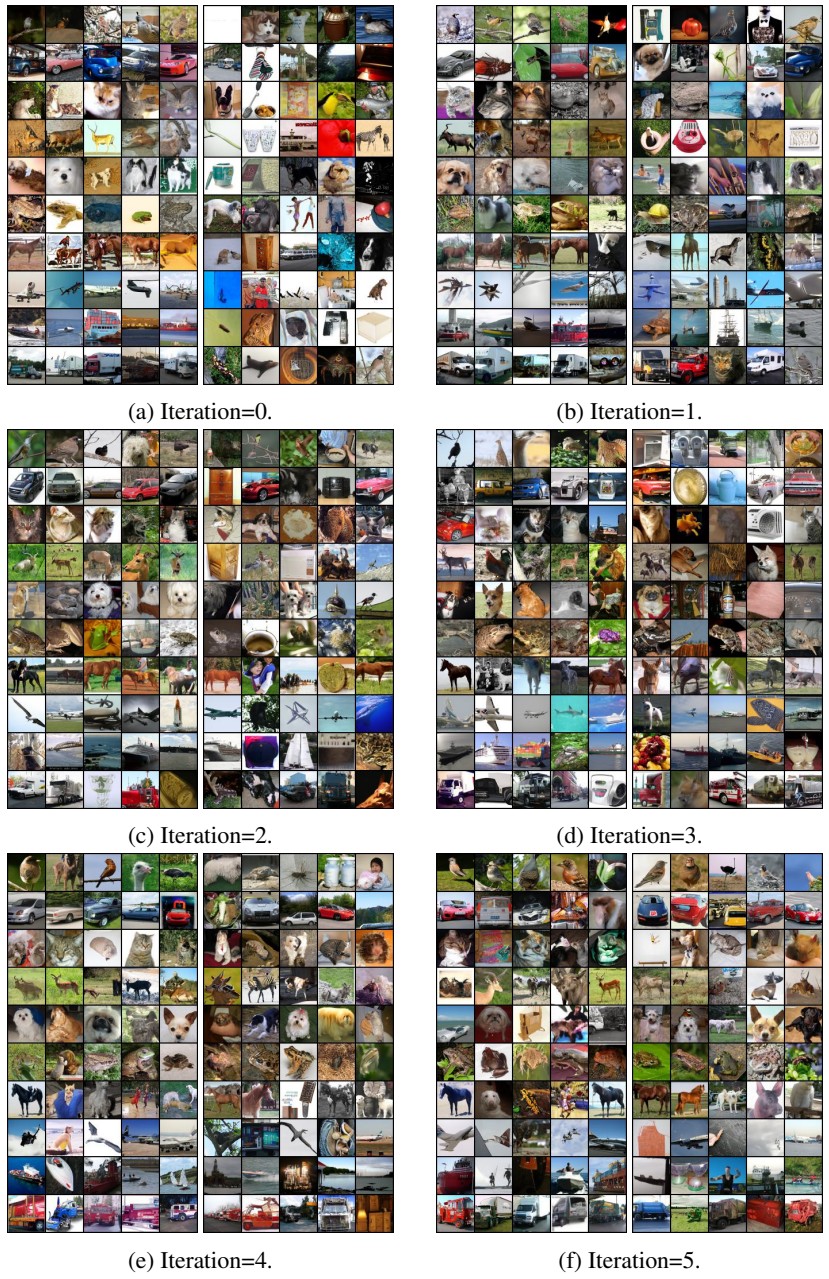

(a) Iteration=0.

(b) Iteration=1.

(c) Iteration=2.

(d) Iteration=3.

(e) Iteration=4.

(f) Iteration=5.

Figure 19: Samples with the highest and the lowest counts in the DP Nearest Neighbors Histogram on CIFAR10. In each subfigure, each row corresponds to one class; the left subfigure shows the samples with the highest counts and the right subfigure shows the samples with the lowest counts.

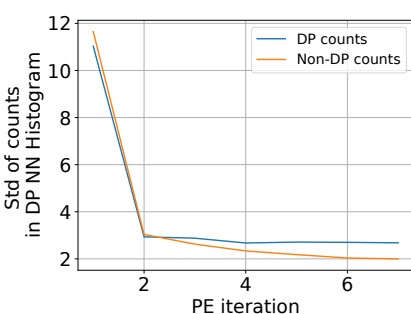

Figure 20: The standard deviation of counts in DP Nearest Neighbors Histogram across different PE iterations on CIFAR10. "DP counts" refers to the counts after adding Gaussian noise and thresholding. "Non-DP counts" refer to the counts before adding Gaussian noise and thresholding.

# K    MORE DETAILS AND RESUTLS ON CAMELYON17 EXPERIMENTS

**Pre-trained model.** We use the checkpoint `imagenet64_cond_270M_250K.pt` released in (Nichol & Dhariwal, 2021).[17]

**API implementation.** Same as App. J.

**Hyperparameters.** We set lookahead degree $k = 8$, and number of generated samples $N_{\text{syn}} = 302436$.

About the experiments in Fig. 21. For RANDOM_API and VARIATION_API, we use DDIM sampler with 10 steps. For VARIATION_API, we take a 2-stage approach. the first stage, we use DDIM sampler with 10 steps and use SDEEdit (Meng et al., 2021) by adding noise till $[10, 10, 10, 10, 9, 9, 9, 9, 9, 8, 8, 8, 8, 8, 7, 7, 7, 7]$ timesteps for each iteration respectively. In the second stage, we use DDIM sampler with 40 steps and use SDEEdit (Meng et al., 2021) by adding noise till $[20, 19, 18, 17, 16, 15, 14, 13, 12, 11, 10, 9, 8, 7, 6, 5]$ timesteps for each iteration respectively. These timesteps can be regarded as the $v$ parameter in § 3.3. We use noise multiplier $\sigma = 2 \cdot \sqrt{2}$ and threshold $H = 4$.

About the experiments in § 5.1.2. For RANDOM_API and VARIATION_API, we use DDIM sampler with 10 steps. For VARIATION_API, we use DDIM sampler with 10 steps and use SDEEdit (Meng et al., 2021) by adding noise till $[10, 10, 10, 10, 9, 9, 9]$ timesteps for each iteration respectively. These timesteps can be regarded as the $v$ parameter in § 3.3. We use noise multiplier $\sigma = 1.381$ and threshold $H = 4$.

**Generated samples.** See Figs. 21 and 22 for generated images and real images side-by-side. Note that the real images in Camelyon17 dataset are 96x96 images, whereas the pre-trained network is 64x64. In Fig. 22, we scale them down to 64x64 for better comparison.

Fig. 23 shows the generated images in the intermediate iterations. We can see that the generated images are effectively guided towards Camelyon17 though it is very different from the pre-training dataset.

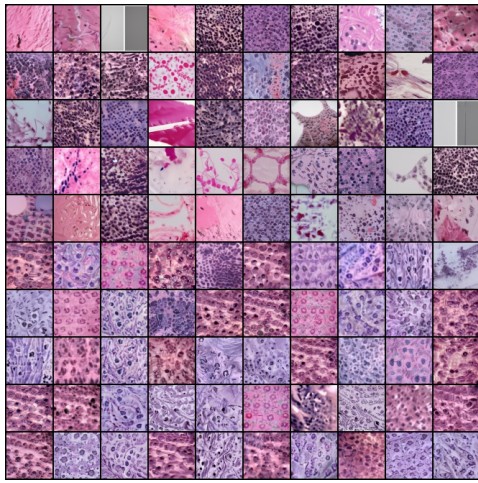

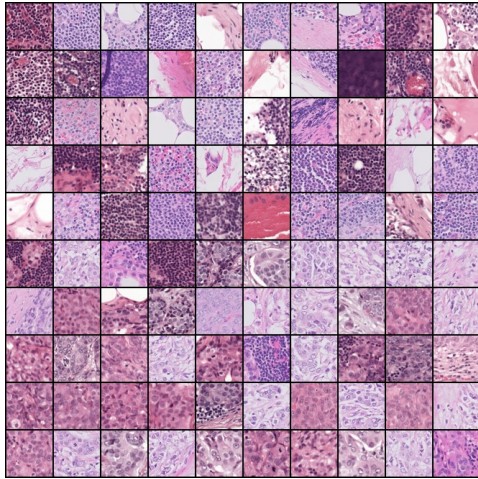

Figure 21: Generated samples on Camelyon17 with $(9.92, 3 \cdot 10^{-6})$-DP. The first five rows correspond to one class, and the rest correspond to the other class. FID=10.66.

Figure 22: Real samples from Camelyon17. The first five rows correspond to one class, and the rest correspond to the other class.

**Nearest samples in the private dataset** Figs. 24 and 25 show generated images and their nearest neighbors in the private dataset evaluated using two distance metrics: $\ell_2$ distance in the inception embedding space and the pixel space. Similar to the results in CIFAR10, we can see that the generated images are different from private images. This is expected due to the DP guarantee.

---

[17] https://github.com/openai/improved-diffusion

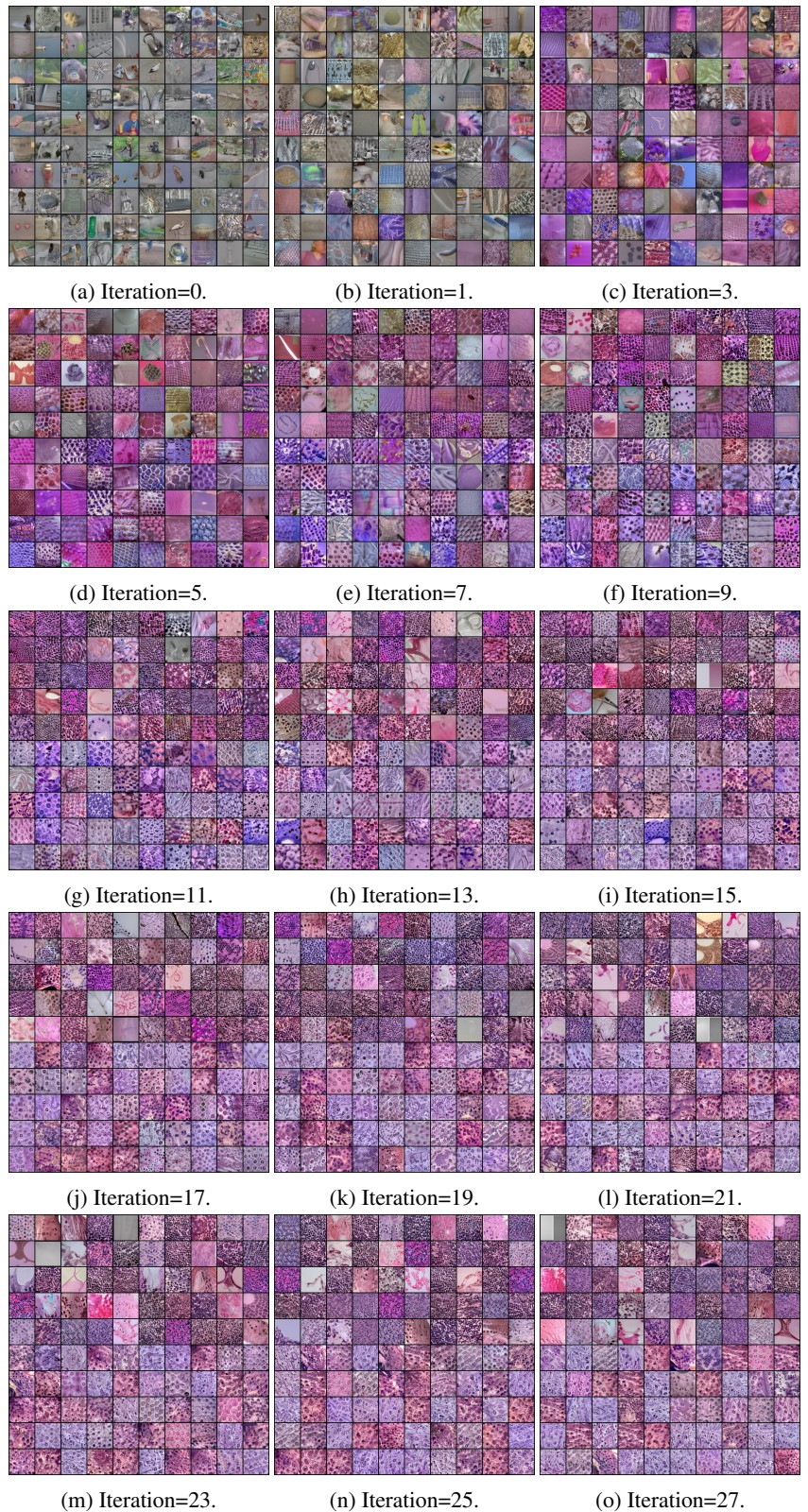

(a) Iteration=0.  (b) Iteration=1.  (c) Iteration=3.

(d) Iteration=5.  (e) Iteration=7.  (f) Iteration=9.

(g) Iteration=11.  (h) Iteration=13.  (i) Iteration=15.

(j) Iteration=17.  (k) Iteration=19.  (l) Iteration=21.

(m) Iteration=23.  (n) Iteration=25.  (o) Iteration=27.

Figure 23: Generated samples on Camelyon17 at the first few iterations. We can see that the generated images are gradually guided from ImageNet, the pre-training dataset, to Camelyon17, the (very different) private dataset. "Iteration=0" means the initial random samples from RANDOM_API.

**Samples with the highest and the lowest votes.** Fig. 26 shows the samples with the highest and the lowest votes in DP Nearest Neighbors Histogram across different PE iterations. We can see that DP Nearest Neighbors Histogram gradually picks samples with the right patterns as the private data, and drops samples with more different patterns. As Camelyon17 is further away from the pre-training dataset ImageNet than CIFAR10, we can see that it converges slower than the case in CIFAR10 (Fig. 19).

**Distributions of the distances to nearest samples.** Continuing the above experiments, we further show the distribution of the distances between (1) generated samples and their nearest real samples and (2) real samples and their nearest generated samples in Figs. 27 and 28 respectively. Similar to the CIFAR10 experiments (Figs. 17 and 18), we see that: (1) During the early PE iterations, the *inception* distances tend to decrease. This means that PE is effective in pushing the generated distribution to be closer to the private distribution. (2) However, as PE continues, the *inception* distances stop decreasing. It is expected, as DP upper bounds the probability of reconstructing any sample in the private dataset. However, one difference to CIFAR10 results is that the distance between each generated sample and its nearest private samples measured in the *original pixel space* (Fig. 27b) tends to increase. We hypothesize that it is be due to nature of this dataset that any shifts of the histological images are still in-distribution but can result in a high distance in the *original pixel space*.

**Distribution of counts in DP Nearest Neighbors Histogram.** Fig. 29 shows that the standard deviation of counts in DP Nearest Neighbors Histogram tends to decrease with more PE iterations on Camelyon17. This accords with the observations and takeaways messages in CIFAR10 (Fig. 20).

**Inspescting the ImageNet class labels of the generated samples.** As discussed in App. I, each generated samples have an associated ImageNet label. Here, we inspect those labels to understand how PE works in this dataset. The list of the labels and their associated number of generated images are: "honeycomb" (164647), "velvet" (83999), "nematode, nematode worm, roundworm" (35045), "handkerchief, hankie, hanky, hankey" (14495), "bib" (3102), "head cabbage" (934), "bubble" (142), "stole" (72). We see that 54.4% images are with the label "honeycomb". Many honeycomb images in ImageNet have a "net-like" structure, which shares some similarities with Camelyon17. However, the details, colors, and structures are still different. We compute the FID between honeycomb images and CIFAR10. The FID score is 162.86, which is much higher than the FID score of the final generated images we get (10.66 in Fig. 21). These results show that the existence of the honeycomb class in ImageNet helps with PE, but PE does more than just picking this class.

**Why PE works under large distribution shifts.** The fact that PE is able to make ImageNet-pre-trained-models to generate Camelyon17 images may appear surprising to readers. Here, we provide intuitive explanations. Even though the diffusion model is trained on natural images, the support of the generated distribution could be very large due to the formulation. More concretely, let's look at two examples: score-based models (Song & Ermon, 2019), which is closely related to diffusion models, and the diffusion model we used (Ho et al., 2020). For score-based models, its modeling target is a distribution perturbed by Gaussian noise, and therefore, the generated distribution spans the entire sample space. For diffusion models (Ho et al., 2020), the latent space has the same size as the image, and the last denoising step is modeled as a distribution derived from a Gaussian distribution that spans the entire pixel space (see Section 3.3 of Ho et al. (2020)). Therefore, the generated distribution of diffusion models also spans the entire sample space.

In other words, for any (trained) score-based models or diffusion models, it is theoretically possible to generate images similar to the private dataset (or in fact, generate any images). The problem is that the probability of generating such images is small if there is a large distribution shift from the pre-training data to the private data. PE is effective in guiding the diffusion model to generate samples from the region that is low-density in the original pre-trained distribution but high-density in the private distribution.

To make it more concrete, we provide how the generated images evolve from natural images to Camelyon17 dataset in Fig. 23 and the selected/filtered samples by PE in Fig. 26. At every iteration, PE selects the set of images that are most similar to Camelyon17 dataset (Fig. 26). Those images might still appear different from Camelyon17 in early iterations. However, as long as we get images that are more similar to Camelyon17 through VARIATION_API at every iteration, we will make progress, and finally, we can get images similar to Camelyon17 (Fig. 21).

We further experiment PE under different levels of distribution shifts. To do that, we take the Camelyon17 dataset and modify the *saturation* of the images to create a sequence of datasets, each with a different saturation change. This way, we create a sequence of datasets with different levels of distribution shifts from ImageNet. From Fig. 30, we can see that no matter what the level of distribution shifts is, PE can consistently improve the generated distribution towards the private data.

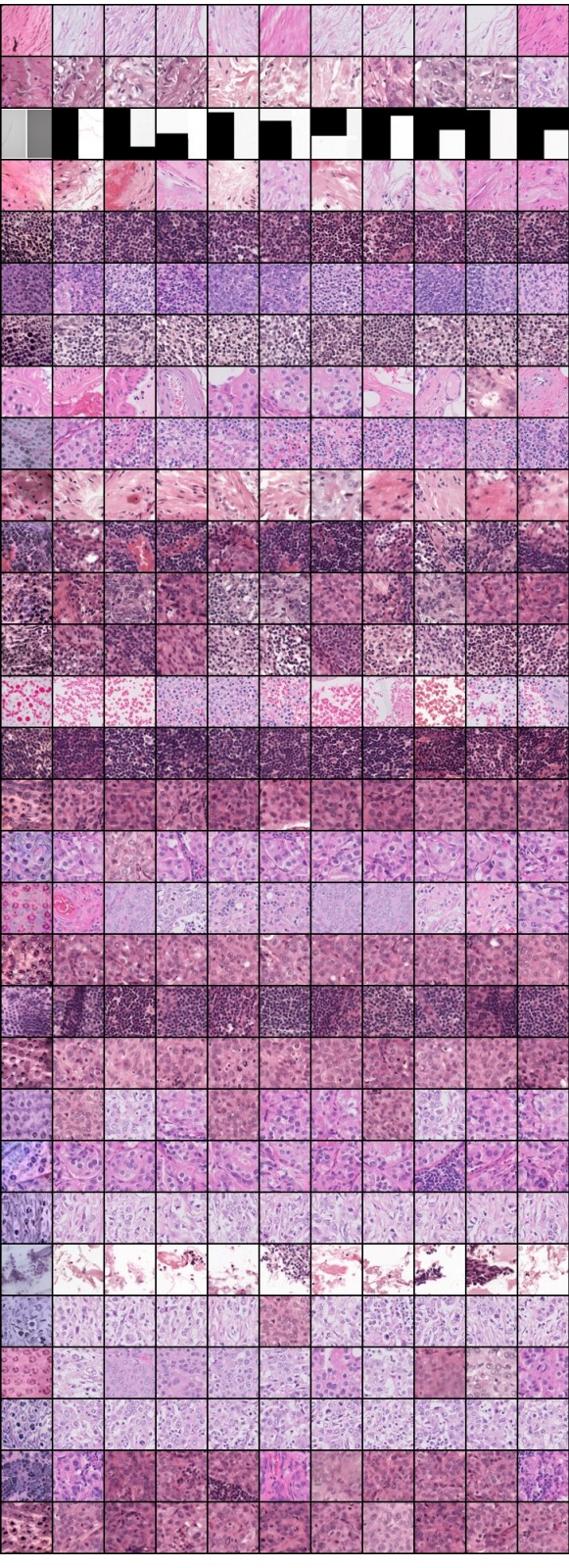

Figure 24: Nearest samples in the private dataset on Camelyon17. In each row, the first column is a generated image (from Fig. 21), and the other columns are its nearest neighbors in the private dataset, sorted by the distance in ascending order. Every fifteen rows correspond to generated image from one class. The distance metric is $\ell_2$ in the **inception embedding space**.

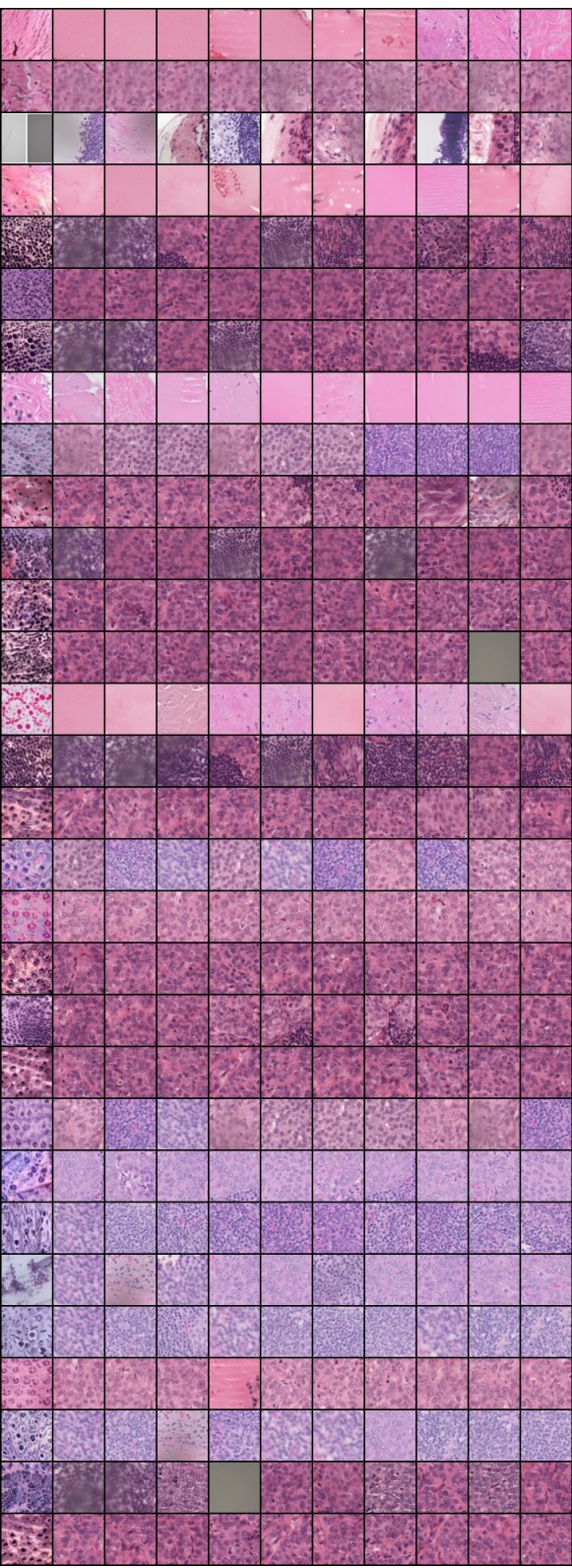

Figure 25: Nearest samples in the private dataset on Camelyon17. In each row, the first column is a generated image (from Fig. 13), and the other columns are its nearest neighbors in the private dataset, sorted by the distance in ascending order. Every fifteen rows correspond to generated image from one class. The distance metric is $\ell_2$ in the **pixel space**.

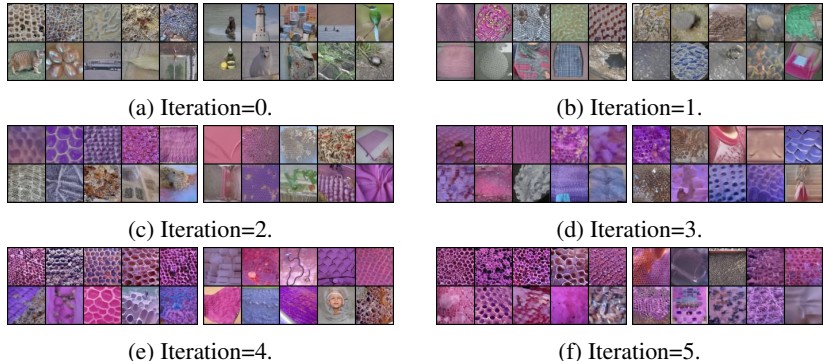

(a) Iteration=0.     (b) Iteration=1.

(c) Iteration=2.     (d) Iteration=3.

(e) Iteration=4.     (f) Iteration=5.

Figure 26: Samples with the highest and the lowest counts in the DP Nearest Neighbors Histogram on Camelyon17. In each subfigure, each row corresponds to one class; the left subfigure shows the samples with the highest counts and the right subfigure shows the samples with the lowest counts.

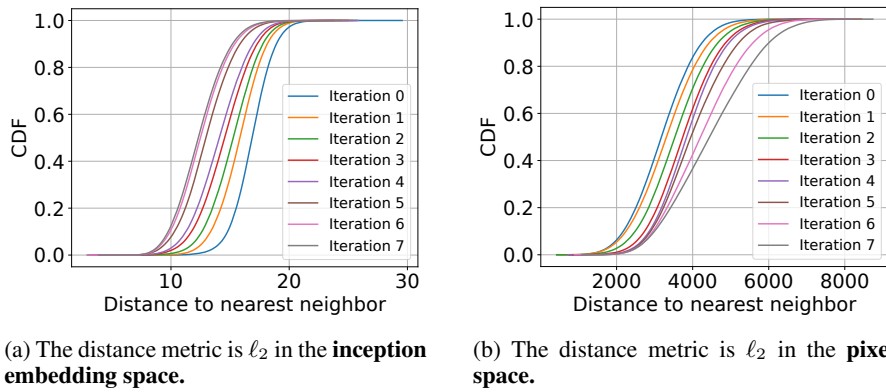

(a) The distance metric is $\ell_2$ in the **inception embedding space.**

(b) The distance metric is $\ell_2$ in the **pixel space.**

Figure 27: CDF of the distributions **between each generated sample and its nearest private samples** on CIFAR10 across different PE iterations. "Iteration 0" refers to the initial random samples from Line 1 in Alg. 1.

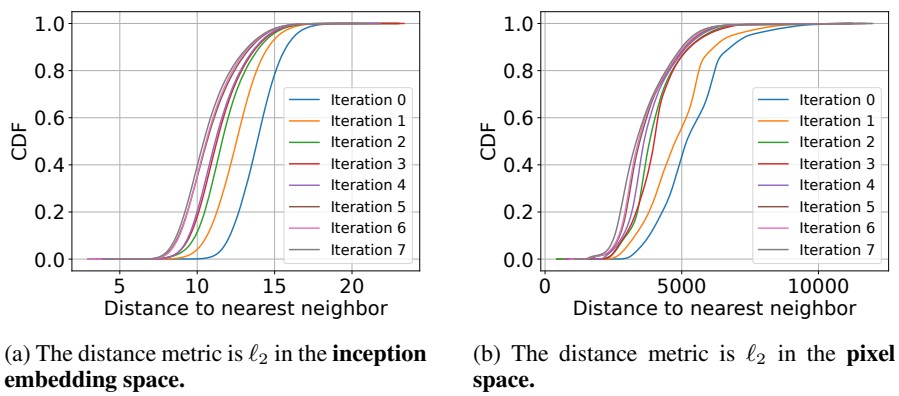

(a) The distance metric is $\ell_2$ in the **inception embedding space.**

(b) The distance metric is $\ell_2$ in the **pixel space.**

Figure 28: CDF of the distributions **between each private sample and its nearest generated samples** on CIFAR10 across different PE iterations. "Iteration 0" refers to the initial random samples from Line 1 in Alg. 1.

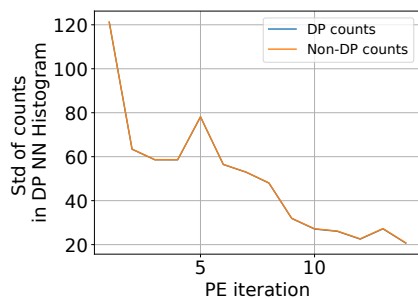

Figure 29: The standard deviation of counts in DP Nearest Neighbors Histogram across different PE iterations on Camelyon17. "DP counts" refers to the counts after adding Gaussian noise and thresholding. "Non-DP counts" refer to the counts before adding Gaussian noise and thresholding.

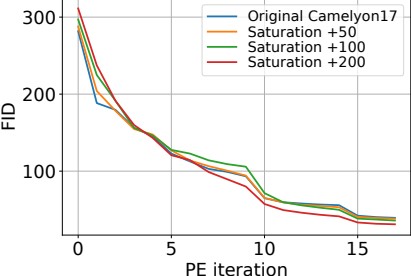

Figure 30: FID vs. PE iterations under Camelyon17 variants with different levels of distribution shifts.

## L    MORE DETAILS AND RESULTS ON STABLE DIFFUSION EXPERIMENTS

**Dataset construction.** We start with cat photos taken by the authors, crop the region around cat faces with a resolution larger than 512x512 manually, and resize the images to 512x512. We construct two datasets, each for one cat with 100 images. See Figs. 31 and 32 for all images. Note that having multiple samples from the same identity is not meaningful from the practical perspective of DP. Instead of regarding these datasets as real-world use cases, they should be treated as "toy datasets" for experimenting DP generative models with a small number of high-resolution images.

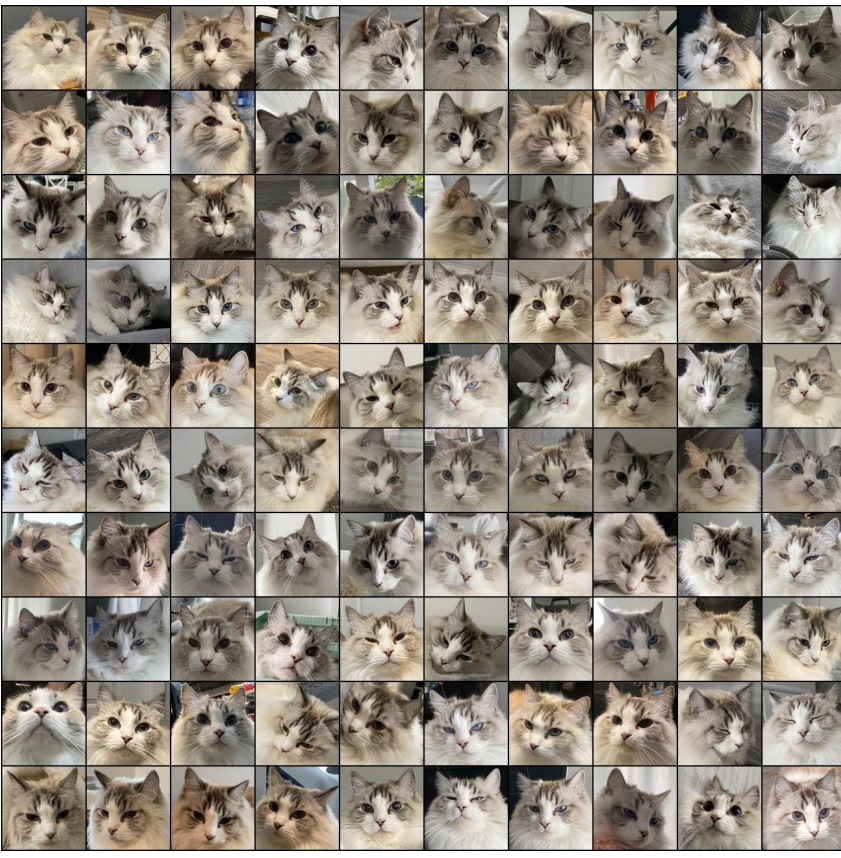

Figure 31: All images from Cat Cookie dataset. The original resolution is 512x512; we resize them to 128x128 here for reducing the file size of the paper.

**API implementation.** We use off-the-shelf open-sourced APIs of Stable Diffusion. For RANDOM_API, we use the text-to-image generation API[18], which is implemented by the standard diffusion models' guided sampling process. For VARIATION_API, we use the image-to-image generation API[19], which allows us to control the degree of variation. Its internal implementation is SDEdit (Meng et al., 2021), which adds noise to the input images and runs diffusion models' denoising process.

**Hyperparameters** We set lookahead degree $k = 8$, and number of generated samples $N_{\text{syn}} = 100$. For RANDOM_API and VARIATION_API, we use the default DDIM sampler with 50 steps. For RANDOM_API, we use the prompt "A photo of ragdoll cat". This gives reaonable cat images but still far away from the private data (Fig. 35). For VARIATION_API, we use variation degrees $[0.98, 0.96, 0.94, 0.92, 0.90, 0.88, 0.84, 0.8, 0.76, 0.72, 0.68, 0.64, 0.6]$ for each iteration re-

---

[18] https://huggingface.co/docs/diffusers/api/pipelines/stable_diffusion/text2img

[19] https://huggingface.co/docs/diffusers/api/pipelines/stable_diffusion/img2img

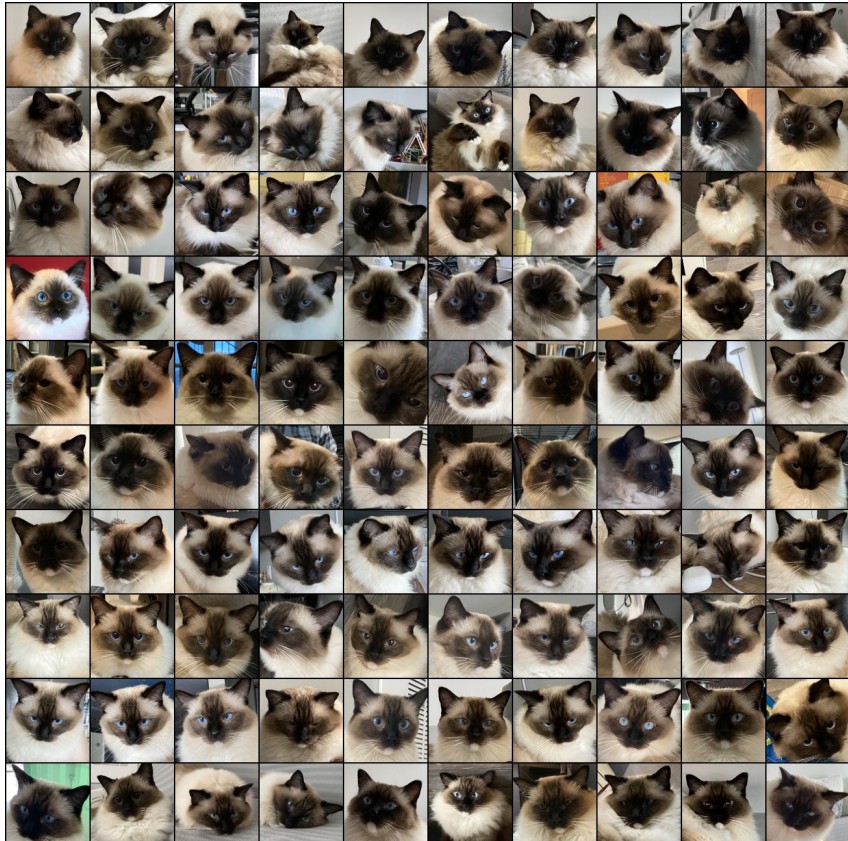

Figure 32: All images from Cat Doudou dataset. The original resolution is 512x512; we resize them to 128x128 here for reducing the file size of the paper.

spectively with the same prompt. We use noise multiplier $\sigma = 2$ and threshold $H = 2$. We use inception embedding for Eq. (1).

**Generated images.** We use the same hyper-parameters to run PE on two datasets separately. This can also be regarded as running the conditional version of PE (Alg. 3) on the whole dataset (with labels Cat Cookie or Cat Doudou) together. All generated images are in Figs. 33 and 34. While the two experiments use completely the same hyper-parameters, and the initial random images are very different from the cats (Fig. 35), our PE can guide the generated distribution in the right direction and the final generated images do capture the key color and characteristics of each of the cats. This demonstrates the effectiveness of PE with large foundation models such as Stable Diffusion.

We also observe that the diversity of generated images (e.g., poses, face directions) is limited compared to the real data. However, given the small number of samples and the tight privacy budget, this is an expected behavior: capturing more fine-grained features of each image would likely violate DP.

**Generated images with more diversity.** To make the generated images more diverse, we can utilize the approach in § 4.2, which passes the generated images through VARIATION_API. We have demonstrated in § 5.1.3 that this approach can generate more samples that are useful for downstream classification tasks. Here, we use it for a different purpose: enriching the diversity of generated samples.

Figure Figs. 36 and 37 show the results. We can see that this simple approach is able to generate cats with a more diverse appearance. This is possible because the foundation model (Stable Diffusion) has good prior knowledge about cats learned from massive pre-training, and PE is able to utilize that effectively.

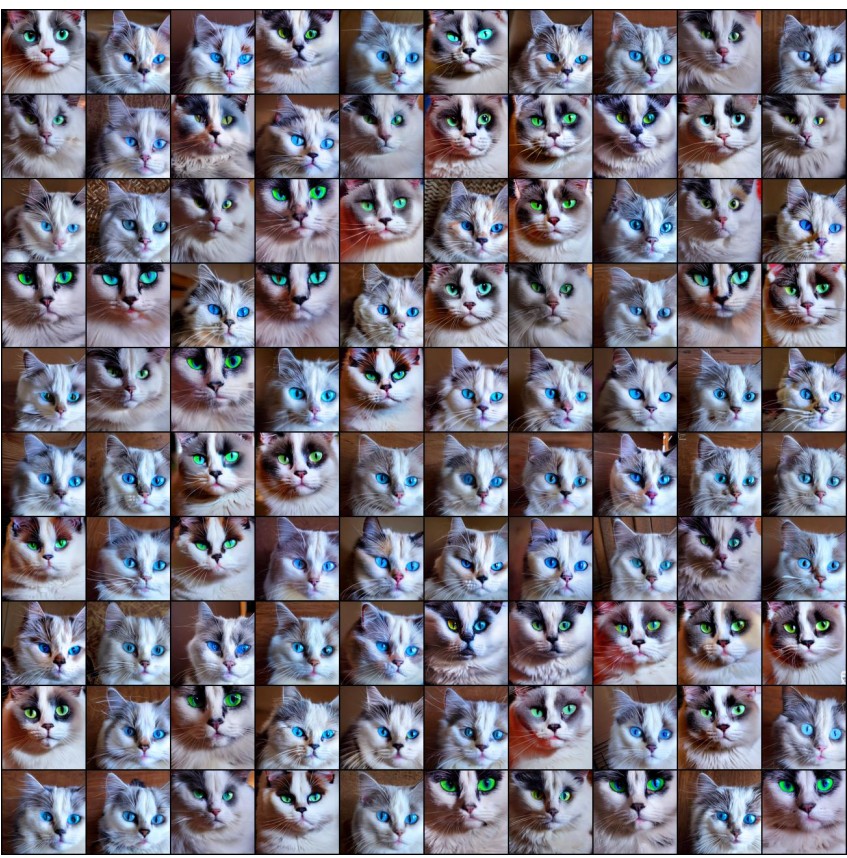

Figure 33: All generated images from Cat Cookie dataset with $(6.62, 10^{-3})$-DP. The original resolution is 512x512; we resize them to 128x128 here for reducing the file size of the paper.

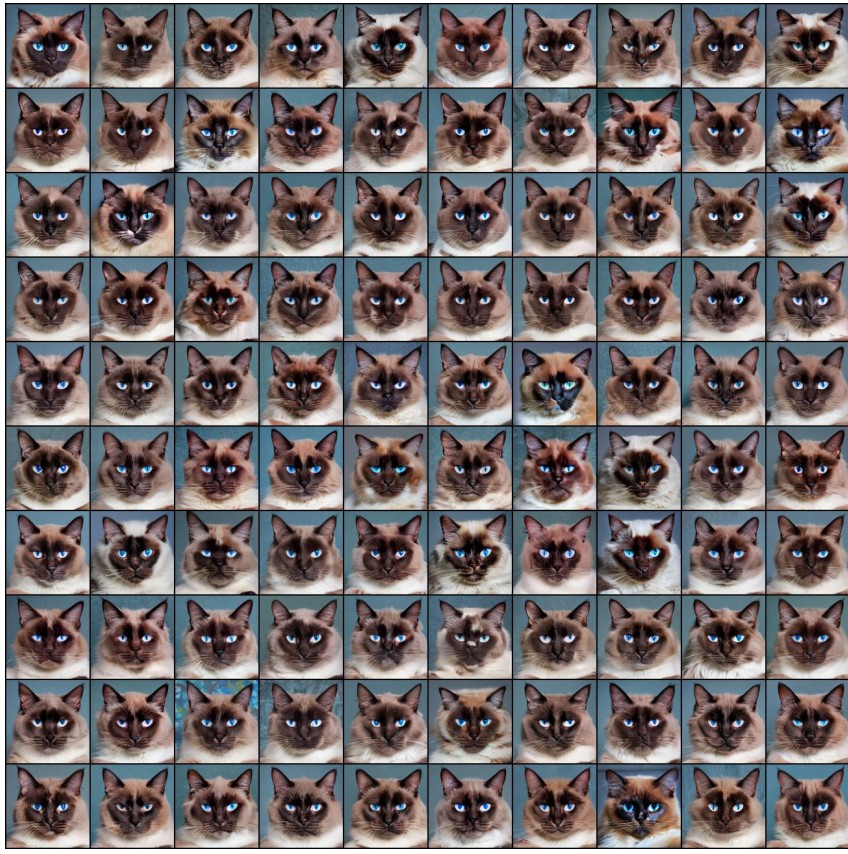

Figure 34: All generated images from Cat Doudou dataset with $(6.62, 10^{-3})$-DP. The original resolution is 512x512; we resize them to 128x128 here for reducing the file size of the paper.

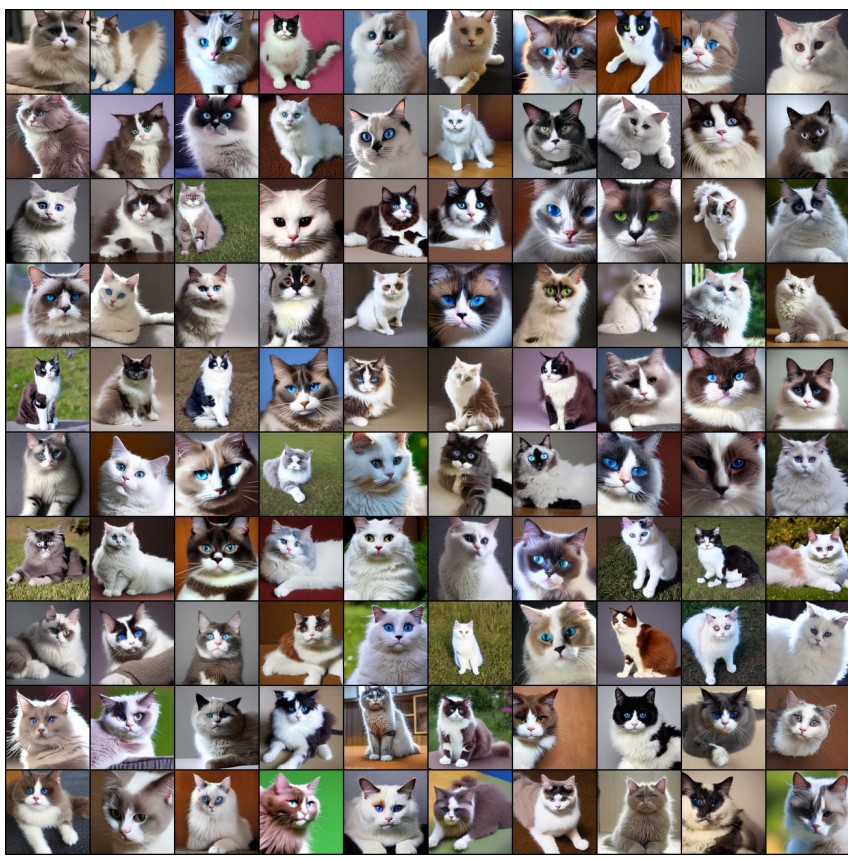

Figure 35: The initial random images from Stable Diffusion. The original resolution is 512x512; we resize them to 128x128 here for reducing the file size of the paper.

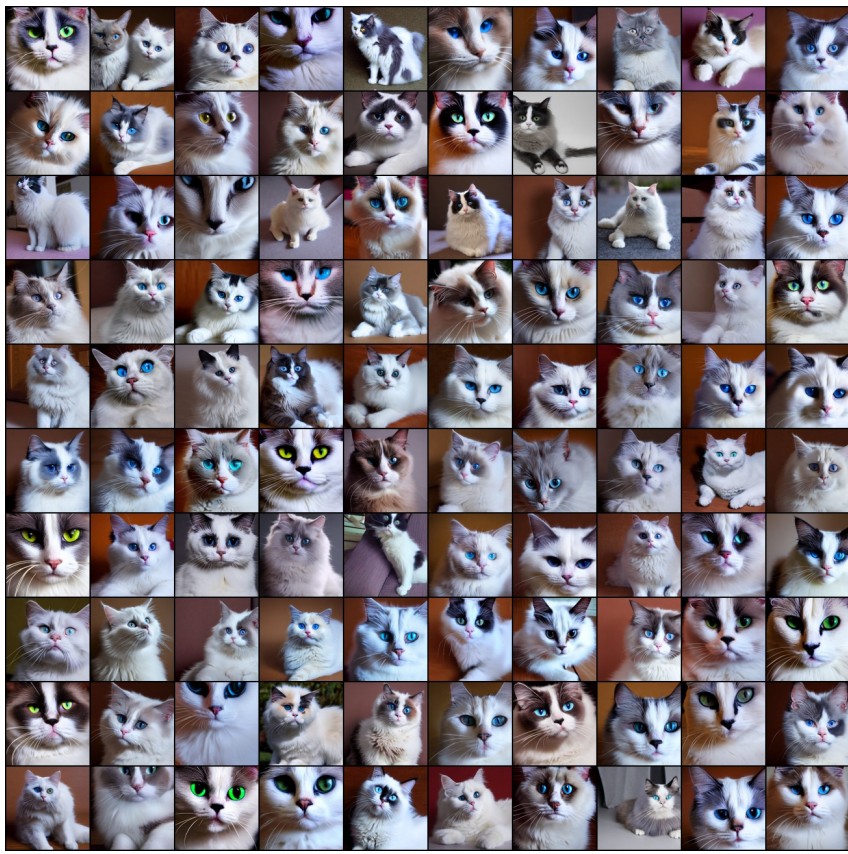

Figure 36: Generated images with enhanced diversity using the approach in § 4.2 on Cat Cookie. The original resolution is 512x512; we resize them to 128x128 here for reducing the file size of the paper.

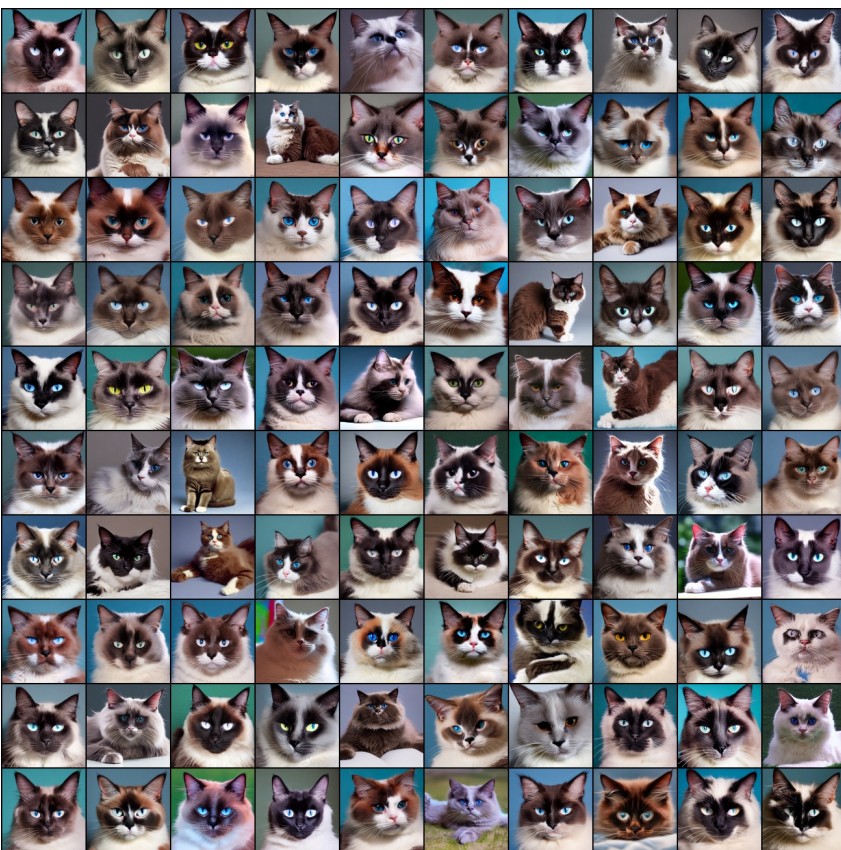

Figure 37: Generated images with enhanced diversity using the approach in § 4.2 on Cat Doudou. The original resolution is 512x512; we resize them to 128x128 here for reducing the file size of the paper.

## M    CINIC EXPERIMENTS

Given that PE is based on sampling from pre-trained foundation models, a natural baseline is to conditionally sample from the classes closest to the private data. For example, in the experiments of § 5.1.1 (CIFAR10 as private data, ImageNet as the public data), it would be interesting to consider baselines that conditionally sample from the same classes in CIFAR10 (with privacy) from a foundation model trained on ImageNet. However, there could be many ways to implement such an algorithm, and the model and hyper-parameter choices could impact the result. To eliminate the influence of these arbitrariness choices, we conduct the following experiment which *gives an estimate on the best FID such an approach could achieve*.

**Experiment setup.** We use CINIC10 dataset (Darlow et al., 2018), which contains CIFAR10 images and the filtered images from ImageNet that belong to CIFAR10 categories. We compute the FID between (a) the subset of CINIC10 images that come from ImageNet (i.e., excluding CIFAR10 images from CINIC10 dataset), and (b) CIFAR10 dataset. This is to simulate the best case when the foundation model (1) learns the ImageNet dataset perfectly and (2) is able to draw only the samples from CIFAR10 classes. In practice, the above two assumptions will certainly not hold; especially, achieving (2) would necessarily incur privacy costs as the knowledge of which samples belong to CIFAR10 is assumed to be private. Therefore, this process gives us a lower bound of the FID (i.e., *the best possible FID*) we can get by such a baseline that samples the same classes as CIFAR10 from an ImageNet model using an arbitrary privacy cost.

**Results.** The FID score from the above is 12.21. We can see from Fig. 4 that PE is able to achieve a smaller FID with $\epsilon$ as small as 0.38. This result suggests that PE does a non-trivial job: it can achieve a lower FID score than simply sampling the same classes as CIFAR10 from an ImageNet model.

## N    MORE ABLATION STUDIES

All ablation studies are conducted by taking the default parameters in unconditional CIFAR10 experiments and modifying one hyperparameter at a time. The default noise multiplier $\sigma = 5 \cdot \sqrt{2}$ and the default threshold $H = 10$.

**Lookahead degree.** Fig. 38 shows how the lookahead degree $k$ (§ 4) impacts the results. We can see that higher lookahead degrees monotonically improve the FID score. However, the marginal benefits diminish as the lookahead degree goes beyond 8, and a higher lookahead degree increases the required of API calls. Throughout all experiments, we used $k = 8$. This experiment suggests that better results can be obtained with a higher $k$. In practice, we suggest users set the lookahead degree as the highest value within their computation or API budget.

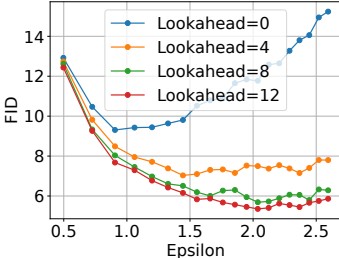

Figure 38: Ablation studies on the lookahead degree $k$ for DP Nearest Neighbors Histogram. Lookahead=0 means lookahead is not used (Eq. (1)).

**Population size.** Fig. 39 shows how the number of generated samples $N_{\text{syn}}$ impacts the results in *non-DP* setting. We can see that, as the number of samples increases, FID score monotonically gets better. This is expected because with more generated samples, there are higher chances to get samples similar to the private data. However, we want to point out that in the DP case, it may not be true, as a large number of samples would flatten the DP Nearest Neighbors Histogram and thus decrease the signal noise ratio.

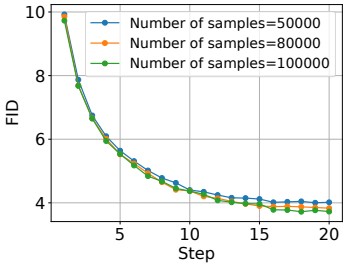

Figure 39: Ablation studies on the number of generated samples $N_{\text{syn}}$ in non-DP setting.

**Histogram threshold.** Fig. 40 shows how the threshold $H$ in DP Nearest Neighbors Histogram impacts the results. We can see that a large threshold results in a faster convergence speed at the beginning. This is because, in the early iterations, many samples are far away from the private data. A larger threshold can effectively remove those bad samples that have a non-zero histogram count due to the added DP noise. However, at a later iteration, the distribution of generated samples is already close to the private data. A large threshold may potentially remove useful samples (e.g., the samples at low-density regions such as classifier boundaries). This may hurt the generated data, as shown in the increasing FID scores at threshold=15. In this paper, we used a fixed threshold across all iterations. These results suggest that an adaptive threshold that gradually decreases might work better.

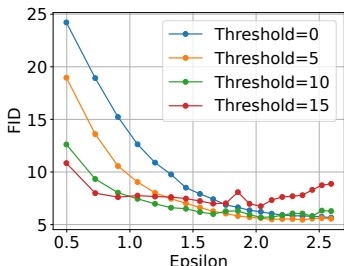

Figure 40: Ablation studies on the threshold $H$ for DP Nearest Neighbors Histogram.

**Embedding.** Fig. 41 compares the results with inception embedding or CLIP embedding in Eq. (1). The results show that both embedding networks work well, suggesting that PE is not too sensitive to the embedding network. Inception embedding works slightly better. One reason is that the inception network is trained on ImageNet, which is similar to a private dataset (CIFAR10). Therefore, it might be better at capturing the properties of images. Another possible reason is that FID score is calculated using inception embeddings, which might lead to some bias that favors inception embedding.

**The number of private samples.** In this experiment, we show how the number of private samples $N_{\text{priv}}$ impacts the performance of PE. Specifically, on CIFAR10, we vary the number of samples $N_{\text{priv}}$ by sub-sampling. For each $N_{\text{priv}} \in \{50000, 20000, 10000, 5000\}$:

- We fix $N_{\text{syn}}/N_{\text{priv}} = 1$ when running PE.

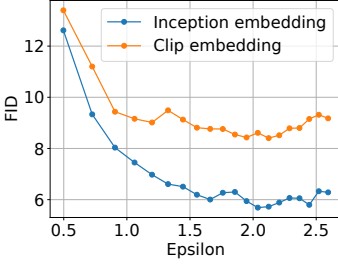

Figure 41: Ablation studies on the embedding network in Eq. (1) to use.

- After PE is done, we use the approach in § 4.2 to augment the number of generated samples back to 50000, and compute the FID between the generated samples and CIFAR10.
- All other hyper-parameters are set the same as the CIFAR10 experiments in Fig. 3.

The results are shown in Table 2, which suggests that larger $N_{\mathrm{priv}}$ helps the performance of PE, similar to the observation in DP-SGD (Anil et al., 2021). We hypothesize that the reason is as follows. When we set $N_{\mathrm{syn}}/N_{\mathrm{priv}} = 1$, although the signal-noise ratio in the DP Nearest Neighbors Histogram remains constant, larger $N_{syn}$ does allow the generated samples to explore larger space, and therefore it is more likely to get a sample closer to the private data, which helps the convergence of PE. Our theorem in App. E also shows that increasing the number of variations (which is controlled by $N_{syn}$) speeds up the convergence.

| Number of samples | FID |
|---|---|
| 50000 (original CIFAR10) | 7.87 |
| 20000 (2.5x smaller) | 10.47 |
| 10000 (5x smaller) | 12.51 |
| 5000 (10x smaller) | 18.60 |

Table 2: FID vs. the size of the private dataset in CIFAR10.

## O  GENERATING MORE SAMPLES

We discussed two ways of generating more samples after PE is done: (1) taking the final generated samples from PE and passing them through VARIATION_API to get more samples (§ 4.2), and (2) using a larger $N_{\mathrm{syn}}$ when running PE (Fig. 39). While we see that both approaches are promising, they have a key difference: the first approach can be applied post hoc without the need to rerun PE, whereas the second approach requires rerunning PE. This difference would result in a very different number of API calls. More concretely, let's say we want to generate $N$ more samples after PE is done. In the former approach, we simply take the final generated samples from PE and call VARIATION_API $N$ times. In contrast, the latter approach requires rerunning the entire PE process and would need to increase the number of API calls by $N \cdot (k+1)$, as we will need to generate more samples at every iteration.

## P  COMPUTATIONAL COST EVALUATION

We compare the GPU hours of the SOTA DP fine-tuning method (Ghalebikesabi et al., 2023) and PE for generating 50k samples in § 5.1.1. Note that PE is designed to use existing pre-trained models and we do so in all experiments. In contrast, DP fine-tuning methods usually require a careful selection of pre-training datasets and architectures (e.g., (Ghalebikesabi et al., 2023) pre-trained their own diffusion models), which could be costly. Even if we ignore this and only consider the computational cost after the pre-training, *the total computational cost of PE is only 37% of (Ghalebikesabi et al., 2023)* while having better sample quality and downstream classification accuracy (§ 5.1.1). See Fig. 42 for a detailed breakdown of the computational cost. Note that PE's computational cost is mostly on the APIs.

The key takeaway is that even if practitioners want to run the APIs locally (i.e., downloading the foundation models and running the APIs locally without using public API providers), there are still benefits of using PE: (1) The computational cost of PE can be smaller than training-based methods. (2) Implementing and deploying PE are easier because PE only requires blackbox APIs of the models and does not require code modifications inside the models.

**Experimental details.** In this experiment, we follow the setting in § 5.1.1, where the user has a private dataset of size $N_{\mathrm{priv}} = 50000$ and a pre-trained model (hosted locally, running on GPUs), and wants to generate a total number of $N_{\mathrm{syn}} = 50000$ synthetic data samples. For the DP fine-tuning baseline, the procedure includes (1) DP fine-tuning the pre-trained model, and then (2) using the fine-tuned model to generate 50000 synthetic samples. Therefore, we break the time into these two parts. For PE, the procedure includes running PE with $N_{\mathrm{syn}} = 50000$. PE includes the steps

of RANDOM_API, VARIATION_API, nearest neighbor search, and feature extraction. Therefore, we break down the time of PE into these four parts.

In the above evaluation, we did not consider the time of generating more samples beyond $N_{\text{syn}} = 50000$, and we discuss its effect here. To generate more samples beyond $N_{\text{syn}} = 50000$ samples, the DP fine-tuning method can directly use the fine-tuned model to generate more samples without more fine-tuning. PE can use the approach in § 4.2, where we pass the generated $N_{\text{syn}}$ samples through VARIATION_API to generate more samples (what we did in § 5.1.3; see results thereof). Note that, for the same size of the model, the same number of generated samples, the same diffusion sampler, and the same total number of denoising steps, *the running time of PE to generate more samples is smaller than the DP fine-tuning baseline.* The reason is as follows. Let's say we use a diffusion sampler with 100 steps in total. In the DP fine-tuning baseline, generated samples must take all 100 steps (i.e., starting from Gaussian noise, passing the image through the diffusion model 100 times iteratively to get the final generated samples). For PE, VARIATION_API is implemented by SDEdit (Meng et al., 2021) (see App. J), where we add noise to input images and let the diffusion model denoise them *starting from the middle of the diffusion process* (e.g., adding Gaussian noise to the image, then passing the image through the diffusion model starting from 20th denoising step for all the rest 80 steps iteratively to get the final generated samples). In other words, each generated does not need to go through all 100 steps, but only a fraction of the steps (80 steps in the above example).

To ensure a fair comparison, we estimate the runtime of both algorithms using 1 NVIDIA V100 32GB GPU.

To evaluate the computational cost of (Ghalebikesabi et al., 2023), we take the open-source diffusion model implementation from (Dhariwal & Nichol, 2021)[20] and modify the hyper-parameters according to (Ghalebikesabi et al., 2023). We obtain a model with 79.9M parameters, slightly smaller than the one reported in (Ghalebikesabi et al., 2023) (80.4M). This difference might be due to other implementation details that are not mentioned in (Ghalebikesabi et al., 2023). To implement DP training, we utilize Opacus library (Yousefpour et al., 2021). To evaluate the fine-tuning cost, we use `torch.cuda.Event` instrumented before and after the core logic of forward and backward pass, ignoring other factors such as data loading time. We estimate the total runtime based on the mean runtime of 10 batches after 10 batches of warmup. We do not implement augmentation multiplicity with data and timestep (Ghalebikesabi et al., 2023); instead, we use multiplicity=1 (i.e., a vanilla diffusion model), and multiply the estimated runtime by 128, the multiplicity used in (Ghalebikesabi et al., 2023). To evaluate the generation cost, we use `torch.cuda.Event` instrumented before and after the core logic of sampling. We estimate the total runtime based on the mean runtime of 10 batches after 1 batch of warmup.

To evaluate the computational cost of our PE, we use a similar method: we use `torch.cuda.Event` instrumented before and after the core logic of each component of our algorithm that involves GPU computation. RANDOM_API and VARIATION_API are estimated based on the mean runtime of 10 batches after 1 batch of warmup. Feature extraction is estimated based on the mean runtime of 90 batches after 10 batch of warmup. The nearest neighbor search is estimated based on 1 run of the full search. We use faiss library[21] for nearest neighbor search. Its implementation is very efficient so its computation time is negligible compared with the total time.

---

[20] https://github.com/openai/guided-diffusion
[21] https://github.com/facebookresearch/faiss

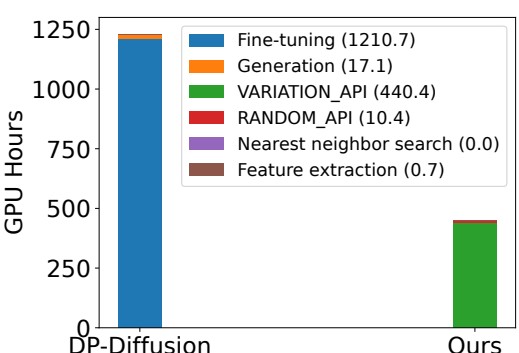

Figure 42: GPU hours (on 1 NVIDIA V100 32GB) required to obtain the samples for § 5.1.1 with DP-Diffusion (Ghalebikesabi et al., 2023) and ours. The legend denotes the steps and the GPU hours; (Ghalebikesabi et al., 2023) contains the fine-tuning and generation steps, whereas ours contains the other steps.

