# OpenReview forum: "Differentially Private Synthetic Data via Foundation Model APIs 1: Images"
_ICLR.cc/2024/Conference — ICLR 2024 poster_

### Official Review · Reviewer_vg5F · 2023-10-21

**Soundness:** 3 good
**Presentation:** 3 good
**Contribution:** 2 fair
**Rating:** 5
**Confidence:** 4

**Summary:**

This work targets the problem of differentially private image synthesis. Unlike prior related works that need full model access in the training, this work chooses to utilize large foundation models, with only the access to APIs (without knowing model weights or even without the need of training).  The PE algorithm is inspired by the evolutionary algorithm, which essentially iteratively refines the generation result so that the generation is close to the private set. The FID and some downstream tasks show its effectiveness.

**Strengths:**

1. Large foundation models have attracted unprecedented attention in recent years. The authors give an example of how to apply the foundation APIs to a real problem.
2. The experimental results look good.
3. The writing is generally easy to follow.

**Weaknesses:**

1. This method is not generic, which only applies to powerful generative AI. If the foundation model is not good enough, I highly doubt the quality of the generation.
2. Though the result looks nice, I am so unconvinced it is true. A generative model basically learns the training distribution, so that it can generate something similar to the training set. However, if the training distribution is different from what you want to generate, how is it possible? Sec 5.1.1 looks reasonable, because the distribution of CIFAR10 has a large overlap with (or maybe is even covered by) ImageNet. However, CAMELYON17 is totally different from ImageNet, why is a diffusion pretrained on ImageNet able to generate CAMELYON17 images? The authors did not give any formal or intuitive explanations on this point.
3. Based on the second concern, it would be more clear and intuitive to highlight samples that are used to update the histogram in each iteration. For example, in Figure 19, iteration=0, what generative samples (I believe not all of them) are closer to the private set and thereby used to update the histogram?
4. By reading Alg. 2, it looks to me that it is possible that the histogram will be more and more concentrated on some samples when the overlap between the generation set and the private set is small, thus implying that the variation of the generation can be low.

**Questions:**

1. I find it hard and a bit confusing to track the pretrained model information in the paper. The term "pretrained model" and "API" seem to be interchangeably used, but you have two APIs. For example, in App. I, you mentioned only one pretrained model, is that your random_API? In  API implementation, you mention both APIs, and the variation_API is SDEdit. Is it also pretrained on ImageNet?
2. It makes no sense to compare the performance at different $(\epsilon, \delta)$-DP, such as the comparison in Sec 5.1.2. Why not targeting the same DP guarantee?
3. In App. H - "Conditional pre-trained networks/APIs", it says that "In the subsequent VARIATION API calls (Line 6), for each image, we will use its associated class label or text prompt as the condition information to the API, and the output samples from VARIATION API will be associated with the same class label or text prompt as the input sample." Do you use this strategy in Sec 5.1.2 experiment? If so, how did the label of ImageNet guide the generation of Camelyn17?
4. Fig. 19(a) is the output of random_API (iteration=0), but some images look weirdly pink-ish, e.g. col 1 row 7, col 7 row 6, col 8 row 10, while the rest looks gray-ish. This is weird if the random_API is well pretrained. Do the authors have any interpretations?

---

> ### Author Response · Authors · 2023-11-18
> **Rebuttal (Q1, Q2)**
>
> Thank you so much for the great suggestions and comments! Please see the answers to all your questions below.
>
> **Q1: This method is not generic, which only applies to powerful generative AI. If the foundation model is not good enough, I highly doubt the quality of the generation.**
>
> **A1:** We respectfully disagree with this assessment.
>
> Firstly, we want to clarify that, in our main experiments (CIFAR10 and Camelyon17), we used a relatively old diffusion model (Nichol & Dhariwal, 2021), which is far from the current state-of-the-art. Secondly, we carefully designed the experiments so that PE uses similar pre-trained models to the ones used in the SOTA DP-fine-tuning method (Ghalebikesabi et al., 2023) for fair experimental comparisons. In particular, their numbers of parameters are similar, and they are pre-trained on the same dataset (ImageNet) (see Section 5.1). Therefore, we do not think that there is evidence suggesting that PE needs more powerful pre-trained foundation models than DP-fine-tuning approaches.
>
> Moreover, as the foundation models are quickly evolving and the most powerful ones are mostly closed-source, API-based methods could potentially be more applicable in the future.
>
> **Q2: Though the result looks nice, I am so unconvinced it is true. A generative model basically learns the training distribution, so that it can generate something similar to the training set. However, if the training distribution is different from what you want to generate, how is it possible? Sec 5.1.1 looks reasonable, because the distribution of CIFAR10 has a large overlap with (or maybe is even covered by) ImageNet. However, CAMELYON17 is totally different from ImageNet, why is a diffusion pretrained on ImageNet able to generate CAMELYON17 images? The authors did not give any formal or intuitive explanations on this point.**
>
> **A2:** Even though the diffusion model is trained on natural images, the support of the generated distribution could be very large due to the formulation and the large latent space (same size as images). More concretely, let’s look at two examples: score-based models (https://arxiv.org/abs/1907.05600, closely related to diffusion models) and the diffusion model we used (Nichol & Dhariwal, 2021). For score-based models, its modeling target is a distribution perturbed by Gaussian noise, and therefore, the generated distribution spans the entire sample space. For diffusion models (Ho et al., 2020), the latent space has the same size as the image, and the last denoising step is modeled as a distribution derived from a Gaussian distribution that spans the entire pixel space (see Section 3.3 of Ho et al. (2020)). Therefore, the generated distribution of diffusion models also spans the entire sample space.
>
> In other words, for any (trained) score-based models or diffusion models, it is theoretically possible to generate images similar to the private dataset (or in fact, generate any images). The problem is that the probability of generating such images is small if there is a large distribution shift from the pre-training data to the private data. PE is effective in guiding the diffusion model to generate samples from the region that is low-density in the original pre-trained distribution but high-density in the private distribution.
>
> To make it more concrete, we have provided how the generated images evolve from natural images to Camelyon17 dataset in Figure 19 in the original submission (Figure 23 in the revision) and the selected/filtered samples by PE in Figure 26 in the revision. At every iteration, PE selects the set of images that are most similar to the Camelyon17 dataset (Figure 26 in the revision). Those images might still appear different from Camelyon17 in early iterations. However, as long as we get images that are more similar to Camelyon17 through VARIATION_API at every iteration, we will make progress, and finally, we can get images similar to Camelyon17 (Figure 17 in the original submission or Figure 21 in the revision).
>
> We added brief explanations in Section 5.1.2, and an expanded version in Appendix J in the revision. We also want to note that (1) we have provided *theoretical* justification of why PE works in Section D in the original submission, and (2) we have provided the source code for reproducing the experiments in the supplementary material.

---

> > ### Author Response · Authors · 2023-11-18
> > **Rebuttal (Q3, Q4, Q5, Q6)**
> >
> > **Q3: Based on the second concern, it would be more clear and intuitive to highlight samples that are used to update the histogram in each iteration. For example, in Figure 19, iteration=0, what generative samples (I believe not all of them) are closer to the private set and thereby used to update the histogram?**
> >
> > **A3:** Thanks for the great idea! Following your suggestions, we add samples with **the highest and the lowest count** for **both CIFAR10 and Camelyon17** in Figure 19 and Figure 26 in the revision. The key observations are summarized below.
> > * On the CIFAR10 dataset, we can see that the DP NN Histogram picks samples similar to the private data as desired. This is more obvious in the first two PE iterations, where the DP NN Histogram assigns high votes on the samples with the correct classes and puts low votes on the samples with incorrect classes.
> > * On the Camelyon17 dataset, we can see that the DP NN Histogram picks samples with the right patterns as the private data and drops samples with more different patterns as desired. As the Camelyon17 dataset is further away from the pre-training dataset ImageNet than CIFAR10, we can see that it converges slower than the case in CIFAR10.
> >
> >
> > **Q4: By reading Alg. 2, it looks to me that it is possible that the histogram will be more and more concentrated on some samples when the overlap between the generation set and the private set is small, thus implying that the variation of the generation can be low.**
> >
> > **A4:** The reality is on the contrary: at the beginning, the histogram is concentrated; as more iterations go on, the histogram becomes more uniform. The reason is as follows. In the beginning, only a few generated samples are close to the private data. Those generated samples get most of the votes from private samples, which results in a concentrated histogram. PE will then pick those highly-voted generated samples and do more variations over them. As a result, private samples that voted for the same generated sample may now find different closest generated samples and distribute the votes, which results in a “more uniform” histogram.
> >
> > To support the above claim, we show the standard deviation of the counts in the DP NN Histogram across different PE iterations on both CIFAR10 and Camelyon17 in Figure 20 and Figure 29 in the revision. The figure shows that the histogram becomes “more uniform” instead of more concentrated across iterations.
> >
> > **Q5: I find it hard and a bit confusing to track the pretrained model information in the paper. The term "pretrained model" and "API" seem to be interchangeably used, but you have two APIs. For example, in App. I, you mentioned only one pretrained model, is that your random_API? In API implementation, you mention both APIs, and the variation_API is SDEdit. Is it also pretrained on ImageNet?**
> >
> > **A5:** Sorry for the confusion! We want to clarify that:
> >
> > * In each experiment, there is only *one* pre-trained model (a diffusion model), and both APIs (RANDOM_API and VARIATION_API) are implemented using the same model.
> > * RANDOM_API is implemented by drawing the samples from the diffusion models using the standard diffusion model sampling process (Appendix I).
> > * SDEdit (Meng et al., 2021) is a method, *not* a model. VARIATION_API is implemented by applying SDEdit on the same diffusion model. SDEdit adds noise to input images and lets the diffusion model denoise them to get their variation (Appendix I).
> >
> > We add clarification in Section 5.1. Hope it is clearer now!
> >
> > **Q6: It makes no sense to compare the performance at different (epsilon,delta)-DP, such as the comparison in Sec 5.1.2. Why not targeting the same DP guarantee?**
> >
> > **A6:** Thank you for the question! Following your suggestions, we rerun PE with a re-computed noise multiplier so that the DP parameter epsilon is exactly 10, the same as the baseline method. The accuracy is 80.33%, slightly higher than 79.56% at epsilon=7.58. We update the results in the revision.
> >
> > Note that these results do not change the takeaway messages in Section 5.1.2: (1) Even with a large distribution shift, PE can still achieve a non-trivial utility of DP synthetic data (much higher than random guess accuracy of 50%). (2) The traditional training-based methods (91.1% accuracy) are still more promising in this case if the privacy-utility trade-off is the only goal. However, given the benefit of API-only assumption and the non-trivial results that PE already got, it is worth further exploiting the potential of PE in future work.

---

> > > ### Author Response · Authors · 2023-11-18
> > > **Rebuttal (Q7, Q8)**
> > >
> > > **Q7: In App. H - "Conditional pre-trained networks/APIs", it says that "In the subsequent VARIATION API calls (Line 6), for each image, we will use its associated class label or text prompt as the condition information to the API, and the output samples from VARIATION API will be associated with the same class label or text prompt as the input sample." Do you use this strategy in Sec 5.1.2 experiment? If so, how did the label of ImageNet guide the generation of Camelyn17?**
> > >
> > > **A7:** Yes, the same strategy was used in all experiments.
> > >
> > > Note that the guidance in PE is *not* through the class label. The only thing we care about in the end is the generated samples, not their associated ImageNet class label. We have to feed an ImageNet class label to the model only because the model requires it. We keep the same class label for the samples generated from VARIATION_API because that was the label condition used to generate the samples. *The true guidance in PE is through DP NN Histogram, where the private samples guide the generation by selecting the closest generated samples.* In that sense, the ImageNet class label is better understood as auxiliary information required by the model, which will be discarded in the end.
> > >
> > > **Q8: Fig. 19(a) is the output of random_API (iteration=0), but some images look weirdly pink-ish, e.g. col 1 row 7, col 7 row 6, col 8 row 10, while the rest looks gray-ish. This is weird if the random_API is well pretrained. Do the authors have any interpretations?**
> > >
> > > **A8:** This is because we used the DDIM sampler (Song et al., 2020) with only 10 sampling steps to draw samples from the diffusion model. It is known in the community that the DDIM sampler with a small number of sampling steps can result in samples with bad quality and/or weird colors. See Figure 6 in https://arxiv.org/pdf/2302.04867v1.pdf, Figures 1,3,5 in https://arxiv.org/pdf/2206.00927.pdf, and Figures 1,6 of https://arxiv.org/pdf/2211.01095.pdf for such examples in other datasets and diffusion models. Better samplers (such as the papers listed above) and more sampling steps can improve the results. In our experiments, we used the DDIM sampler because it is the default one implemented in the Improved Diffusion codebase (the model we used as the API). We only used 10 sampling steps because it is faster to generate samples. We found that such a less optimal configuration is sufficient for PE to work reasonably well so we did not try better configurations.

---

> ### Author Response · Authors · 2023-11-20
> **Follow-up**
>
> Dear Reviewer vg5F,
>
> Thank you again for your valuable time in reviewing our paper! Have we adequately addressed all your concerns? According to your suggestions, we visualized the images with the highest and the lowest counts in the DP NN Histogram, the distribution of the counts in DP NN Histogram, and added PE results with epsilon=10 on the Camelyon17 dataset. We also addressed your other concerns in the rebuttal and the revision.
>
> If you have any further questions, please kindly inform us, and we will be happy to elaborate further before the Nov. 22 deadline.
>
> Thank you!
>
> The authors

---

> ### Comment · Reviewer_vg5F · 2023-11-21
> **Final response**
>
> Thank the authors for all the effort in additional experiments! The newly added figure 26 is very useful.
>
> As a summary:
>
> The technical novelty for achieving DP (adding noise to histogram) is limited by prior works, but I do acknowledge that the authors have done a smart application of foundation API in DP image synthesis.
>
> Moreover, I still remain suspicious about the generation if there is great distribution shift between pretrained public dataset and private dataset. Figure 26 can possibly explain the success of generation on Camelyn17, i.e. the pretrained ImageNet contains a class **honeycomb**, which happens to look similar to the texture in Camelyn17 images, and the later iterations simply generate increasingly more pink-ish honeycombs, which moves closer to the samples in Camelyn17.
>
> Plus, by saying that "this method only applies to powerful generative AI", I mean a well-pretrained model on a largely spanned public dataset. If your pretrained dataset is not diverse enough, this method is not going to work.
>
> I will keep my score, because my original score would be 4 but there is no such option. Thanks again for authors' effort in addressing my questions.

---

> ### Author Response · Authors · 2023-11-22
> **Reply to Reviewer vg5F**
>
> Dear Reviewer vg5F,
>
>
> We very much thank the reviewer for reading our rebuttal and sharing your thoughts! These discussions are very helpful for the paper. We understand that the reviewer has marked it as a “final response”. But we would like to post our response here in case the reviewer expect one.
>
> * **Novelty about “adding noise”.** It was shown that any non-constant deterministic algorithm cannot be DP [1], so **any DP algorithm has to introduce randomness**. Adding noise, proposed back in 2006 [2], is used in many popular DP frameworks such as DP-SGD, PATE, etc. Therefore, we respectively disagree that *"our technical novelty diminishes because we add noise"*. The key novelty/difference between these different algorithms is *where they add noise*. As we explained to Reviewer eX7J, we add noise to the Nearest Neighbor Histogram, which is new in this work.
>
>   We also want to thank the reviewer for acknowledging that “the authors have done a smart application of foundation API in DP image synthesis”.
>
> * **Requiring a well-pretrained model on a largely spanned public dataset.** Thank you for clarifying the question. We agree with the reviewer that a more diverse pretrained dataset is beneficial for PE. We illustrated this intuition in Figure 2 (left). This requirement accords with the current trend in foundation models, which are trained on larger and larger datasets. Please see the post “ To AC, Reviewers, and Public Readers” for more discussions about this point.
>
> * **How PE works in Camelyon17**. Firstly, we would like to thank the reviewer for pointing out honeycomb class in ImageNet, which is very helpful in providing more insights about this experiment. According to the suggestion, we checked our results again. We found that the existence of this class is indeed helpful, but PE does more than just picking this class.
>
>   * **Finding 1: The ImageNet labels of generated samples contain other further-away categories.** The ImageNet labels and their associated number of generated images are: “honeycomb” (164647), “velvet” (83999), “nematode, nematode worm, roundworm” (35045), “handkerchief, hankie, hanky, hankey” (14495), “bib” (3102), “head cabbage” (934), “bubble” (142), “stole” (72). 54.4% images are indeed with label “honeycomb”, but others are from further away classes.
>   * **Finding 2: Simply picking images from honeycomb yields bad fidelity.** We compute the FID between honeycomb images and CIFAR10. The FID score is 162.86, which is much higher than the FID score of the final generated images we get (10.66). Indeed, although honeycomb has a similar a “net-like” pattern as Camelyon17, the details, colors, and structures are still different (see https://images.cv/dataset/honeycomb-image-classification-dataset for a quick visualization of honeycomb images). Figure 26 (a) also suggests that the initial generated diffusion models from ImageNet are still very different from Camelyon17 and the final generated samples in Figure 21.
>
> These findings are very interesting and are deeper than what we understood, and they add a lot value to the paper (thanks to the reviewer). We added the above findings to Appendix J. But we do not see contradictions between these findings and the messages we conveyed in the paper—if the reviewers notice any contradiction, please let us know and we are happy to modify them. Note that this Camelyon17 experiment follows the same experiment setting as the SOTA DP-fine-tuning work (Ghalebikesabi et al., 2023), i.e., both methods use the same pre-training data (ImageNet) and private data (Camelyon17) so they encounter the same challenge (large distribution shift) and the same advantage (the existence of honeycomb). Here, the wording “significant distribution shift” comes from Ghalebikesabi et al. (2023), and we just follow their terminology when describing this experiment (which refers to the large distance between the distributions of the pre-training data and private data).
>
> Thank the reviewer again for this great insight!! Feel free to ping us with any more questions. If we have addressed your final questions satisfactorily, please consider raising the score.
>
> [1] See page 16 of: Dwork, C. and Roth, A., 2014. The algorithmic foundations of differential privacy. Foundations and Trends® in Theoretical Computer Science, 9(3–4), pp.211-407.
>
> [2] Dwork, Cynthia, Frank McSherry, Kobbi Nissim, and Adam Smith. "Calibrating noise to sensitivity in private data analysis." In Theory of Cryptography: Third Theory of Cryptography Conference, TCC 2006, New York, NY, USA, March 4-7, 2006. Proceedings 3, pp. 265-284. Springer Berlin Heidelberg, 2006.

---

### Official Review · Reviewer_wkXZ · 2023-10-29

**Soundness:** 2 fair
**Presentation:** 3 good
**Contribution:** 2 fair
**Rating:** 6
**Confidence:** 5

**Summary:**

The paper introduces a novel approach to generate differentially private (DP) synthetic data via API calls to a foundation model, focusing on image data. The authors present a framework named Private Evolution (PE) -- each evolution iteration involves building a privatized histogram, bootstrapping the distribution, and calling the API to generate more similar images. Experiments on CIFAR10 and a medical image dataset demonstrates compelling results.

**Strengths:**

1. The paper introduces a novel approach to generating DP synthetic data without the need for training, which is a significant departure from traditional methods.
2. The utilization of the foundation model APIs allow for easier deployment and the exploitation of the capability obtained in large-scale pre-training.
3. The paper presents strong experimental results, particularly the improvement in privacy cost while achieving competitive FID scores on CIFAR10.

**Weaknesses:**

1. **Privacy guarantee**: 1) The query at each iteration depends on the output of the last iteration; such adaptive queries lead to correlated outputs. This may potentially amplify the privacy loss. 2) If the underlying model of the API memorizes information from the queries (and can even update the model based on user interactions, such as what OpenAI does for GPT-4), there could be long-term privacy implications, especially that at later iterations, the data fed back to the API are highly similar to the private data.
2. **Generalization beyond images**: The algorithm in this paper is highly dependent on the capability of the API. In this work the authors only focused on the image data. Can the authors comment on the generalizability of the method on other data types, e.g., text data, tabular data, etc.?
3. **Conditional generation**: The authors treat the conditional generation as unconditional generation for each class. This is a valid approach, but think about the scenario where there are many classes and only a few images per class (e.g., CelebA dataset with 10k classes and ~20 images per class), it may be better for the algorithm to not limit itself to one class of data. Can the authors comment on how they would adapt the algorithm for such type of data, and the utility? (One might expect that the threshold needs to be low to accommodate for the few number of images per class, but this means more noise).
4. **Monetary cost of the algorithm**: The authors mentioned the foundation models "including GPT4, Bard, and DALLE2 (that) only provide API access without releasing model weights or code". For those models: 1) often the API calls are not inexpensive; 2) they may support fine-tuning access. Can the authors discuss the costs of the two routes (DPSDA vs. DP finetuning)?
5. **Diversity-utility trade-off**: I'm curious about whether calling more VARIATION_API will hurt the utility. Apparently the diversity increases because there's no longer constraints in producing nearest neighbor images. But without this constraint, and without the knowledge of how the foundation model API produces a "similar" image, how can we be sure about retaining similar level of data utility? The results in Fig. 5 are encouraging results, but I'd appreciate more evaluation to directly corroborate the argument, e.g., using the initial 50k images to train a classifier, and compare with another 50k after calling the API for a few times.
6. **Missing related work**: A few published works on DP synthetic data are missing, to name a few, [1,2,3].
7. **Evaluation**:
   1. Fig. 6 presented generated Camelyon17 images with (9.92, 3x10^{-6})-DP. I wonder why the authors didn't report the accuracy on this generated dataset while instead reported the numbers for the \epsilon=7.58 dataset? Clearly it's more fair to compare the \epsilon=9.92 results with the \epsilon=10 results from the prior work. I appreciate it if the authors can add the result.
   2. More details on ensemble in the main paper is appreciated. Are the 5 classifiers trained on disjoint datasets each of 0.2M samples? Is this the same setup as Ghalebikesabi et al.? (because 0.2M samples for a CIFAR classifier seem a lot.)
   3. I appreciate the effort of the authors putting up their own private dataset for evaluation. However, I have reservations about the experimental setting in Sec 5.2. In its current form, Sec 5.2 only serves the visual demonstration purpose. But one dataset with datapoints contributed all by one single identity makes little sense when talking about DP. A dataset with e.g. 5 cats and 20 images each would be better. It would be possible to further evaluate the utility of the synthesized data then.
8. **Minor**
   1. Grammar issue: Sec 4.1, "the (privatized) number of private numbers whose"
   2. The position and the order of the footnotes are messed up
   3. It is not appropriate to use "Cat Cookie" and "Cat Doudou" in a research paper. Use Cat A and Cat B instead

**References**

[1] Vinaroz, M., Charusaie, M. A., Harder, F., Adamczewski, K., & Park, M. J. (2022, June). Hermite polynomial features for private data generation. In International Conference on Machine Learning (pp. 22300-22324). PMLR.

[2] Hu, Y., Wu, F., Li, Q., Long, Y., Garrido, G., Ge, C., ... & Song, D. (2023, October). SoK: Privacy-Preserving Data Synthesis. In 2024 IEEE Symposium on Security and Privacy (SP) (pp. 2-2). IEEE Computer Society.

[3] Cao, T., Bie, A., Vahdat, A., Fidler, S., & Kreis, K. (2021). Don’t generate me: Training differentially private generative models with sinkhorn divergence. Advances in Neural Information Processing Systems, 34, 12480-12492.

**Questions:**

Most of the points in weaknesses are presented as questions. I'd appreciate it if the authors can provide corresponding responses.

---

> ### Author Response · Authors · 2023-11-18
> **Rebuttal (Q1)**
>
> Thank you so much for the great suggestions and comments! Please see the answers to all your questions below.
>
> **Q1.1: The query at each iteration depends on the output of the last iteration; such adaptive queries lead to correlated outputs. This may potentially amplify the privacy loss.**
>
> **A1.1:** We want to clarify that our privacy analysis already takes the adaptive queries into account. The related discussions are in Step 3 and Step 4 of the privacy analysis in Section 4.3.
> It is well known that differential privacy satisfies the adaptive composition theorem (see [1]), i.e., if $M_1$ and $M_2$ are two DP algorithms, then first running $M_1$ on database $D$ to get $M_1(D)$ and then running $M_2$ on $D$ as well as the output of $M_1$, i.e., $M_2(D,M_1(D))$ is still differentially private. (Here we need that $M_2(D,z)$ is DP for all values of z, which is true in our case.) This is how we can care of any correlations that arise.
>
>
> In particular, in Step 3, we mention that PE can be regarded as $T$ (adaptive) compositions of the Gaussian mechanism. Then, in Step 4, we use the property that such composition can be regarded as one Gaussian mechanism with noise multiplier $\sigma/\sqrt{T}$. It is a standard result from Dong et al. (2022) (see Corollary 3.3 therein).
>
> Please let us know if it is unclear, and we are happy to clarify further!
>
> [1] Dwork, Cynthia, and Aaron Roth. "The algorithmic foundations of differential privacy." Foundations and Trends® in Theoretical Computer Science 9.3–4 (2014): 211-407.
>
> **Q1.2: If the underlying model of the API memorizes information from the queries (and can even update the model based on user interactions, such as what OpenAI does for GPT-4), there could be long-term privacy implications, especially that at later iterations, the data fed back to the API are highly similar to the private data.**
>
> **A1.2:** We want to clarify that releasing all the (intermediate) generated sets $S_1,\dots,S_T$ also satisfies the same DP guarantees as the final generated dataset. Note that the API only sees $S_1,\dots,S_T$ during the interaction. Therefore, even if the API memorizes all the previous queries and feeds them back at later PE iterations, it does not violate the DP guarantee. We discussed this point in Section 4.3; please let us know if you have further questions.
>
> ---
> **(Please see the last block for Q2)**

---

> > ### Author Response · Authors · 2023-11-18
> > **Rebuttal (Q3, Q4)**
> >
> > **Q3: The authors treat the conditional generation as unconditional generation for each class. This is a valid approach, but think about the scenario where there are many classes and only a few images per class (e.g., CelebA dataset with 10k classes and ~20 images per class), it may be better for the algorithm to not limit itself to one class of data. Can the authors comment on how they would adapt the algorithm for such type of data, and the utility? (One might expect that the threshold needs to be low to accommodate for the few number of images per class, but this means more noise).**
> >
> > **A3:** Thanks for the great question!
> >
> > In the paper, we test PE in the setting where there are only 100 images (the cat dataset), and PE works reasonably well. Although we did not test it on an even smaller dataset, we believe that the number of private samples ($N_{priv}$) itself is not the bottleneck of the performance of PE; rather, the ratio between the number of generated samples ($N_{syn}$) and the number of private samples, i.e., $N_{syn}/N_{priv}$, is more important for the performance of PE. Note that $N_{priv}$ determines the total of the votes from the private samples in the DP NN histogram (Line 2-4 in Algorithm 2), and Gaussian noise is added to all positions in the DP NN histogram of size $N_{syn}$ (Line 5 in Algorithm 2). Therefore, if $N_{syn}/N_{priv}$ is too large, the true votes in the DP NN histogram will be destroyed by the Gaussian noise. In all our experiments, we set $N_{syn}/N_{priv}=1$ and it works well empirically. Note that it does not mean that PE can only generate as many samples as the private dataset; as we illustrated in Sections 4.2 and 5.1.3, PE can generate unlimited (useful) samples by calling VARIATION_API on these $N_{syn}$ samples.
> >
> > We also want to clarify that the threshold $H$ does not affect the privacy guarantees; it only affects the utility. We conducted the ablation study across a wide range of values (0~15) (see Figure 31 in the original submission or Figure 40 in the revision). Empirically, we found that setting threshold $\in[1,2]\sigma$ ($\sigma$ is the noise multiplier) works well across all datasets of different sizes we evaluated.
> >
> > Hope this answers your question :)
> >
> > **Q4: The authors mentioned the foundation models "including GPT4, Bard, and DALLE2 (that) only provide API access without releasing model weights or code". For those models: 1) often the API calls are not inexpensive; 2) they may support fine-tuning access. Can the authors discuss the costs of the two routes (DPSDA vs. DP finetuning)?**
> >
> > **A4:** We discuss the two sub-questions separately below.
> >
> > * Indeed, the APIs of those *black box models* could be expensive. In this paper, we did not take the number of API calls into account when optimizing PE. One future work is to optimize the number of API calls along with privacy-utility tradeoffs. We discussed this point in Section 6.
> >
> >   We also want to mention that, besides *black box models*, PE can also use *local models* such as Improved Diffusion and Stable Diffusion (as in our experiments). In this case, both PE and DP fine-tuning are feasible. We showed in Appendix O of the original submission that **the computational cost of PE can be smaller than DP fine-tuning**. This demonstrates the practical value of PE.
> > * Indeed, some of those models provide fine-tuning APIs. However, so far they do not support DP fine-tuning and do not provide gradients. Also, uploading sensitive data to these APIs controlled by other companies can lead to privacy violations. We explained it in footnote 2.

---

> > > ### Author Response · Authors · 2023-11-18
> > > **Rebuttal (Q5, Q6, Q7)**
> > >
> > > **Q5: I'm curious about whether calling more VARIATION_API will hurt the utility. Apparently the diversity increases because there's no longer constraints in producing nearest neighbor images. But without this constraint, and without the knowledge of how the foundation model API produces a "similar" image, how can we be sure about retaining similar level of data utility? The results in Fig. 5 are encouraging results, but I'd appreciate more evaluation to directly corroborate the argument, e.g., using the initial 50k images to train a classifier, and compare with another 50k after calling the API for a few times.**
> > >
> > > **A5:** Sorry for the confusion, but we want to clarify how “generating more samples” in Figure 5 is done. The reviewer is right that if we recursively call VARIATION_API multiple times, i.e., VARIATION_API(VARIATION_API(...($S_{syn}$))), then the generated samples may not have good utility because the generated samples drift away from the private samples with each call of VARIATION_API. **However, it is not what “generating more samples” is implemented (see Section 4.2).** Instead, there is only one call of VARIATION_API for each sample. For example, to generate 1M samples from the $S_{syn}$ with 50000 samples, we call VARIATION_API 20 times, each with $S_{syn}$ as input: [VARIATION_API($S_{syn}$), …, VARIATION_API($S_{syn}$)]. If one wants to tune the diversity of these augmented samples, the variation degree parameter $v$ (Section 3.3) can be adjusted.
> > >
> > > We clarify the above points in Section 4.2. Thank you for pointing it out!
> > >
> > > **Q6: Missing related work**
> > >
> > > **A6:** Thanks for mentioning the important related work. We added the citations and discussion in the revision.
> > >
> > > **Q7.1: Fig. 6 presented generated Camelyon17 images with (9.92, 3x10^{-6})-DP. I wonder why the authors didn't report the accuracy on this generated dataset while instead reported the numbers for the \epsilon=7.58 dataset? Clearly it's more fair to compare the \epsilon=9.92 results with the \epsilon=10 results from the prior work. I appreciate it if the authors can add the result.**
> > >
> > > **A7.1:** Thank you for the question! Following your suggestions, we add two experiments: (1) We compute the classification accuracy of the epsilon=7.58 dataset. (2) We rerun PE with a re-computed noise multiplier so that the DP parameter epsilon is exactly 10, the same as the baseline method.
> > >
> > > The results are as follows. (1) The epsilon=9.92 dataset has an accuracy of 78.53%, slightly lower than the accuracy under epsilon=7.58 (79.56%). This accords with the finding in Appendix M that the data quality might degrade at the later stage of PE when the hyper-parameters are not optimal. Nevertheless, these two accuracies are very close. (2) The epsilon=10 dataset has an accuracy of 80.33%, slightly higher than the above two cases. We update the results in the revision.
> > >
> > > Note that these results do not change the takeaway messages in Section 5.1.2: (1) Even with a large distribution shift, PE can still achieve a non-trivial utility of DP synthetic data (much higher than random guess accuracy of 50%). (2) The traditional training-based methods (91.1% accuracy) are still more promising under large distribution shifts if the privacy-utility trade-off is the only goal. However, given the benefit of API-only assumption and the non-trivial results that PE already got, it is worth further exploiting the potential of PE in future work.
> > >
> > > **Q7.2: More details on ensemble in the main paper is appreciated. Are the 5 classifiers trained on disjoint datasets each of 0.2M samples? Is this the same setup as Ghalebikesabi et al.? (because 0.2M samples for a CIFAR classifier seem a lot.)**
> > >
> > > **A7.2:** We follow the same setup as Ghalebikesabi et al. (2023). For the right-most point in Figure 5 (1M samples), we train 5 classifiers with the same hyper-parameters on all 1M samples; the only difference between these 5 classifiers is the random seed. The ensemble of the classifier is implemented by ensembling the logits (Appendix I).

---

> > > > ### Author Response · Authors · 2023-11-18
> > > > **Rebuttal (Q7, Q8)**
> > > >
> > > > **Q7.3: I appreciate the effort of the authors putting up their own private dataset for evaluation. However, I have reservations about the experimental setting in Sec 5.2. In its current form, Sec 5.2 only serves the visual demonstration purpose. But one dataset with datapoints contributed all by one single identity makes little sense when talking about DP. A dataset with e.g. 5 cats and 20 images each would be better. It would be possible to further evaluate the utility of the synthesized data then.**
> > > >
> > > > **A7.3:** This is a great question! We also gave thoughtful consideration to this point when we did the experiment. Even though the training set of Stable Diffusion is public, it is hard to check if a public image or its variants (e.g., cropped, scaled) have been used to produce images in it. Therefore, we decided not to use public datasets, and we resorted to constructing our own private dataset so that we are 100% sure that the private data and the pre-training data have no overlap. We discussed the above considerations in footnote 6 in the original submission.
> > > >
> > > > To construct our private dataset, we did not come up with a better way than using the photos of the *only* two cats we have. That being said, we agree with the reviewer that having multiple samples from the same identity is not meaningful from the practical perspective of DP. Instead of regarding this dataset as a real-world use case, we would rather treat it as a “toy dataset”, the same as other widely used benchmarks in DP literature including CIFAR10 and ImageNet, where there are also duplicated and/or similar samples from the same identity. The experiment of PE on this cat dataset is not for showing a real use case but is rather a proof-of-concept for showing “the feasibility of applying PE on large foundation models with Stable Diffusion” (Section 5.2).
> > > >
> > > > We added more clarifications to the above question in Section K. Thank you again for bringing it up!
> > > >
> > > > **Q8: typos.**
> > > >
> > > > **A8:** Thank you for catching these issues! We fixed them in the revision.

---

> > ### Author Response · Authors · 2023-11-19
> > **Rebuttal (Q2)**
> >
> > **Q2: The algorithm in this paper is highly dependent on the capability of the API. In this work the authors only focused on the image data. Can the authors comment on the generalizability of the method on other data types, e.g., text data, tabular data, etc.?**
> >
> > **A2:** Thank you for the question. We discussed briefly in the original submission how PE can be implemented for text, and we added initial results showing that PE indeed works for texts below.
> >
> > * **How PE can be implemented for text.** We discussed in Section 3.3: “In our algorithm design and experiments, we use 2 APIs, both of which are either directly provided in the APIs of popular models (e.g., DALLE 2, Stable Diffusion **or can be easily implemented by adapting current APIs (e.g., using appropriate text prompts in GPT APIs)**”. Indeed, the natural language interface of large language models (LLMs) gives us even more flexibility in implementing RANDOM_API and VARIATION_API (for texts) than the diffusion models (for images). Let’s take email datasets as an example. For RANDOM_API, we can use the prompt “Please generate a random email:” to query the LLM. For VARIATION_API, we can use the prompt “Please rephrase the following email: <the original email>” to query the LLM. Such an approach works for any LLMs that provide chat capability (e.g., GPTs, LLaMAs). The above prompts are just a simple example; given LLMs’ powerful instruction-following capability, more prompt engineering can potentially give better results.
> >
> > * **Results of PE on texts.** We evaluate PE on text in our follow-up work. We compare PE with the recent ACL 2023 paper on DP synthetic text generation [1]. We follow their experimental setting:
> >
> >   * We take GPT2, GPT2-Medium, and GPT2-Large as the pre-trained model (to implement RANDOM_API and VARIATION_API in PE, or to DP-fine-tune in [1]).
> >   * We consider the Yelp review dataset and the private data.
> >   * We consider rating classification and category classification as the two downstream tasks for evaluating the utility.
> >   * We set the DP parameter epsilon=4.
> >
> > The following table shows the results.
> >
> >
> > |                |             | Classification accuracy on rating | Classification accuracy on category |
> > |----------------|-------------|-----------------------------------|-------------------------------------|
> > | DP fine-tuning [1] | GPT2        | 66.56%                            | 74.78%                              |
> > | PE             | GPT2        | 65.62%                            | 74.75%                              |
> > | DP fine-tuning [1] | GPT2-Medium | 67.56%                            | 74.86%                              |
> > | PE             | GPT2-Medium | 66.85%                            | 75.01%                              |
> > | DP fine-tuning [1] | GPT2-Large  | 69.36%                            | 75.68%                              |
> > | PE             | GPT2-Large  | 67.49%                            | 74.64%                              |
> >
> > As a reference, directly training a DP classifier on the original data achieves 70.14% accuracy on rating and 76.44% accuracy on category. We can see that PE achieves similar (and sometimes better) accuracies compared to the state-of-the-art DP fine-tuning approach, no matter whether the pre-trained model is GPT2, GPT-Medium, or GPT2-Large.

---

> ### Author Response · Authors · 2023-11-20
> **Follow-up**
>
> Dear Reviewer wkXZ,
>
> Thank you again for your valuable time in reviewing our paper! Have we adequately addressed all your concerns? According to your suggestions, we added the results of PE on text, and PE results with epsilon=10 on the Camelyon17 dataset. We also addressed your other concerns in the rebuttal and the revision.
>
> If you have any further questions, please kindly inform us, and we will be happy to elaborate further before the Nov. 22 deadline.
>
> Thank you!
>
> The authors

---

> > ### Author Response · Authors · 2023-11-23
> >
> > Dear Reviewer wkXZ,
> >
> > Have we addressed your concerns sufficiently? We have not heard back from you since we posted the rebuttal 4 days ago, and the discussion phase ends in 5 hours. Thank you again for your valuable time and suggestions!
> >
> > Thank you,
> >
> > The Authors

---

> > > ### Comment · Reviewer_wkXZ · 2023-11-23
> > >
> > > The authors’ response has addressed most of my questions, and I will increase my score by 1. On the other hand, I do believe some of the concerns raised by Reviewer vg5F are valid, and should be taken into consideration when the AC makes the final decision.
> > >
> > > I am taking the thanksgiving holiday and I apologize for not responding promptly.

---

> > > > ### Author Response · Authors · 2023-11-23
> > > >
> > > > Thanks very much for taking the time to read our response during the holiday! We really appreciate it.
> > > >
> > > > Regarding the concerns from Reviewer vg5F, we have provided our responses to *all* their final questions.
> > > >
> > > > Happy Thanksgiving!

---

### Official Review · Reviewer_yzwb · 2023-11-01

**Soundness:** 3 good
**Presentation:** 4 excellent
**Contribution:** 3 good
**Rating:** 8
**Confidence:** 4

**Summary:**

This paper focuses on the problem of generating differentially private synthetic data. In particular, this paper proposes a DP Synthetic Data via APIs method to generate a DP synthetic dataset whose distance to private dataset is minimized by utilizing foundation model inference APIs while protecting privacy of private samples. The proposed method mainly works as follows:
1) Calling RANDOM API to initialize the population through randomly generating samples;
2) Differentially private evaluation of the usefulness of each sample in the population by computing the the noisy number of private samples whose nearest neighbor in the population is this particular sample; and
3) Calling VARIATION API to generate variants of the useful samples.

**Strengths:**

1)  A novel way of addressing the problem of generating differentially private synthetic data through iteratively using private samples to vote for the most similar samples generated from the blackbox model and ask the blackbox models to generate more of those similar samples;

2) A nice application of leveraging the power of large foundation models;

3) Providing theoretical analysis proving the convergence of the distribution of generated samples by the proposed method to the private distribution

4) Very clear presentation and easy-to-follow

5) Proposing a training-free approach while matching or even outperforming state-of-the-art (SOTA) methods that they need a customized training process and significant ML engineering efforts

**Weaknesses:**

1) Can you discuss the effect of the size of the private dataset on the performance of the proposed method?
2) Incomplete downstream classification accuracy v.s. privacy. Figure 5 compares the downstream classification accuracy of the proposed method versus another DP synthetic data generation approach. Can you compare the proposed method with SOTA DP training models on raw data to see which one is better for the downstream classification tasks?
3) Overclaim. Section 4 starts by saying "Data type. While our framework above and our algorithms in § 4 are general for any data type, we focus on images in our experiments." However, it is not clear how this can be done for example for text because of the complexity of textual information and also for audio because of the lack of speech-based APIs. I would either remove this statement or try to discuss how this extension can be done.
4) As this paper says in Section 4 "Foundation models have a broad and general model of our world from their extensive training data." so how do you make sure the images that you consider as private (for example CIFAR-10 in your case) are not used in the training of foundation models of used APIs? I think this somehow affects the practicality of the proposed method in practice. After reading the experiment section, it seems this paper does not use real-world APIs and instead, this paper tries to simulate this in a controlled environment by considering pre-trained models.
5) The claim of the proposed method being easier to deploy than existing methods is not supported. I would at least empirically validate this by measuring its costs in comparison to existing ones.
6) It would be good to also talk about the limitations of DP synthetic data in general especially when data is distributed across users and cannot be centralised in one place.

**Questions:**

I have made some suggestions in the Weaknesses box. I am already happy with this submission, but willing to increase my score further if the authors can address those suggestions.

---

> ### Author Response · Authors · 2023-11-18
> **Rebuttal (Q1, Q2)**
>
> Thank you so much for the great suggestions and comments! Please see the answers to all your questions below.
>
> **Q1: Can you discuss the effect of the size of the private dataset on the performance of the proposed method?**
>
> **A1:** In our experiments, the number of private samples varies widely, ranging from 100 (for the cat dataset) to 50,000 (for CIFAR10) and 302,436 (for Camelyon17). We observe that PE works reasonably well across all these cases.
>
> Compared to the number of private samples ($N_{priv}$), the ratio between the number of generated samples ($N_{syn}$) and the number of private samples, i.e., $N_{syn}/N_{priv}$, is more important for the performance of PE. Note that $N_{priv}$ determines the total of the votes from the private samples in the DP NN histogram (Line 2-4 in Algorithm 2), and Gaussian noise is added to all positions in the DP NN histogram of size $N_{syn}$ (Line 5 in Algorithm 2). Therefore, if $N_{syn}/N_{priv}$ is too large, the true votes in the DP NN histogram will be destroyed by the Gaussian noise. In all our experiments, we set $N_{syn}/N_{priv}=1$ and it works well empirically across all datasets. Note that it does not mean that PE can only generate as many samples as the private dataset; as we illustrated in Sections 4.2 and 5.1.3, PE can generate unlimited (useful) samples in CIFAR10 by calling VARIATION_API on these N_{syn} samples.
>
>
> **Q2: Incomplete downstream classification accuracy v.s. privacy. Figure 5 compares the downstream classification accuracy of the proposed method versus another DP synthetic data generation approach. Can you compare the proposed method with SOTA DP training models on raw data to see which one is better for the downstream classification tasks?**
>
> **A2:** The SOTA DP classifier pre-trained on ImageNet (without DP) and fine-tuned on CIFAR10 (with DP) [1] achieves 94.8% and 95.4% accuracies with epsilon=1 and 2 respectively. It is not surprising that DP classifiers outperform PE (and other DP synthetic data approaches as well) on classification tasks, as DP classifiers are targeted at and optimized for a single task whereas DP synthetic data is general-purpose. See similar discussions in Section 5.2 of the SOTA DP Diffusion work (Ghalebikesabi et al., 2023).
>
> We added the numbers and discussions in Section 5.1.1 of the revision.
>
> [1] De, Soham, et al. "Unlocking high-accuracy differentially private image classification through scale." arXiv preprint arXiv:2204.13650 (2022).

---

> > ### Author Response · Authors · 2023-11-18
> > **Rebuttal (Q3)**
> >
> > **Q3: Overclaim. Section 4 starts by saying "Data type. While our framework above and our algorithms in § 4 are general for any data type, we focus on images in our experiments." However, it is not clear how this can be done for example for text because of the complexity of textual information and also for audio because of the lack of speech-based APIs. I would either remove this statement or try to discuss how this extension can be done.**
> >
> > **A3:** Thank you for the question. We discussed briefly in the original submission how PE can be implemented for text, and we added initial results showing that PE indeed works for texts below.
> >
> > * **How PE can be implemented for text.** We discussed in Section 3.3: “In our algorithm design and experiments, we use 2 APIs, both of which are either directly provided in the APIs of popular models (e.g., DALLE 2, Stable Diffusion **or can be easily implemented by adapting current APIs (e.g., using appropriate text prompts in GPT APIs)**”. Indeed, the natural language interface of large language models (LLMs) gives us even more flexibility in implementing RANDOM_API and VARIATION_API (for texts) than the diffusion models (for images). Let’s take email datasets as an example. For RANDOM_API, we can use the prompt “Please generate a random email:” to query the LLM. For VARIATION_API, we can use the prompt “Please rephrase the following email: <the original email>” to query the LLM. Such an approach works for any LLMs that provide chat capability (e.g., GPTs, LLaMAs). The above prompts are just a simple example; given LLMs’ powerful instruction-following capability, more prompt engineering can potentially give better results.
> >
> > * **Results of PE on texts.** We evaluate PE on text in our follow-up work. We compare PE with the recent ACL 2023 paper on DP synthetic text generation [1]. We follow their experimental setting:
> >
> >   * We take GPT2, GPT2-Medium, and GPT2-Large as the pre-trained model (to implement RANDOM_API and VARIATION_API in PE, or to DP-fine-tune in [1]).
> >   * We consider the Yelp review dataset and the private data.
> >   * We consider rating classification and category classification as the two downstream tasks for evaluating the utility.
> >   * We set the DP parameter epsilon=4.
> >
> > The following table shows the results.
> >
> >
> >
> > |                |             | Classification accuracy on rating | Classification accuracy on category |
> > |----------------|-------------|-----------------------------------|-------------------------------------|
> > | DP fine-tuning [1] | GPT2        | 66.56%                            | 74.78%                              |
> > | PE             | GPT2        | 65.62%                            | 74.75%                              |
> > | DP fine-tuning [1] | GPT2-Medium | 67.56%                            | 74.86%                              |
> > | PE             | GPT2-Medium | 66.85%                            | 75.01%                              |
> > | DP fine-tuning [1] | GPT2-Large  | 69.36%                            | 75.68%                              |
> > | PE             | GPT2-Large  | 67.49%                            | 74.64%                              |
> >
> > As a reference, directly training a DP classifier on the original data achieves 70.14% accuracy on rating and 76.44% accuracy on category. We can see that PE achieves similar (and sometimes better) accuracies compared to the state-of-the-art DP fine-tuning approach, no matter whether the pre-trained model is GPT2, GPT-Medium, or GPT2-Large.
> >
> > Nevertheless, given that this submission only focuses on images, we removed the above claim in the revision following the reviewer’s suggestion.
> >
> > [1] Yue, Xiang, et al. "Synthetic text generation with differential privacy: A simple and practical recipe." arXiv preprint arXiv:2210.14348 (2022).

---

> > ### Comment · Reviewer_yzwb · 2023-11-20
> > **Reviewer yzwb follow up on Q1 (concerns regarding the effect of the number of private samples)**
> >
> > > **In our experiments, the number of private samples varies widely, ranging from 100 (for the cat dataset) to 50,000 (for CIFAR10) and 302,436 (for Camelyon17). We observe that PE works reasonably well across all these cases.**
> >
> > This is a bit surprising. Can you please elaborate on why the performance of PE is independent of the number of samples? Also ideally we need to fix a dataset and vary its number of samples instead of seeing its effects across different datasets. For example, is it possible to vary the number of samples in your CIFAR-10 dataset and measure the performance of your method for each chosen number of samples?

---

> ### Author Response · Authors · 2023-11-18
> **Rebuttal (Q4, Q5, Q6)**
>
> **Q4: As this paper says in Section 4 "Foundation models have a broad and general model of our world from their extensive training data." so how do you make sure the images that you consider as private (for example CIFAR-10 in your case) are not used in the training of foundation models of used APIs? I think this somehow affects the practicality of the proposed method in practice. After reading the experiment section, it seems this paper does not use real-world APIs and instead, this paper tries to simulate this in a controlled environment by considering pre-trained models.**
>
> **A4:** It is a great point! Your understanding is correct. We added the following discussions to Section 7 of the revision.
>
> As PE does **not** provide DP guarantees for the pre-training data of the foundation models, one should be careful that data considered private is not used in pre-training. Depending on whether the APIs are from *black box models*, which can only be accessed through APIs  (e.g., DALLE3), or *local models*, whose weights and architectures are accessible by the users (e.g., Stable Diffusion), it has different implications.
> * *Using APIs from black box models.* Since most black box models do not reveal their training dataset, it is safer to only consider private data that was never been shared or posted online. This includes scenarios where hospitals generate synthetic medical images using their proprietary patient records.
> * *Using APIs from local models.* For local models, we have full control over the model weights and architectures. We can pre-train the models on data that surely has no overlap with the private data. In all experiments of the paper, we use local models including Improved Diffusion and Stable Diffusion. We directly take the pre-trained models from prior work, and we make sure that the private data and the pre-training data have no overlap.
>
>
> **Q5: The claim of the proposed method being easier to deploy than existing methods is not supported. I would at least empirically validate this by measuring its costs in comparison to existing ones.**
>
> **A5:** In Appendix O of the original submission and the revision, we provide experimental results showing that the **computational cost** of PE can be smaller than training-based methods under the same setting (e.g., using the same pre-trained model, generating the same number of generated samples). This means that if practitioners want to run the APIs locally (i.e., downloading the foundation models and running the APIs locally without using public API providers), PE provides computational benefits.
>
> **Q6: It would be good to also talk about the limitations of DP synthetic data in general especially when data is distributed across users and cannot be centralised in one place.**
>
> **A6:** Thanks for the great point. We agree that decentralized settings are more challenging to solve. In Section 6 of the original submission, we list it as one future work: “Solving DPSDA in the Local/Shuffle DP model and in federated learning settings.” We believe it is interesting to study how to extend PE to federated learning settings,
> In fact, PE is quite amenable to adapting to decentralized settings. Suppose each client/user has one private sample for simplicity. If we share the current iteration of generated samples with each user (these are differentially private anyway), then the user can vote for the nearest sample. These votes can be aggregated by a central server either by employing Differentially Private Frequency Estimation under Local Differential Privacy (such as in [1]) or by using frequency estimation under the shuffle-DP model which further amplifies the privacy guarantees as shown in [2].
>
>
> [1] Private Frequency Estimation via Projective Geometry. Feldman et al. ICML 2022.
>
> [2] Hiding Among the Clones: A Simple and Nearly Optimal Analysis of Privacy Amplification by Shuffling. Feldman et al. FOCS 2021.

---

> > ### Comment · Reviewer_yzwb · 2023-11-20
> > **Reviewer yzwb follow up on Q5 (concerns regarding computational cost)**
> >
> > Thanks for pointing me to the computational costs of your method and the existing ones in Appendix O.
> >
> > After looking at your results and discussion in Appendix O, I have a couple of concerns.
> > It is difficult to understand Figure 42 as there is no detailed analysis of the setting/results and the computational time of each method is represented differently:
> > 1. As shown in Figure 42, the generation time of DP-Diffusion is very small. Does this mean once the DP-Diffusion is fine-tuned you can generate unlimited samples very fast?
> > 2. Is the generation time of your method equal to the total time of all colours?
> > 3. How does the running time of your method scale with the number of generated samples?
> > 4. Why is it fair to compare the timing of your method with finetuning+generation time of DP-Diffusion as opposed to only its generation time if the fine-tuning needs to be done one time only?
> > 5. The running time of your method should be computed based on using real-world APIs and take into account the delay/corruption that might happen. But it seems you again simulate it locally (?)

---

> > > ### Comment · Reviewer_yzwb · 2023-11-20
> > > **Reviewer yzwb follow up on Q4 (concerns regarding your claim of using APIs)**
> > >
> > > > **Using APIs from black box models. Since most black box models do not reveal their training dataset, it is safer to only consider private data that was never been shared or posted online.**
> > >
> > > > **Using APIs from local models. For local models, we have full control over the model weights and architectures. We can pre-train the models on data that surely has no overlap with the private data.**
> > >
> > > > **In all experiments of the paper, we use local models including Improved Diffusion and Stable Diffusion. We directly take the pre-trained models from prior work, and we make sure that the private data and the pre-training data have no overlap.**
> > >
> > > Thanks for confirming that you use local models in all experiments instead of real-world APIs and confirming that one should be careful that data considered private is not used in pre-training. Therefore, I would suggest toning down and removing most of your claims regarding using real-world APIs as 1) you did not provide a strong application/example where using APIs do not violate your privacy guarantees (I do not think this is the case in your hospital example); 2) you have not done any experiments using APIs.

---

> > > ### Author Response · Authors · 2023-11-21
> > > **Reply to follow-up questions (Q5)**
> > >
> > > Thank you for the follow-up questions! All these are great points! We answer them below.
> > >
> > > **Q5:** Sorry for the confusion in the presentation of this result. Here, we first provide an overview of the rationale behind this evaluation, and then provide answers to all your questions (we reorder the questions based on their dependencies).
> > >
> > > Here we consider the setting where the user has a private dataset of size $N$ and a pre-trained model (hosted locally, running on GPUs), and wants to generate a total number of $N$ synthetic data samples. For the DP fine-tuning baseline, the procedure includes (1) DP fine-tuning the pre-trained model, and then (2) using the fine-tuned model to generate $N$ synthetic samples. Therefore, we break the time into these two parts. For PE, the procedure includes running PE with $N_{syn}=N$. PE includes the steps of RANDOM_API, VARIATION_API, nearest neighbor search, and feature extraction. Therefore, we break down the time of PE into these four parts. **Note that the experimental setting here follows exactly the experiments in Section 5.1.1, where we see that PE has a better privacy-utility trade-off than the DP fine-tuning baseline.**
> > >
> > > To generate more samples beyond $N$ samples, as the reviewer points out, the DP fine-tuning method can directly use the fine-tuned model to generate more samples without more fine-tuning. PE can use the approach in Section 4.2, where we pass the generated $N$ samples through VARIAION_API to generate more samples (this is what we did exactly in the experiments in Section 5.1.3). Note that, for the same size of the model, the same number of generated samples, the same diffusion sampler, and the same total number of denoising steps, **the running time of PE to generate more samples is smaller than the DP fine-tuning baseline.** The reason is as follows. Let’s say we use a diffusion sampler with 100 steps in total. In the DP fine-tuning baseline, generated samples must take all 100 steps (i.e., starting from Gaussian noise, passing the image through the diffusion model 100 times iteratively to get the final generated samples). For PE, VARIATION_API is implemented by SDEdit (Meng et al., 2021) (see Section I), where we add noise to input images and let the diffusion model denoise them *starting from the middle of the diffusion process* (e.g., adding Gaussian noise to the image, then passing the image through the diffusion model starting from 20th denoising step for all the rest 80 steps iteratively to get the final generated samples). In other words, each generated image does not need to go through all 100 steps, but only a fraction of the steps (80 steps in the above example). This is the only difference between the procedures of generating more samples in the DP fine-tuning baseline and PE. **Therefore, excluding the time for generating more samples is only in favor of the baseline, not PE.**
> > >
> > > ---
> > >
> > > Now, we answer your questions below.
> > >
> > > **1. Why is it fair to compare the timing of your method with finetuning+generation time of DP-Diffusion as opposed to only its generation time if the fine-tuning needs to be done one time only?**
> > > **2. As shown in Figure 42, the generation time of DP-Diffusion is very small. Does this mean once the DP-Diffusion is fine-tuned you can generate unlimited samples very fast?**
> > > **3. How does the running time of your method scale with the number of generated samples?**
> > >
> > > We answer the above three questions together as they are related. As explained above, we consider the running time of the steps users need to do when they want to generate $N$ samples given a private dataset of size $N$ and a pre-trained model. To generate even more samples, PE is NOT more costly than the DP fine-tuning baseline, when the pre-trained model is the same.
> > >
> > > **4. Is the generation time of your method equal to the total time of all colours?**
> > >
> > > Yes, that is correct. PE does not have a ``fine-tuning’’ stage.
> > >
> > > **5. The running time of your method should be computed based on using real-world APIs and take into account the delay/corruption that might happen. But it seems you again simulate it locally (?)**
> > >
> > > As the reviewer pointed out in the other question, all our experiments use local models. In particular, in this experiment, we follow the setting in Section 5.1.1, where the user has access to an ImageNet-pre-trained diffusion model locally and wants to generate DP synthetic data for CIFAR10. The takeaway is that “even if practitioners want to run the APIs locally (i.e., downloading the foundation models and running the APIs locally without using public API providers), there are still benefits of using PE” in terms of computation cost (mentioned in Section O).
> > >
> > > ---
> > >
> > > Sorry again that the description was not clear enough. We updated the revision with more explanations. Please let us know if you have more questions.

---

> ### Author Response · Authors · 2023-11-20
> **Follow-up**
>
> Dear Reviewer yzwb,
>
> Thank you again for your valuable time in reviewing our paper! Have we adequately addressed all your concerns? According to your suggestions, we added the results of PE on text, and the results of the DP fine-tuning baseline. We also addressed your other concerns in the rebuttal and the revision.
>
> If you have any further questions, please kindly inform us, and we will be happy to elaborate further before the Nov. 22 deadline.
>
> Thank you!
>
> The authors

---

> ### Author Response · Authors · 2023-11-21
> **Reply to follow-up questions (Q4, Q1)**
>
> **Q4:** Firstly, please allow us to deviate a bit and clarify what we meant by “APIs”. When saying “APIs”, we do not exclusively refer to APIs of the black box models (e.g., DALLE2), but also include the APIs of local models. For example, Stable Diffusion is a local model, but it provides a bunch of APIs that users can directly use https://huggingface.co/docs/diffusers/api/pipelines/stable_diffusion/overview. PE applied on these local APIs also provides the benefit that users do not need to train a model.
>
> Now, we go back to the question. We agree with the reviewer that we should clarify earlier in the paper that we only experimented with APIs of local models instead of public APIs. We added a clarification in the introduction and fixed relevant statements that may create confusion.
>
> We also want to clarify the point “you did not provide a strong application/example where using APIs do not violate your privacy guarantees”. Does the reviewer refer to the “privacy guarantees” about the pre-training data of the foundation model, or the private data used to run the PE algorithm? If it is the latter, we want to clarify that, for the private data used to run the PE algorithm, we provide DP guarantees even from the API provider. See Section 4.3 and A1.2 to Reviewer wkXZ for the discussions.
>
>
>
> **Q1:** When setting $N_{syn}/N_{priv}=1$, a smaller number of private samples $N_{priv}$ would mean that the number of generated samples from PE $N_{syn}$ is also smaller. These $N_{syn}$ samples could be of high quality, but there are fewer of them. That being said, the performance of PE is NOT *independent* of the number of samples. To generate more samples, we would then need to rely on the approach in Section 4.2, where we pass the samples through VARIATION_API to ”augment” these $N_{syn}$ samples (See Section 5.1.3 for the results).
>
> We agree with the reviewer that it would be interesting to see how the results of PE change with respect to different numbers of private samples $N_{priv}$ *of the same dataset*. We launched the experiments and will update the results here once they are done. In the meantime, if the reviewer has other concerns and/or suggestions about this question/experiment, please let us know!

---

> > ### Author Response · Authors · 2023-11-22
> > **Experiment results on different number of private samples**
> >
> > Dear Reviewer yzwb,
> >
> > We thank the reviewer again for this insightful suggestion! We conduct the following experiment: On CIFAR10, we vary the number of samples $N_{priv}$ by sub-sampling. For each $N_{priv}\in\\{50000, 20000, 10000, 5000\\}$:
> > * We fix $N_{syn}/N_{priv}=1$ when running PE
> > * After PE is done, we use the approach in Section 4.2 and Section 5.1.3 to augment the number of generated samples back to 50000, and compute the FID between the generated samples and CIFAR10
> > * All other hyper-parameters are set the same as the CIFAR10 experiments in Figure 3.
> >
> > The results are:
> > * When $N_{priv}=50000$ (original CIFAR10), FID=7.87
> > * When $N_{priv}=20000$ (2.5x smaller), FID=10.47
> > * When $N_{priv}=10000$ (5x smaller), FID=12.51
> > * When $N_{priv}=5000$ (10x smaller), FID=18.60
> >
> > **So the reviewer’s intuition is correct: larger $N_{priv}$ does helps the performance of PE, similar to the observation in DP-SGD [1].** We hypothesize that the reason is as follows. When we set  $N_{syn}/N_{priv}=1$, although the signal-noise ratio in the DP Nearest Neighbors Histogram remains constant (as we explained previously in Q1), larger $N_{syn}$ does allow the generated samples to explore larger space, and therefore it is more likely to get a sample closer to the private data, which helps the convergence of PE. Our theorem in Appendix D also shows that increasing the number of variations (which is controlled by $N_{syn}$) speeds up the convergence. Though the configuration $N_{syn}/N_{priv}=1$ might not be optimal, which we look forward to explore in the future.
> >
> > We added the above experiments to Appendix M. Thank the Reviewer yzwb very much again for suggesting this experiment, which provides important new insights to the algorithm.
> >
> > **We have now concluded all the questions and experiments you suggested.** If you have more questions, please feel free to ask before the discussion phase ends in 15 hours.
> >
> > [1] Anil, Rohan, Badih Ghazi, Vineet Gupta, Ravi Kumar, and Pasin Manurangsi. "Large-scale differentially private BERT." arXiv preprint arXiv:2108.01624 (2021).

---

> ### Author Response · Authors · 2023-11-23
>
> Dear Reviewer yzwb,
>
> Have we addressed your final questions sufficiently? The discussion phase ends in 3.5 hours. If we have addressed your concerns satisfactorily, please consider raising the score.
>
> Thank you again for your valuable time and suggestions!
>
> The Authors

---

> > ### Comment · Reviewer_yzwb · 2023-11-23
> > **I read all your responses, and this is my final response**
> >
> > Glad to hear my suggestions were helpful.
> >
> > I think the main two limitations of this work are: 1) using local APIs instead of real APIs as it is almost impossible to understand whether data considered private is not used in the pre-training of models behind real APIs; 2) you need to always go through at least a subset of steps to generate images as opposed to existing training-based ones that we just need to run inference.

---

### Official Review · Reviewer_eX7J · 2023-11-01

**Soundness:** 3 good
**Presentation:** 4 excellent
**Contribution:** 3 good
**Rating:** 6
**Confidence:** 4

**Summary:**

This paper studies the differentially private (DP) generation of image data through the black-box usage, namely via APIs, of pre-trained foundation generative models like Stable Diffusion. In particular, this work proposes using evolutionary algorithms that progressively refine the synthetic dataset/distribution to better align with the real private distribution. The real private distribution is represented as a histogram, , which quantifies how frequently each current synthetic sample is the nearest neighbor (determined by l2 distance in an embedding space) to real data samples from the private dataset. To achieve DP, noise is added to the histogram. Experimental evidence shows that the suggested application of pre-trained generative APIs yields noticeable improvements compared to previous methods that either do not use public data or utilize it within a pre-training & fine-tuning setup.

**Strengths:**

- The paper is generally well-written and easy to follow.
- The idea of exploiting pre-trained foundation model APIs is natural and practical
- The experimental findings are mostly encouraging and show promising results. The results are extensive and cover various key aspects regarding the usage of the proposed approach.

**Weaknesses:**

- The core DP component of the approach (i.e., adding DP noise to the histogram) is largely an existing idea/mechanism, which slightly diminishes the overall technical contribution of this submission.

- While the proposed approach demonstrates encouraging results on Camelyon17 dataset where the distribution shifts between the public and private distribution is relatively large, its performance may start to fall short compared to the more straightforward pre-training and fine-tuning paradigm (as the curves look not saturating and only one privacy level is presented).  A more in-depth exploration of the limits (in terms of tolerance to distribution shifts) of the pure API method (potentially by considering other medical datasets) and a discussion on potential refinements or extensions of the proposed approach would further strengthen the submission.

**Questions:**

- Several hyperparameters appear to be crucial for the proposed approach, including the threshold for DP NN and the lookahead degree. It would be helpful to provide further discussion on the selection of these hyperparameters. Specifically, insights into their transferability across different datasets and any computational trade-offs in practical scenarios would be beneficial.

- I appreciate the qualitative results, such as the generated samples and their NN among real data. However, I would expect additional quantitative evaluation, perhaps in the form of histograms or average results, for the NN distances (both from real to closest generated and from generated to closest real). This may help provide critical insights into the fidelity of the generated samples, the approximated data likelihood, and the empirical privacy leakage.

---

> ### Author Response · Authors · 2023-11-18
> **Rebuttal (Q1, Q2)**
>
> Thank you so much for the great suggestions and comments! Please see the answers to all your questions below.
>
>
> **Q1: The core DP component of the approach (i.e., adding DP noise to the histogram) is largely an existing idea/mechanism, which slightly diminishes the overall technical contribution of this submission.**
>
> **A1:** We agree with the reviewer that “adding DP noise to the histogram” is an existing idea. But we also want to mention that it is only one step among the other components we proposed, and the way we construct the histogram (using private samples to vote for the nearest generated samples) is different from the usual application of this idea (e.g., DP density estimation where the histogram is constructed by binning the private samples). Overall, our contributions are not on “adding DP noise to the histogram”, but on the following aspects:
> * **New problem:** We highlight the importance of studying DP Synthetic Data via foundation model APIs.
> * **New framework:** We propose Private Evolution, a new framework for generating DP synthetic data, that only requires APIs of the foundation models and does not need any model training.
> * **Promising results:** Private Evolution can even outperform the SOTA DP-fine-tuning-based approaches on the privacy-utility trade-off in the CIFAR10 experiments.
>
> **Q2: While the proposed approach demonstrates encouraging results on Camelyon17 dataset where the distribution shifts between the public and private distribution is relatively large, its performance may start to fall short compared to the more straightforward pre-training and fine-tuning paradigm (as the curves look not saturating and only one privacy level is presented). A more in-depth exploration of the limits (in terms of tolerance to distribution shifts) of the pure API method (potentially by considering other medical datasets) and a discussion on potential refinements or extensions of the proposed approach would further strengthen the submission.**
>
> **A2:** Following your suggestion, we conducted a new experiment showing how PE performs under different levels of distribution shifts. To do that, we take the Camelyon17 dataset and modify the *saturation* of the images to create a sequence of datasets, each with a different saturation change. This way, we create a sequence of datasets with different levels of distribution shifts from ImageNet. We decided to use this methodology because it is: (1) more controllable, as we can use saturation change to control the level of the distribution shifts, (2) more comparable, as these datasets are derived from the same base dataset, and (3) comprehensive, as we can show the performance under *multiple* levels of distribution shifts.
>
> The results are shown in Figure 30 in the revision. We can see that among all levels of distribution shifts we test here, the FID scores decrease with more PE iterations. This means that PE is effective in pushing the generated distribution towards private data. At the time we post the rebuttal, the FID scores are still decreasing and have not converged yet; we will keep updating this figure as the experiment continues.
>
> However, as we discussed in Section 5.1.2 and Section 6, larger distribution shifts do make the convergence slower and the results tend to be worse compared to the DP fine-tuning approaches. One possible improvement is to consider better variation degree ($v$ in Section 3.3) choices. Currently, we use a single variation degree in each step. As the theoretical analysis in Section D suggested, using multiple variation degrees (so that the variations can cover the entire space no matter whether the distribution shift is large or small) benefits convergence. It would be interesting to explore this direction in future work.

---

> > ### Author Response · Authors · 2023-11-18
> > **Rebuttal (Q3, Q4)**
> >
> > **Q3: Several hyperparameters appear to be crucial for the proposed approach, including the threshold for DP NN and the lookahead degree. It would be helpful to provide further discussion on the selection of these hyperparameters. Specifically, insights into their transferability across different datasets and any computational trade-offs in practical scenarios would be beneficial.**
> >
> > **A3:** In Appendix M of the original submission and the revision, we have conducted ablation studies on both the threshold and the lookahead degree and discussed the rationale behind our selection. We supplemented more discussions in the revision according to the reviewer’s suggestion. The results and takeaway messages are summarized below.
> > * **Lookahead degree:** We conducted the ablation study across a wide range of values (0~12) (see Figure 29 in the original submission, Figure 38 in the revision). We see that higher lookahead degrees monotonically improve the FID score. However, the marginal benefits diminish as the lookahead degree goes beyond 8, and a higher lookahead degree increases the number of API calls. In all our experiments, we set lookahead degree=8, and it works well across all datasets we evaluated. In practice, we suggest users set the lookahead degree as the highest value within their computation or API budget.
> > * **Threshold:** We conducted the ablation study across a wide range of values (0~15) (see Figure 31 in the original submission, Figure 40 in the revision). We can see that a large threshold results in faster convergence at the beginning. This is because, in the early iterations, many samples are far away from the private data. A larger threshold can effectively remove those bad samples that have a non-zero histogram count due to the added DP noise. However, at a later iteration, the distribution of generated samples is already close to the private data. A large threshold may potentially remove useful samples (e.g., the samples at low-density regions such as classifier boundaries). This may hurt the generated data, as shown in the increasing FID scores at threshold=15. Empirically, we found that setting threshold $\in[1,2]\sigma$ ($\sigma$ is the noise multiplier) works well across all datasets we evaluated. These results also suggest that an adaptive threshold that gradually decreases might work better, which we plan to study in future work. Note that this parameter does not affect computational cost.
> >
> > **Q4: I appreciate the qualitative results, such as the generated samples and their NN among real data. However, I would expect additional quantitative evaluation, perhaps in the form of histograms or average results, for the NN distances (both from real to closest generated and from generated to closest real). This may help provide critical insights into the fidelity of the generated samples, the approximated data likelihood, and the empirical privacy leakage.**
> >
> > **A4:** Thanks for the great idea! Following your suggestions, we plotted the distributions of the distances between (1) generated samples and their nearest real samples and (2) real samples and their nearest generated samples, on both CIFAR10 (Figures 17, 18) and Camelyon17 (Figures 27, Figure 28), for generated samples from different PE iterations. We plot the distribution in CDFs instead of histograms, as CDFs are easier to read when plotting multiple distributions in one figure.
> >
> > Two key observations are: (1) During the early PE iterations, the inception distances tend to decrease. This means that PE is effective in pushing the generated distribution to be closer to the private distribution. (2) However, as PE continues, the inception distances stop decreasing. It is expected, as DP upper bounds the probability of reconstructing any sample in the private dataset. Please refer to the revision for more discussions.

---

> > ### Author Response · Authors · 2023-11-22
> > **Update on the results of distribution shifts**
> >
> > Dear Reviewer eX7J,
> >
> > The experiment has now finished and updated in Figure 30. The message remains the same as our original posting: We can see that among all levels of distribution shifts we test here, the FID scores decrease with more PE iterations. This means that PE is effective in pushing the generated distribution towards private data.
> >
> > **We have now concluded all the questions and experiments you suggested.** If you have more questions, please feel free to ask before the discussion phase ends in 15 hours.

---

> ### Author Response · Authors · 2023-11-20
> **Follow-up**
>
> Dear Reviewer eX7J,
>
> Thank you again for your valuable time in reviewing our paper! Have we adequately addressed all your concerns? According to your suggestions, we added the experiments of PE with various distribution shifts, and distribution plots of the nearest neighbor distances. We also addressed your other concerns in the rebuttal and the revision.
>
> If you have any further questions, please kindly inform us, and we will be happy to elaborate further before the Nov. 22 deadline.
>
> Thank you!
>
> The authors

---

> ### Author Response · Authors · 2023-11-23
>
> Dear Reviewer eX7J,
>
> Have we addressed your concerns sufficiently? We have not heard back from you since we posted the rebuttal 4 days ago, and the discussion phase ends in 5 hours. If we have addressed your concerns, please consider raising the score.
>
> Thank you again for your valuable time and suggestions!
>
> The Authors

---

> > ### Comment · Reviewer_eX7J · 2023-11-23
> >
> > Dear Authors,
> >
> > Thank you for your detailed rebuttal and the additional experimental results. Your response has addressed most of the questions I initially raised.
> >
> > The approach to investigating distribution shift, although it differs from what I had expected (I anticipated an analysis more focused on the concept/semantic shift, which might also resonate with the concerns of other reviewers), still presents a valuable perspective. Furthermore, I perceive a potential limitation in the real-world applicability of your approach, especially concerning private data that significantly deviates in distribution from the pre-training data of foundation models. Despite this, I believe that your paper offers a viable potential solution and provides valuable insights that could propel future research. Consequently, I consider this submission to be above the acceptance threshold for ICLR from my perspective.

---

### Author Response · Authors · 2023-11-18
**Rebuttal Summary**

We sincerely thank all reviewers for their constructive feedback and suggestions, which are very helpful to us. We are encouraged that the reviewers recognize that (1) exploiting APIs for DP synthetic data is a practical and nice application of foundation models (eX7J, yzwb, wkXZ, vg5F), (2) the proposed approach is novel and easier to deploy (yzwb, wkXZ), (3) the results are promising and extensive (eX7J, yzwb, wkXZ, vg5F), and (4) the writing is clear and easy to follow (eX7J, yzwb, vg5F).

Following the reviewers’ suggestions, we added 7 new experiments/results, and we addressed the questions in both the rebuttal and the revision (modifications are in blue). The added experiments/results are:

* The experiments of Private Evolution on text (yzwb, wkXZ).
* The experiments of Private Evolution on Camelyon17 dataset with epsilon=10 (wkXZ,vg5F).
* The experiments of Private Evolution under different levels of distribution shifts (eX7J).
* The histogram of nearest neighbor distances (both from real samples to their closest generated samples and from generated samples to their closest real samples) (eX7J).
* Generated samples that get high and low counts in DP NN Histogram (vg5F).
* Standard deviation of the counts in the DP NN Histogram (vg5F).
* Baseline results of directly DP fine-tuning the classifiers (yzwb).

In addition, we updated the generated images on CIFAR10 in Figures 3, 13, 15, and 16. The images should have come from PE iteration 7 with FID 7.87 and epsilon=0.67, but we mistakenly take the images from PE iteration 5 with FID 8.71. In other words, we put worse images (in terms of FID) than what they should be in the original submission.

If the reviewers have more questions, please feel free to post them and we are happy to elaborate more.

---

### Author Response · Authors · 2023-11-22
**To AC, Reviewers, and Public Readers**

As the discussion phase comes to an end, we want to use this opportunity to reiterate **the motivation and the goal of our paper, and discuss the limitations and outlooks of the work.**

DP fine-tuning is very successful in DP synthetic data generation. However, it does not mean that it will always be the most suitable solution. Our personal, humble opinion is that, *we need to look ahead and think about techniques for the requirements in the future.* We see the trend that (1) Foundation models trained with large datasets are more and more powerful; (2) The most powerful ones are mostly closed-source, whose weights and gradients are not released, which renders DP fine-tuning infeasible; the other open-sourced foundation models are resource-intensive to DP fine-tune due to their large sizes. Non-DP ML applications have already benefited a lot from these foundation models *by using their APIs*, but the field of DP synthetic data lags behind. Therefore, **we see an urgent need to develop API-based DP synthetic data methods tailored to this new trend.** We hope that this work can draw the community’s attention to this topic.

With the constraint of relying solely on inference APIs, the problem is much more challenging. Expecting an immediate solution mirroring all the advantages of DP-SGD with just API access is ambitious. Private Evolution is the first trial toward the goal and may *not* be the ultimate solution. We appreciate the reviewers for discussing the limitations. **We summarize the key points here, and we hope to discuss and explore them with the community together.**

* *PE relies on foundation models trained on “largely spanned” public datasets*. Thank reviewer vg5F for this point. We agree with the intuition and we discussed it in Figure 2 in the original submission. But we want to point out that, this could be less of a concern in the future, *as the trend is to train foundation models on larger and larger datasets*, exemplified by the series of models in Stable Diffusion, DALLE, GPT, LLaMA, and so on. We also admit the importance of improving the PE algorithm to alleviate this requirement (Section 6).

* *PE does not protect the privacy of pre-training data*. Thank reviewer yzwb for this point. We want to mention that the DP guarantee of PE is rigorous, where the DP neighboring dataset is defined *with respect to the data $S_{priv}$ used in the PE algorithm*. That being said, PE does *not* address the privacy of *pre-training data* of foundation models, which is a different goal. PE has no control over the pre-training data—any privacy breaches in the pre-training data are attributable to the data holder (e.g., leaking the data publicly) or the foundation model developer (e.g., using data without permission). However, as a PE user, it is advisable to ensure no overlap between pre-training data and $S_{priv}$ for liability reasons. Two scenarios arise:

  * *Using APIs from local (open-source) models.* We have full control of the weights and therefore can make sure that the pre-training data and private data have no overlap.
  * *Using APIs from blackbox (API-based) models such as DALLE3.* While checking pre-training data is challenging, ensuring no overlap between $S_{priv}$ and pre-training data is still possible. For instance, a hospital A sharing DP synthetic medical records can safely run PE if it has never released medical records to any other party, making it impossible for those records to be in the pre-training data of any foundation model. One might argue that the pre-training data might contain medical records from *other* hospitals, but it is not the liability of hospital A and hostipal A has no control over those records. PE ensures the privacy of the medical records $S_{priv}$ on hostipal A’s behalf.

    However, we want to emphasize that we cannot overlook the privacy concerns associated with pre-training data (e.g., the medical records from other hospitals in the above example). Though being out of the scope of this work, it is a significant issue impacting all applications of foundation models, as acknowledged by the community [2,3]. It is crucial to explore improved protection and auditing methods in response to this challenge and it requires foundation model developers to work together.

Finally, we want to mention that despite the above challenges to solve, we believe the direction of “DP synthetic data via foundation model APIs” is promising, exemplified by the fact that **PE already outperforms prior DP-fine-tuning-based SOTA by a large margin on the standard “ImageNet->CIFAR10” benchmark.** We believe in the practical importance of this direction, and we are excited about the research opportunities opened up by the work. We hope to explore them with the community together!


[1] Wang, Boxin, et al. "DecodingTrust: A Comprehensive Assessment of Trustworthiness in GPT Models."

[2] Carlini, Nicolas, et al. "Extracting training data from diffusion models."

---

### Meta-Review · Area_Chair_LTon · 2023-12-01

**Metareview:**

This paper created a new question, namely, "can we use powerful API models to generate synthetic data that mimics privacy-sensitive data?". The question is timely, and their approach seems reasonable. There was a concern about the algorithm being not novel. I see that this algorithm is based on a published work, so in terms of the algorithm itself the contribution seems limited. There was also a concern about the case where there is a large domain gap between the public dataset which API models are trained on and private datasets. The authors did show improvement over DP-Diffusion in generating Camelyon17 from Imagenet-trained APIs. The last reviewer's concern was that Imagenet has a class that mimics the texture of images in Camelyon17, so the domain gap there might not be as large as the authors think. This seems an intriguing observation. Perhaps authors could add some failure scenarios where this approach won't work to be fair, also to give some extra caution to the future users of this algorithm. Overall, however, the merits of the suggested method seems to outweigh than the downsides of the proposed method. Three reviewers agreed on this, and therefore I suggest acceptance to ICLR2024.

**Justification For Why Not Higher Score:**

Poster seems appropriate for this paper, because of the two fore-mentioned concerns raised by the reviewers. I also do not see any particular high levels of excitements about this paper from all four reviewers.

**Justification For Why Not Lower Score:**

Lower means not accepting. I think this paper is good enough to be accepted, both in terms of their contributions, novelty, and received reviewer's opinions.

---

### Decision · Program_Chairs · 2024-01-16

Accept (poster)